

# CALIPSO (IIR-CALIOP) Retrievals of Cirrus Cloud Ice Particle Concentrations

David L. Mitchell[1], Anne Garnier[2,3], Jacques Pelon[4], and Ehsan Erfani[5]

[1]Desert Research Institute, Reno, 89512-1095, USA

[2]Science Systems and Applications, Inc., Hampton, Virginia, USA

[3]NASA Langley Research Center, Hampton, Virginia, USA

[4] Laboratoire Atmosphères, Milieux, Observations Spatiales, Sorbonne Université, CNRS, Paris, France

[5]George Mason University, Fairfax, Virginia, USA

*Correspondence to*: David Mitchell (david.mitchell@dri.edu)

**Abstract.**  A new satellite remote sensing method is described whereby the sensitivity of thermal infrared wave resonance absorption to small ice crystals is exploited to estimate cirrus cloud ice particle number concentration N, effective diameter $D_e$, and ice water content IWC.  This method uses co-located observations from the Infrared Imaging Radiometer (IIR) and from the CALIOP (*Cloud and Aerosol Lidar with Orthogonal Polarization*) lidar aboard the CALIPSO (*Cloud-Aerosol Lidar and Infrared Pathfinder Satellite Observation)* polar orbiting satellite,

employing IIR channels at 10.6 μm and 12.05 μm.  Using particle size distributions measured over several flights of the TC4 (Tropical Composition, Cloud and Climate Coupling) and the mid-latitudes SPARTICUS (Small Particles in Cirrus) field campaigns, we show for the first time that N/IWC is tightly related to $\beta_{eff}$; the ratio of effective absorption optical depths at 12.05 μm and 10.6 μm.  Relationships developed from in situ aircraft measurements are applied to $\beta_{eff}$ derived from IIR measurements to retrieve N.  This satellite remote sensing method is constrained by

measurements of $\beta_{eff}$ from the IIR and is by essence sensitive to the smallest ice crystals.  Retrieval uncertainties are discussed, including uncertainties related to in situ measurement of small ice crystals (D < 15 μm), which are studied through comparisons with IIR $\beta_{eff}$.  The method is applied here to single-layered semi-transparent clouds having a visible optical depth between about 0.3 and 3, where cloud base temperature is ≤ 235 K.  Two years of CALIPSO data have been analyzed for the years 2008 and 2013, with the dependence of cirrus cloud N and $D_e$ on

altitude, temperature, latitude, season (winter vs. summer) and topography (land vs. ocean) described.  The results for the mid-latitudes show a considerable dependence on season.  In the high latitudes, N tends to be highest and $D_e$ smallest, whereas the opposite is true for the tropics.  The frequency of occurrence of these relatively thick cirrus clouds exhibited a strong seasonal dependence in the high latitudes, with the occurrence frequency during Arctic winter being at least twice that of any other season.  Processes that could potentially explain some of these micro-

and macroscopic cloud phenomena are discussed.



# 1 Introduction

The microphysical and radiative properties of ice clouds are functions of the ice particle size distribution or PSD, which is often characterized by the PSD ice water content (IWC), a characteristic ice particle size and the ice particle number concentration N; all of which can be measured in situ using suitable instruments. To date, satellite
remote sensing methods can retrieve two of these properties; the PSD effective diameter $D_e$ and IWC. Most parameterizations of ice cloud optical properties in climate models are based on these parameters (e.g. Fu, 1996; 2007). However, the ice cloud PSD is not fully constrained by $D_e$ and IWC, and ice cloud optical properties at terrestrial wavelengths are not always well defined by $D_e$ and IWC (Mitchell et al., 2011a). Moreover, satellite retrievals of N would be useful for advancing our understanding of ice nucleation in the atmosphere. To realistically
predict $D_e$ in climate models, realistic predictions of ice crystal nucleation rates are essential since they determine $D_e$. Realistic satellite retrievals of N would provide a powerful constraint for parameterizing ice nucleation in climate models.

Retrievals of cloud microphysical properties from satellite have evolved considerably since the first developments using passive observations (Inoue, 1985; Parol et al., 1991; Ackerman et al., 1995). The advent of the A-Train has
enabled passive and active observations to be combined and more precisely analyzed to study the vertical structure of clouds and the atmosphere (Delanoë and Hogan, 2008, 2010; Deng et al., 2010, Garnier et al., 2012, 2013, Deng et al., 2013; Sourdeval et al., 2016). Such retrievals from satellite have been extensively compared to in situ observations (King et al., 2003, Deng et al., 2010, 2013) to validate these retrievals and to estimate representative microphysical parameters such as $D_e$ and IWC (or ice water path, IWP) that can be compared with corresponding
large scale model outputs to improve their cloud parametrizations and general climate applications (Eliasson et al., 2011; Stubenrauch et al., 2013; the Global Energy and Exchanges Process Evaluation Studies (GEWEX PROES) https://gewex-utcc-proes.aeris-data.fr/). This has advanced a convergence between in situ and satellite studies on ice clouds, where satellite studies do not suffer from certain aircraft probe limitations such as ice particle shattering (e.g. Field et al., 2006; Mitchell et al., 2010; Korolev et al., 2011; Cotton et al., 2013), and in situ studies do not depend
on certain relationships relating radiation to cloud properties (e.g. Delanoë and Hogan, 2008).

Recent progress has been made regarding efforts to retrieve N via satellite. The retrieval of N as a function of latitude and topography is of particular importance as it provides insight into specific physical processes controlling N. The satellite remote sensing study by Zhao et al. (2018) has advanced our understanding of the complex relationship between aerosol particles and cirrus clouds, showing the importance of homogeneous ice nucleation
under relatively clean (i.e. relatively low aerosol optical depth) conditions. A satellite retrieval for N has been proposed that builds upon the lidar-radar (hence DARDAR) retrieval described in Delanoë and Hogan (2008, 2010), as described in Gryspeerdt et al. (2018) and Sourdeval et al. (2018). Relating satellite retrievals of N and $D_e$ to





mineral dust observations (Gryspeerdt et al., 2018) and to cirrus cloud-aerosol modeling outcomes (Zhao et al., 2018) have yielded insights into the relative importance of homogeneous and heterogeneous ice nucleation in cirrus clouds as a function of aerosol concentrations.

This study describes a new approach for estimating cloud layer N, $D_e$, and IWC in selected semi-transparent cirrus

clouds. The technique uses co-located observations from the 10.6 μm and 12.05 μm channels on the Imaging Infrared Radiometer (IIR) aboard the CALIPSO (*Cloud-Aerosol Lidar and Infrared Pathfinder Satellite Observation*) polar orbiting satellite, augmented by the scene classification and extinction profile from CALIOP (*Cloud and Aerosol Lidar with Orthogonal Polarization*) and by interpolated temperatures from the Global Modeling Assimilation Office (GMAO) Goddard Earth Observing System Model, version 5 (GEOS 5) (Garnier et

al., 2012, 2013). CALIOP and IIR are assembled in a near-nadir looking configuration. The cross-track swath of IIR is by design centered on the CALIOP track where observations from the two instruments are perfectly temporally co-located. The spatial co-location is nearly perfect, as CALIOP samples the same cloud as the 1-km IIR pixel, but with three laser beam spots per km having a horizontal footprint of 90 m. While the IIR retrieves layer-average cloud properties, CALIOP vertically profiles cloud layers, thus providing estimates of representative cloud

temperature and the temperature dependence of N, $D_e$ and IWC.

In this paper, we compare IIR retrievals with in situ observations performed during two field campaigns conducted in the tropics and mid-latitudes and develop a method to derive cirrus microphysical parameters from CALIPSO observations. Using different assumptions, several formulations for this retrieval scheme are presented to illustrate the inherent uncertainties associated with the retrieved cloud properties. The objective of this work was not to

determine absolute magnitudes for the retrieved quantities, but rather to show how their relative differences vary in terms of temperature, cloud thickness, latitude, season and topography.

Section 2 describes the rationale for developing the retrieval method, along with the retrieval physics and methodology, and discusses several plausible assumptions and formulations. Section 3 describes the retrieval equation that incorporates CALIPSO observations, as well as retrieval uncertainties. In Sect. 4, retrieved layer-

average cloud properties are compared with corresponding cloud properties measured in situ. Different retrieval scheme formulations are used to illustrate the inherent uncertainties associated with retrieved and in situ cloud properties. In Sect. 5, IIR-CALIOP retrieval results for N and $D_e$ are reported for 2008 and 2013 during the winter and summer seasons for all latitudes. These results are discussed in Sect. 6, which includes comparisons with previous work using the combined radar-lidar approach (Sourdeval et al., 2018; Grysperdt et al., 2018). The

discussion also addresses the radiative significance of these cirrus cloud retrievals and a potential link between Arctic cirrus and mid-latitude weather. A brief summary of the results and concluding comments are ending the paper in Sect. 7.



## 2 Developing a satellite remote sensing method sensitive to N

### 2.1 Satellite retrievals from infrared absorption methods

It is widely recognized that the ratio of absorption optical depth from ice clouds, $\beta$, based on wavelengths in the thermal infrared domain at 12 μm and 11 μm (or similar wavelengths), is rich in cloud microphysical information

(Inoue, 1985; Parol et al., 1991; Cooper et al., 2003; Dubuisson et al., 2008; Heidinger and Pavolonis, 2009; Mishra et al., 2009; Mitchell and d'Entremont, 2009; Pavolonis, 2010; Mitchell et al., 2010; Cooper and Garrett, 2010; Garnier et al., 2012; Mitchell and d'Entremont, 2012; Garnier et al., 2013). These studies have used a retrieved $\beta$ to estimate the effective diameter $D_e$, IWP, the mass-weighted ice fall speed ($V_m$), the average fraction of liquid water in a cloud field, the relative or actual concentration of small ice crystals in ice PSDs and the cloud droplet number

concentration in mixed phase clouds. However, the main reason for the emissivity differences in satellite remote sensing channels centered on these two wavelengths was not understood until after the development of the modified anomalous diffraction approximation (MADA) that, to a first approximation, allowed various scattering/absorption processes to be isolated and evaluated independently (Mitchell, 2000; Mitchell et al., 2001; Mitchell, 2002; Mitchell et al., 2006). For wavelengths between 2.7 and 100 μm, the most critical process parameterized was wave

resonance, also referred to as photon tunneling (e.g. Nussenzveig, 1977; Guimaraes and Nussenzveig, 1992; Nussenzveig, 2002). This process was found to be primarily responsible for the cloud emissivity difference between these wavelengths (12 μm and 11 μm) in ice clouds, as described in Mitchell et al. (2010).

It was originally thought that $\beta$ resulted from differences in the imaginary index of ice, $m_i$, at two wavelengths ($\lambda$) near 11 μm and 12 μm, but it is actually due to differences in the real index of refraction, $m_r$. At these $\lambda$, $m_i$ is

sufficiently large so that most ice particles in the PSD experience area-dependent absorption (i.e. no radiation passes through the particle), and the absorption efficiency $Q_{abs}$ for a given ice particle will be ~ 1.0 for both $\lambda$ when $Q_{abs}$ is based only on $m_i$ (i.e. the $Q_{abs}$ predicted by Beer's law absorption or anomalous diffraction theory). The observed difference between $Q_{abs}(12$ μm$)$ and $Q_{abs}(11$ μm$)$ is due to differences in the photon tunneling contribution to absorption that primarily depends on $m_r$ (Mitchell, 2000). That is, $m_r$ is substantial when $\lambda = 12$ μm but is relatively

low when $\lambda = 11$ μm, producing a substantial difference between $Q_{abs}(12$ μm$)$ and $Q_{abs}(11$ μm$)$. Figure 1 shows the size dependence of the tunneling contribution for hexagonal columns at 12 μm.

The greatest tunneling contribution to absorption occurs when the ice particle size and wavelength are comparable and the real refractive index $m_r$ is relatively high relative to $m_r$ for the 11 μm wavelength (Mitchell, 2000). In this case $\beta$ is sensitive to the tunneling process and the relative concentration of small (this contribution is becoming

smaller than 10% for sizes larger than 60 μm) ice crystals in cirrus clouds. It is thus evident that this contribution is making $\beta$ well suited for detecting recently nucleated (small) ice crystals that primarily determine N (Krämer et al., 2009).





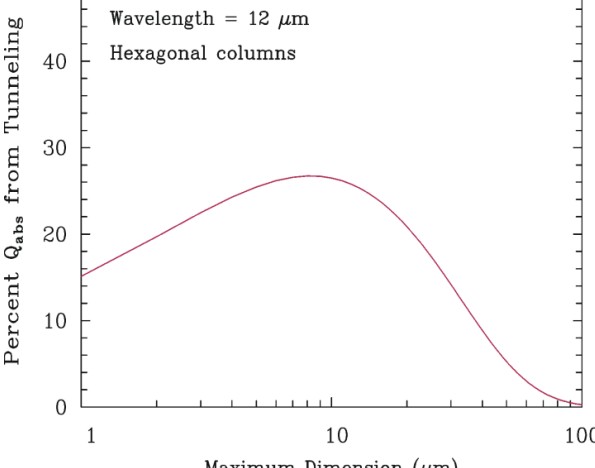

**Figure 1: Percent contribution of wave resonance absorption to the overall absorption efficiency at 12 µm wavelength as a function of maximum dimension D for hexagonal columns, as estimated by the MADA. It is decreasing to below 10% of its maximum for sizes larger than about 60 µm.**

### 2.2 Retrieving $\beta_{eff}$ from IIR and CALIOP observations

In this study, we use CALIPSO IIR channels at 10.6 µm and 12.05 µm and define β as

$$\beta = \tau_{abs}(12.05\ \mu m)/\tau_{abs}(10.6\ \mu m) \tag{1}$$

where $\tau_{abs}$ is the retrieved absorption optical depth for a given λ retrieved from the effective emissivity. However, what is retrieved is not exactly β, but a β that also includes the effects of scattering, defined as the effective β or $\beta_{eff}$. $\beta_{eff}$ is described analytically in Parol et al. (1991). For a given cirrus cloud, retrieved $\tau_{abs}(10.6\ \mu m)$ may be slightly less than $\tau_{abs}(11\ \mu m)$ since $m_i$ at 10.6 µm is less than $m_i$ at 11 µm, meaning that some Beer's law type absorption may contribute to emissivity differences between the 10.6 µm and 12 µm channels when PSD are sufficiently narrow. This acts to slightly extend the dynamic range of retrievals relative to the 11 µm-12µm channel combination (e.g. the limiting maximum $D_e$ retrieved will be greater using 10.6 and 12 µm relative to 11 and 12 µm). Although it is feasible to retrieve $D_e$ using the IIR 8.65 µm channel (e.g. Dubuisson et al., 2008; Garnier et al., 2013), the same channels are used for both $D_e$ and N to ensure self-consistent retrievals with respect to PSD moments. In addition, the analytical PSD calculations for $\beta_{eff}$ are more accurate when using the 12.05/10.6 combination relative to the 12/8.65 combination (Figs. 1b and 2b in Garnier et al., 2013, JAMC) since there is less multiple scattering with the 12.05/10.6 combination. IIR retrievals have a resolution of 1 km. Finally, IIR calibrated radiances are from the recently released Version 2 Level 1b products (Garnier et al., 2018). IIR retrievals





are based on the CALIPSO IIR Version 3 Level 2 track product (Vaughan et al., 2017). This product includes a scene typing built from the CALIOP Version 3 5-km cloud and aerosol layer products. The scene classification has been refined for this study to account for additional dense clouds in the planetary boundary layer reported in the CALIOP Version 3 333-m layer product. The methodology for retrieving CALIPSO IIR effective emissivity and

$\beta_{eff}$ from co-located CALIOP observations and IIR radiances is detailed in Garnier et al. (2012, 2013). IIR retrievals are further corrected to reduce possible biases, as described in the following sub-sections. Version 3 CALIOP cloud extinction coefficient profiles are used for some of the corrections. These improvements will be implemented in the next version 4 of the IIR Level 2 products.

For each IIR channel, $\tau_{abs}$ is derived from the effective emissivity, $\varepsilon$, as

$\tau_{abs} = - \ln (1 - \varepsilon)$ (2)

with

$$\varepsilon = \frac{R_m - R_{BG}}{R_{BB} - R_{BG}}$$ (3)

where $R_m$ is the measured calibrated radiance, $R_{BB}$ is the opaque (i.e. blackbody) cloud radiance, and $R_{BG}$ is the clear sky background radiance that would be observed in the absence of the studied cloud, as described in Garnier et al.

15 (2012).

The cloud emissivity $\varepsilon$ and the subsequent $\beta_{eff}$ are retrieved for carefully selected cirrus clouds, as described in Sect. 2.2.1. The retrievals are cloud layer average quantities as seen from space, whose representative altitude and temperature are estimated using additional information from CALIOP vertical profiling in the cloud, as presented in Sect. 2.2.3. As such, they can differ from in situ local measurements in the cloud.

**2.2.1 Cloud selection**

A number of selection criteria are defined for the robustness of the retrievals. In this study, the retrievals were applied only to single-layered semi-transparent cirrus clouds (one cloud layer in an atmospheric column) that do not fully attenuate the CALIOP laser beam, so that the cloud base is detected by the lidar. The cloud base is in the troposphere and its temperature is required to be colder than -38°C (235 K) to ensure that the cloud is entirely

composed of ice. This is likely to exclude liquid-origin cirrus clouds from our data set (Luebke et al., 2016). Because the relative uncertainties in $\tau_{abs}$ and in $\beta_{eff}$ increase very rapidly as cloud emissivity decreases (Garnier et al., 2013), the lidar layer-integrated attenuated backscatter (IAB) was chosen greater than 0.01 sr$^{-1}$ to avoid very large uncertainties at the smallest visible optical depths (ODs). This resulted in an OD range of about 0.3 to 3.0.



Similarly, clouds for which the radiative contrast $R_{BG} - R_{BB}$ between the surface and the cloud is less than 20 K in brightness temperature units are discarded. IIR observations must be of good quality according to the quality flag reported in the IIR Level 2 product (Vaughan et al., 2017).

### 2.2.2 Background radiance, $R_{BG}$

The brightness temperature $T_{BG}$ associated to the background radiance $R_{BG}$ is derived from the FASt RADiative (FAStRAD) transfer model (Dubuisson et al., 2005) fed by atmospheric profiles and skin temperatures from GMAO GEOS5 along with pre-defined surface emissivities inferred from the International Geosphere and Biosphere Program (IGBP) surface types and a daily updated snow/ice index (Garnier et al., 2012). For this study, remaining biases at 12.05 μm and 10.6 μm between FAStRAD model and observations are corrected using monthly maps of

mean differences between observed and computed brightness temperatures (called BTDoc) in clear sky conditions. The corrections are applied over ocean and over land with a resolution of 2 degrees in latitude and 4 degrees in longitude, by separating daytime and nighttime data. After correction, BTDoc is equal to zero on average for both channels.

### 2.2.3 Blackbody radiance, $R_{BB}$

In version 3 IIR products, $R_{BB}$ (and the associated blackbody temperature $T_{BB}$) is derived from the cloud temperature, $T_{caliop}$, evaluated at the centroid altitude of the CALIOP 532 nm attenuated backscatter profile using GMAO GEOS5 temperature (Garnier et al, 2012). In this study, a correction is further applied using CALIOP extinction profiles in the cloud layer as described in Sect. 3 of Garnier et al. (2015). The CALIOP lidar 532 nm

extinction profile in the cloud is used to determine an IIR weighting profile that is used to compute $R_{BB}$ as the weighted averaged blackbody radiance. The lidar vertical resolution is 60-m, and emissivity is weighted in a similar way with the 532 nm extinction profile. The weight of each 60-m bin is its emissivity at 12.05 μm attenuated by the overlying infrared absorption optical depth, normalized to the cloud emissivity.

In addition, the altitude and temperature associated to the layer average $\beta_{eff}$ are characterized using the centroid

altitude, $Z_c$, and centroid temperature, $T_c$, of the IIR weighting profile. Note that $T_c$ and $T_{BB}$ are not identical, but typically differ by less than a few tenths of Kelvins.

### 2.2.4 Estimated uncertainties

The uncertainty in $\beta_{eff}$, $\Delta\beta_{eff}$, is computed by propagating the errors in $\tau_{abs}(12.05\ \mu m)$ and $\tau_{abs}(10.6\ \mu m)$. These errors are themselves computed by propagating the errors in i) the measured brightness temperatures $T_m$ associated to $R_m$,





ii) the blackbody brightness temperature $T_{BB}$, and iii) the background brightness temperatures $T_{BG}$ (Garnier et al., 2015). The uncertainties in $T_m$ at 10.6 μm and 12.05 μm are random errors set to 0.3 K according to the IIR performance assessment established by the Centre National d'Etudes Spatiales (CNES) assuming no systematic bias in the calibration. They are assumed to be statistically independent. In contrast, the error in $T_{BB}$ is the same for both

channels, because the same cloud temperature $T_{BB}$ is used to compute $\tau_{abs}$(12.05 μm) and $\tau_{abs}$(10.6 μm). A random error of ± 2K in $T_{BB}$ is estimated to include errors in the atmospheric model. After correction for systematic biases based on differences between observations and computations in cloud-free conditions (see Sect. 2.2.2), the error in $T_{BG}$ is considered a random error, which is taken equal to the standard deviation of the corrected distributions of BTDoc. As a result, the uncertainty in $T_{BG}$ at 12.05 μm is set to ± 1K over ocean, and to ± 3K over land for both

night and day. Standard deviations of the distributions of [BTDoc(10.6 μm) - BTDoc(12.05 μm)] are generally smaller than 0.5 K. Accounting for the contribution from the measurements, which is estimated to be √2x0.3 = 0.45 K, this indicates that the biases in $T_{BG}$ at 10.6 and at 12.05 μm are nearly canceling out and can therefore be considered identical. More details about the uncertainty analysis can be found in the appendix.

The relative uncertainty $\Delta\beta_{eff}/\beta_{eff}$ is mostly due to random measurement errors, because systematic biases associated

with the retrieval of $\tau_{abs}$(12.05 μm) and $\tau_{abs}$(10.6 μm) tend to cancel when these are ratioed to calculate $\beta_{eff}$.

### 2.3.   Relating $\beta_{eff}$ to N/IWC and $D_e$ based on aircraft PSD measurements

Using aircraft data from the DOE ARM supported Small Particles in Cirrus (SPartICus) field campaign in the central United States (Mace et al., 2009) and the NASA supported Tropical Composition, Cloud and Climate Coupling (TC4) field campaign near Costa Rica (Toon et al., 2010), $\beta_{eff}$ was related to cirrus cloud microphysical

properties. The SPARTICUS field campaign, conducted from January through June of 2010 in the central United States (for domain size, see Fig. 2 in Mishra et al., 2014), was designed to better quantify the concentrations of small (D < 100 μm) ice crystals in cirrus clouds (Mace et al., 2009). Regarding SPARTICUS, the data set described in Mishra et al. (2014) was used, and the TC4 data is described in Mitchell et al. (2011b). Details regarding field measurements, the flights analyzed and the microphysical processing are described in these articles. PSD sampling

times during TC4 were often much longer than for SPARTICUS (< 2 minutes), with mean TC4 sampling times for horizontal legs within aged anvils being 10.56 minutes (Lawson et al., 2010; Mitchell et al., 2011b). This, along with more in-cloud flight hours during SPARTICUS, resulted in fewer PSD samples for TC4 relative to SPARTICUS. The PSDs were measured by the 2D-S probe (Lawson et al., 2006) where ice particle concentrations were measured down to 10 μm (5–15 μm size bin) and up to 1280 μm in ice particle length (when using "all-in" data

processing criteria). $\beta_{eff}$ was calculated from these PSDs using the method described in Parol et al. (1991) and Mitchell et al. (2010). This method was tested in Garnier et al. (2013, Fig. 1b) where $\beta_{eff}$ calculated from a radiative transfer model (FASDOM; Dubuisson et al., 2005) was compared with $\beta_{eff}$ calculated analytically via Parol et al.





(1991), with good agreement found between these two methods. More specifically, to calculate $\beta_{eff}$ from PSD in this study, the PSD absorption efficiency $\overline{Q}_{abs}$ is given as $\overline{Q}_{abs} = \beta_{abs} / A_{PSD}$, where $\beta_{abs}$ is the PSD absorption coefficient (determined by MADA from measured PSD) and $A_{PSD}$ is the measured PSD projected area. The PSD effective diameter was determined from the measured PSD as described in Mishra et al. (2014), but in essence is

given as $D_e = (3/2)$ IWC/($\rho_i$ $A_{PSD}$), where $\rho_i$ is the density of bulk ice (0.917 g cm$^{-3}$). The PSD extinction efficiency $\overline{Q}_{ext}$ was determined in a manner analogous to $\overline{Q}_{abs}$. The single scattering albedo $\omega_o$ was calculated as $\omega_o = 1 - \overline{Q}_{abs}/\overline{Q}_{ext}$ and the PSD asymmetry parameter g was obtained from $D_e$ using the parameterization of Yang et al. (2005). Knowing $\overline{Q}_{abs}$, $\omega_o$ and g, $\beta_{eff}$ was calculated from the PSD as:

$$\beta_{eff} = Q_{abs,eff}(12.05\mu m)/Q_{abs,eff}(10.6\ \mu m)\ , \tag{4}$$

$$Q_{abs,eff} = \overline{Q}_{abs}\ (1 - \omega_o\ g)\ /\ (1 - \omega_o). \tag{5}$$

Note that $\beta$ (i.e. $\beta_{eff}$ without scattering effects) is also equal to $\overline{Q}_{abs}(12.05\ \mu m)/\overline{Q}_{abs}(10.6\ \mu m)$.

Figure 2 shows measurements of N/IWC from SPARTICUS flights over the central United States, based on PSD sampled from synoptic (blue squares) and anvil (black squares) cirrus clouds. Cirrus cloud PSD were measured using the 2D-S probe, which produces shadowgraph images with true 10 μm pixel resolution at aircraft speeds up to

170 m s$^{-1}$, measuring ice particles between 10 and 1280 (or more) μm (Lawson et al., 2006). The 2D-S PSD data

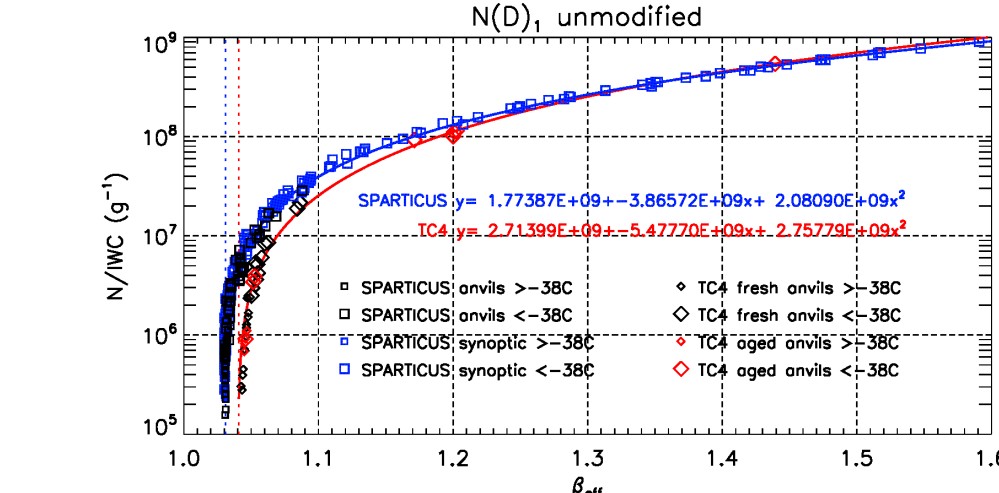

**Figure 2:** **Dependence of N/IWC (g$^{-1}$) on the effective absorption optical depth ratio $\beta_{eff}$ as predicted from the method of Parol et al. (1991), based on PSDs from SPARTICUS synoptic cirrus (blue squares) and anvils (black squares), and TC4 aged (red diamonds) and fresh (black diamonds) anvils, where the first size-bin is included. The larger (smaller) symbols denote PSDs measured at a temperature colder (warmer) than -38°C. The curve-fit equations are for SPARTICUS synoptic cirrus (blue) and for TC4 aged and fresh anvils (red).**



was post-processed using an ice particle arrival time algorithm that identifies and removes ice shattering events from the data stream. All size-bins of the 2D-S probe were used here. Two SPARTICUS flights from April 28[th] were added to this dataset (giving a total of 15 flights) since April 28[th] was previously mislabeled as an anvil cirrus case

study, but was actually a ridge-crest cirrus event (a type of synoptic cirrus) as described in Muhlbauer et al. (2014). This "ridge crest cirrus" had high N (500-2200 L$^{-1}$) for T < -60°C. Also shown are N/IWC measurements from the TC4 field campaign for maritime "fresh" (black diamonds) anvil cirrus (during active deep convection where the anvil is linked to the convective column) and for aged (red diamonds) anvil cirrus (anvils detached from convective column). Figure 2 relates $\beta_{eff}$ to the N/IWC ratio, where $\beta_{eff}$ was calculated from the same PSD measurements used

to calculate N and IWC, based on the MADA method. The PSD measurements include size-resolved estimates of ice particle mass concentration based on Baker and Lawson (2006a), size-resolved measurements of ice projected area concentration, and the size resolved number concentration. This PSD information is the input for the MADA method that yields the coefficients of absorption and extinction. The tunneling efficiency $T_e$ used in MADA was estimated from Table 1 in Mitchell et al. (2006), where for 1 μm < D < 30 μm, droxtals and hexagonal columns are

assumed and $T_e = 0.90$; for 30 μm < D < 100 μm, budding bullet rosettes and hexagonal columns are assumed and $T_e = 0.50$; for D > 100 μm bullet rosettes and aggregates are assumed and $T_e = 0.15$. This shape-dependence on ice particle size was guided by the ice particle size-shape observations reported in Lawson et al. (2006), Baker and Lawson (2006b) and Erfani and Mitchell (2016), where small hexagonal columns (for which we can estimate $T_e$) are substituted for small irregular crystals. These ice particle shape assumptions affect only $T_e$, and the cloud optical

properties are primarily determined through the PSD measurements noted above (i.e. not the value of $T_e$). Due to $\beta_{eff}$'s sensitivity to tunneling and small ice crystals, a tight and useful relationship is found between N/IWC and $\beta_{eff}$ for N/IWC > ~ $10^7 g^{-1}$ for both campaigns. As far as we know, this relationship was not known previously. The associated PSDs were measured at temperatures colder than -38 °C (large symbols), which is the cloud base temperature of the cirrus clouds targeted for this study. For N/IWC < ~ $10^6 g^{-1}$, $\beta_{eff}$ reaches a low limit and is not

sensitive to N/IWC, so that N/IWC cannot be estimated from $\beta_{eff}$. Much of the data with $\beta_{eff}$ reaching the "no sensitivity" low limit were derived from ice cloud PSDs at temperatures between -20 and -38 °C (small symbols) where PSD tend to be relatively broad.

Because the clouds selected in this study are single-layered semi-transparent cirrus clouds, the relationships seen for synoptic cirrus during SPARTICUS could be deemed the most relevant for this study. But interestingly, synoptic

and anvil cirrus during SPARTICUS (squares) follow similar relationships, and similarly, aged and fresh cirrus anvils (diamonds) follow similar relationships during TC4. Thus, the fact that anvil and synoptic cirrus during SPARTICUS follow a similar relationship suggests that the relationship established from anvils during TC4 could also be relevant for this study. The blue line shows the curve fit derived from SPARTICUS synoptic cirrus, while





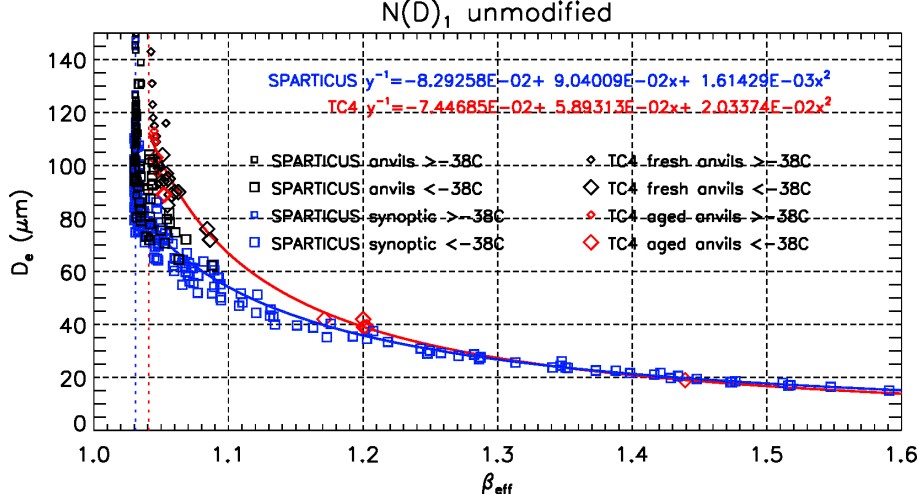

**Figure 3: Dependence of the PSD effective diameter $D_e$ (μm) on the effective absorption optical depth ratio $\beta_{eff}$ as predicted from the method of Parol et al. (1991), based on PSDs from SPARTICUS synoptic cirrus (blue squares) and anvils (black squares), and TC4 aged (red diamonds) and fresh (black diamonds) anvils, where the first size-bin is included ($N(D)_1$ unmodified). The larger (smaller) symbols denote PSDs measured at a temperature colder (warmer) than -38°C. The curve-fit equations give $1/D_e$ in μm$^{-1}$; they are for SPARTICUS synoptic cirrus (blue) and for TC4 aged and fresh anvils (red).**

the red line is derived from aged and fresh anvils during TC4. The blue and the red lines are similar for the largest $\beta_{eff}$ ($\beta_{eff} > 1.2$) and progressively depart as $\beta_{eff}$ decreases. The $\beta_{eff}$ low limit is 1.031 during SPARTICUS, for both synoptic and anvil cirrus, whereas it is 1.041 during TC4. The different $\beta_{eff}$ low limits during SPARTICUS and TC4 may reflect different PSD shapes measured during these two campaigns. This indicates that as $\beta_{eff}$ decreases and its sensitivity to N/IWC decreases, the sensitivity of the relationships to the PSD increases. These two curves are indicative of the possible dispersion in the relationships, and therefore will both be considered in the analysis.

Using this same in situ data and methodology, $\beta_{eff}$ has also been related to $D_e$ as shown in Fig. 3, where all PSD bins were used. $D_e$ is defined as $(3/2) IWC/(\rho_i A_{PSD})$ where $\rho_i$ is the density of bulk ice (Mitchell, 2002). Accordingly, $D_e$ was calculated from the measured PSD (see Mishra et al., 2014). The relationships derived from SPARTICUS synoptic cirrus and from TC4 anvils are shown in blue and red, respectively. They are only useful for $D_e < 90$ or $D_e < 110$ μm, respectively, since $\beta_{eff}$ is only sensitive to the smaller ice particles. The relationships are similar for $D_e$ below 30 μm. This emphasizes the fact that $\beta_{eff}$ is a measure of the relative concentration of small ice crystals in a PSD (Mitchell et al., 2010). $A_{PSD}$ and $\beta_{eff}$ (PSD integrated quantities) may be associated with a substantial portion of larger ice particles (D > 50 μm) before $\beta_{eff}$ loses sensitivity to changes in $D_e$.

### 2.4 Impact of the smallest size bin in PSD measurements

The relationships shown in Figs. 2 and 3 were derived by using all the size bins of the 2D-S probe. However, the data in the smallest size bin (5–15 μm) has the greatest uncertainty since the sample volume of the 2D-S probe



depends on particle size, and this volume is smallest (with greatest measurement uncertainty) for the smallest size-bin (Paul Lawson, personal communication). This motivated us to formulate the retrieval relationships in two ways: (1) assuming that the PSD first size-bin, $N(D)_1$, is valid and unmodified (as considered earlier), and (2) assuming that $N(D)_1$ equals zero. Jensen et al. (2013a) argues that $N(D)_1$ as measured by the 2D-S probe is anomalously high

since it tends to be considerably higher than the adjacent size-bins [$N(D)_2$ for example] and that these small ice crystals should rapidly grow or sublimate to larger or smaller sizes (> 15 µm or < 5 µm) due to the relative humidity with respect to ice, $RH_i$, being significantly different than ice saturation ($RH_i = 100\%$). Therefore, $N(D)_1 > N(D)_2$ would imply frequent ice nucleation events to sustain these higher $N(D)_1$ values, which appears unlikely. This argument provided an additional incentive to formulate this retrieval using assumption (2).

However, there are also physical reasons that argue in favor of the first assumption [$N(D)_1$ is a valid measurement]. For example, if strong competition for water vapor due to a relatively high small ice crystal concentration (e.g. due to a homogeneous ice nucleation event) rapidly reduces the $RH_i$ to ~ 100%, then this relatively high concentration may last for time periods comparable to the lifetime of the cirrus cloud. High ice crystal concentrations (~ 300 to 10,000 $L^{-1}$) associated with $RH_i$ ~ 100% were documented by aircraft measurements in the tropical tropopause layer

(TTL), existing in layers ranging from meters to 0.4 km in depth (Jensen et al., 2013b). These layers were embedded within a deeper cirrus cloud having N typically less than 20 $L^{-1}$ (where $RH_i$ was higher). Evidence that $RH_i$ near 100% is common in cirrus clouds is shown in Figs. 6 and 7 of Krämer et al. (2009), where for the relationships most representative of cirrus clouds, the relaxation time $\tau$ for $RH_i$ to develop a quasi-steady state (i.e. dynamical equilibrium denoted $RH_{qsi}$) is on the order of 5-10 minutes. $RH_{qsi}$ is where $d(RH_i)/dt \approx 0$, where the rate

of vapor uptake by ice approximately balances the rate of supersaturation development. Between the time of initial in-cloud supersaturation (corresponding to cloud formation) and $\tau$, the cloud updraft w tends to be higher than the w occurring after $RH_{qsi}$ is attained. Since cirrus cloud lifetimes tend to be considerably longer than 5-10 minutes, w is relatively low with $RH_i$ ~ 100% for time $t > \tau$. This may explain the relatively high frequencies of occurrence of $RH_i$ near 100% in Fig. 7 of Krämer et al. (2009). It can also be argued that cirrus clouds are formed by atmospheric

wave activity, and that $RH_i$ near 100% results from averaging transient wave-induced fluctuations of $RH_i$. However, Fig. 1 in Krämer et al. (2009) shows water vapor concentrations being fairly constant with time over periods of 30 to 50 minutes, while also showing evidence of wave-induced fluctuations in $RH_i$ during another period.

Figure 4 shows the N/IWC- $\beta_{eff.}$ (top) and $D_e$-$\beta_{eff}$ (bottom) relationships derived from SPARTICUS and TC4 assuming $N(D)_1=0$. Assuming $N(D)_1 =0$ not only reduces N, but it also reduces $\beta_{eff.}$, especially when $\beta_{eff}$ is large.

For instance, assuming $N(D)_1=0$ reduces the maximum value of in situ $\beta_{eff.}$ from 1.6 to 1.24 during SPARTICUS, and from 1.44 to 1.27 during TC4. As a result, the N/IWC- $\beta_{eff}$ relationships are fairly close for both assumptions. For example, assuming $N(D)_1=0$ reduces N/IWC by about 20% for the largest $\beta_{eff}$, thereby confirming the tight relationship between N/IWC and $\beta_{eff}$ highlighted earlier. Assuming $N(D)_1=0$ has a larger impact on the $D_e$- $\beta_{eff}$



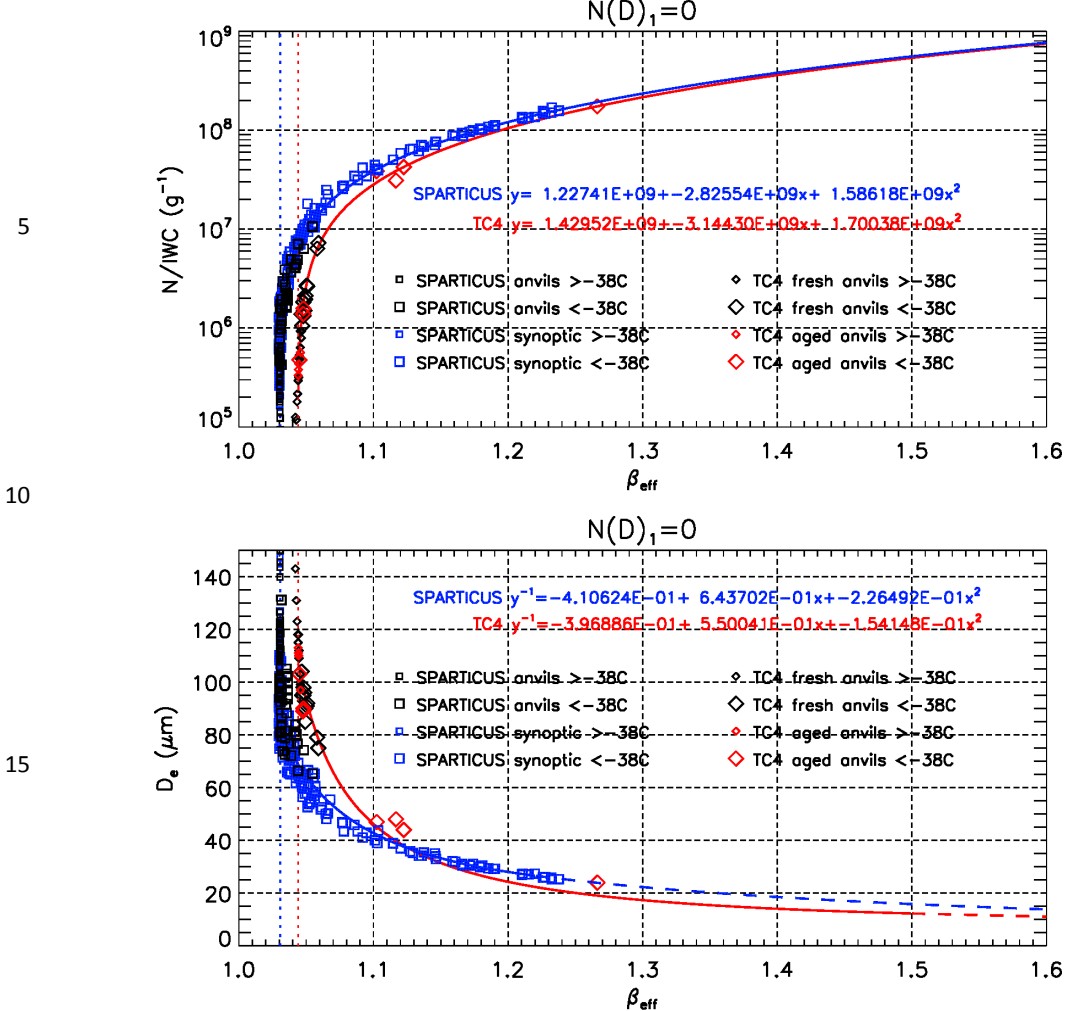

**Figure 4: Same as Fig. 2 (top) and Fig. 3 (bottom), but where the first size-bin of the PSDs is not included (N(D)$_1$=0). The dashed lines in the lower panels are where the curve-fit equations giving 1/D$_e$ in µm$^{-1}$ are extrapolated (see Table 1).**

relationship. For SPARTICUS, the largest relative change in D$_e$ is around $\beta_{eff}$ =1.15, where D$_e$ is reduced by 26 % assuming N(D)$_1$=0 instead of N(D)$_1$ unmodified. For TC4, the largest relative reduction in D$_e$ is 38 % at $\beta_{eff}$ =1.2.

The unmodified N(D)$_1$ assumption can be viewed as an upper limit while the N(D)$_1$ = 0 assumption is clearly a lower limit for the actual value of N(D)$_1$, and in this way our relationships are bracketed by these two limiting conditions. The comparison of in situ $\beta_{eff}$ for each of these assumptions with $\beta_{eff}$ derived independently from CALIPSO IIR is an additional piece of information, as discussed in Sect. 4.



Uncertainties regarding the photon tunneling efficiency ($T_e$) values assumed in Sect. 2.3 were evaluated, but these were much smaller than the uncertainties described above. For example, assuming bullet rosettes for all sizes results in $T_e$ values of 0.70, 0.40 and 0.15 for the three size categories considered (from smallest to largest). Among plausible crystal shape assumptions, this assumption yielded the lowest $T_e$ values, reducing $\beta_{eff}$ by no more than

2.6%. Note that for D < 80 μm, > 85% of the ice particles tend to be irregular (e.g. blocky or quasi-spherical) or spheroidal in shape (Lawson et al., 2006b; Baker and Lawson, 2006b), and these shapes should correspond to relatively high $T_e$ values (Mitchell et al., 2006).

### 2.5   Retrieving N from CALIPSO $\beta_{eff}$

Using cloud layer average $\beta_{eff}$ derived from IIR and CALIOP observations (Sect. 2.2), the N/IWC- $\beta_{eff}$- and $D_e$-$\beta_{eff}$

relationships established from in situ measurements are used to derive the cloud layer N from CALIPSO, as presented in this sub-section. The sensitivity ranges (N/IWC > ~ $10^7$ $g^{-1}$ & $D_e$ < 90-110 μm) are usually compatible with cirrus clouds (T < -38 °C) since PSDs tend to be narrower at these temperatures, containing relatively small ice particles (e.g. Mishra et al., 2014). We choose to use the relationships established during SPARTICUS and TC4, assuming measured unmodified $N(D)_1$ (Figs. 2 and 3) and $N(D)_1=0$ (Fig. 4), as discussed above. Using these four

relationships provides a means of estimating the uncertainty in N resulting from regional differences in cirrus microphysics.

Using CALIPSO $\beta_{eff}$ and the N/IWC-$\beta_{eff}$ and $D_e$-$\beta_{eff}$ relationships, the cloud layer N is retrieved as: N = IWC x (N/IWC) with IWC computed as:

$$IWC = \frac{\rho_i}{3} \times \alpha_{ext} \times D_e \qquad (6)$$

where $\rho_i$ is the bulk density of ice (0.917 g $cm^{-3}$), and $\alpha_{ext}$ is the effective layer-average visible extinction coefficient, which is derived from CALIOP and IIR as:

$$\alpha_{ext} = \frac{\left(2/Q_{abs,eff}(12\mu m)\right)}{\Delta z_{eq}} \times \tau_{abs}(12.05\mu m). \qquad (7)$$

The quantity 2/$Q_{abs,eff}$(12 μm), where 2 is the value of $\bar{Q}_{ext}$ for ice PSDs in the visible spectrum, converts $\tau_{abs}$(12.05 μm) to an equivalent visible extinction optical depth (OD). It is obtained from $\beta_{eff}$ as illustrated in Fig. 5 for both

$N(D)_1$ assumptions. For a given $\beta_{eff}$, 2/$Q_{abs,eff}$ (12 μm) is smaller by less than 7 % assuming $N(D)_1=0$, and consequently, IIR $\alpha_{ext}$ is also smaller by less than 7 %. The effective cloud thickness, $\Delta z_{eq}$, accounts for the fact that the IIR instrument does not sense equally all of the cloud profile that contributes to thermal emission. This is taken



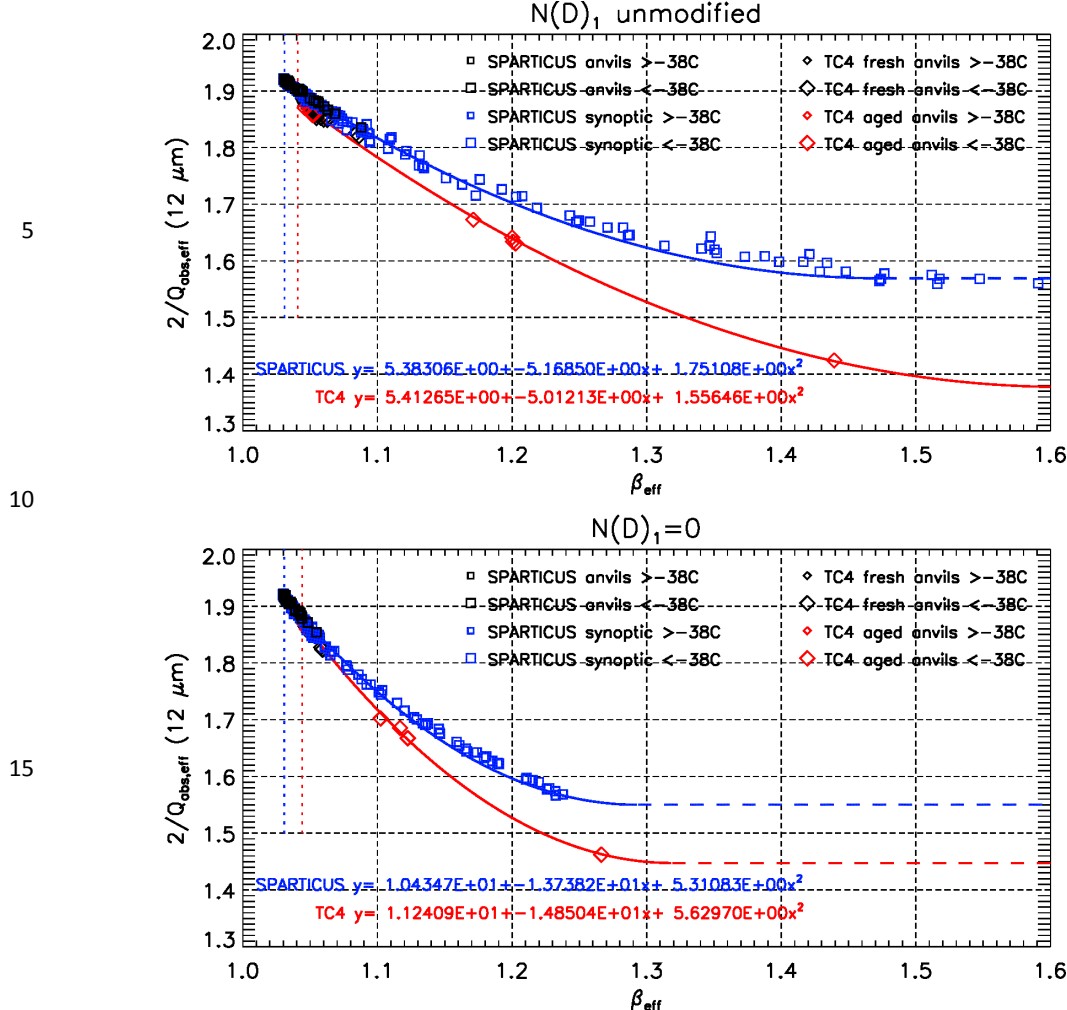

**Figure 5: The $\beta_{eff}$ dependence of the term that converts the absorption optical depth $\tau_{abs}$ into visible optical depth in Eq. 7, based on PSDs from SPARTICUS synoptic cirrus (blue squares) and anvils (black squares), and TC4 aged (red diamonds) and fresh (black diamonds) anvils, where the first size-bin is included (top, $N(D)_1$ unmodified), or not included (bottom, $N(D)_1=0$). The larger (smaller) symbols denote PSDs measured at a temperature colder (warmer) than -38°C. The curve-fit equations are for SPARTICUS synoptic cirrus (blue) and for TC4 aged and fresh anvils (red). The dashed lines are where the curve-fit equations are extrapolated (see Table 1).**

into account through the IIR weighting profile introduced in Sect. 2.2.3, which gives more weight to large emissivity and therefore to the large extinctions in the cloud profile. Using the IIR weighting profile, the layer absorption coefficient $\alpha_{abs}(12.05\ \mu m)$ for the IIR 12.05 μm channel is computed from the weighted averaged absorption coefficient profile. This yields $\alpha_{abs}(12.05\ \mu m) > \alpha_{abs,mean}(12.05\ \mu m)$, where $\alpha_{abs,mean}(12.05\ \mu m)$ is the mean absorption coefficient, that is, the ratio of $\tau_{abs}(12.05\ \mu m)$ to the CALIOP layer geometric thickness, $\Delta z$. Thus, an





equivalent effective thickness is defined as $\Delta z_{eq}$ where $\alpha_{abs}(12.05\ \mu m) = \tau_{abs}(12.05\ \mu m)/\Delta z_{eq}$, or alternatively, $\Delta z_{eq} = \Delta z\ (\alpha_{abs,mean}(12.05\ \mu m)/\alpha_{abs}(12.05\ \mu m)\ )$. In practice, $\Delta z_{eq}$ is found equal to 30% to 90 % of $\Delta z$.

To summarize, the retrieval equation is:

$$N = \frac{\rho_i}{3} \times \frac{\left[ {}^{2}\!\!\diagup\!\!{}_{Q_{abs,eff}}(12\mu m) \right] \cdot \tau_{abs}(12.05\mu m)}{\Delta z_{eq}} \times D_e \times \left( \frac{N}{IWC} \right) \tag{8}$$

The retrieval of $\tau_{abs}(12.05\ \mu m)$ and $\tau_{abs}(10.6\ \mu m)$ combined with the CALIOP extinction profile provides $\beta_{eff}$, $N/IWC$, $D_e$, $\alpha_{ext}$, and therefore layer-average IWC and finally layer N. Perhaps the most unique aspect of this retrieval method is its sensitivity to small ice crystals via $\beta_{eff}$.

### 3 Applying the retrieval method

### 3.1 Regression curves

Regression curves derived from SPARTICUS and TC4 for both assumptions [N(D)$_1$ is unmodified and N(D)$_1 = 0$] for the quantities N/IWC, $D_e$ and $2/Q_{abs,eff}(12\ \mu m)$ are given in Table 1, constituting the four formulations of this retrieval scheme. A number of adjustments of the second-order polynomials were needed to provide retrievals for any value of $\beta_{eff}$. They correspond to the dashed lines in Fig. 4 (bottom) and in Fig. 5. For the N(D)$_1$=0 assumption

and $\beta_{eff} > 1.22$, $1/D_e$ is extrapolated linearly from the tangent value at $\beta_{eff} = 1.22$. For the N(D)$_1$ unmodified assumption, when $\beta_{eff} > 1.48$, then $2/Q_{abs,eff}(12\ \mu m)$ is set to 1.57. For the N(D)$_1 = 0$ assumption, when $\beta_{eff} > 1.30$, then $2/Q_{abs,eff}(12\ \mu m)$ is set to 1.55. Furthermore, when calculating N/IWC, $D_e$, and $2/Q_{abs,eff}(12\ \mu m)$ from $\beta_{eff}$, if the retrieved $\beta_{eff}$ is less than the lower sensitivity limit, then $\beta_{eff}$ is set to this value. For instance, N/IWC corresponding to this value via the regression curves is about $2.3 \times 10^5\ g^{-1}$. As shown in Table 2 for the year 2013, this practice

affected 15% and 18% of the N/IWC and $D_e$ retrievals over ocean and land, respectively, when using the SPARTICUS relationships, for which the lower sensitivity limit is $\beta_{eff} = 1.031$. Using the TC4 relationships, 20% and 22% of the samples over ocean and land, respectively, had $\beta_{eff}$ larger than the sensitivity limit of 1.044. To better estimate the median values and percentiles for N and $D_e$, N and $D_e$ retrievals calculated using these limiting values are accounted for when determining these statistics.

The retrieved N for the four formulations of the retrieval scheme can be compared through the product of the three $\beta_{eff}$-dependent quantities, N/IWC, $D_e$ and $2/Q_{abs,eff}(12\ \mu m)$, as shown in Fig. 6. The upper and lower bounds are the





**Table 1. Regression curve variables and coefficients for second-order polynomials of the form $y = a_0 + a_1 x + a_2 x^2$, used in the CALIPSO retrieval. Units for N/IWC and $D_e$ are in $g^{-1}$ and microns, respectively.**

| $N(D)_1$ | $y$ | $x$ | $a_o$ | $a_1$ | $a_2$ |
|---|---|---|---|---|---|
| | | | SPARTICUS – Synoptic cirrus | | |
| unmodified | N/IWC | $\beta_{eff}$ | $1.77387 \times 10^9$ | $-3.86572 \times 10^9$ | $2.08090 \times 10^9$ |
| unmodified | $1/D_e$ | $\beta_{eff}$ | -0.0829258 | 0.0904009 | 0.00161429 |
| unmodified | $2/Q_{abs,eff}$ | $\beta_{eff}<1.476$ | 5.38306 | -5.16850 | 1.75108 |
| | (12 μm) | $\beta_{eff}>1.476$ | 1.56921 | 0 | 0 |
| = 0 | N/IWC | $\beta_{eff}$ | $1.22741 \times 10^9$ | $-2.82554 \times 10^9$ | $1.58618 \times 10^9$ |
| = 0 | $1/D_e$ | $\beta_{eff}<1.22$ | -0.410624 | 0.643702 | -0.226492 |
| | | $\beta_{eff}>1.22$ | -0.0735133 | 0.0910615 | 0 |
| = 0 | $2/Q_{abs,eff}$ | $\beta_{eff}<1.293$ | 10.4347 | -13.7382 | 5.31083 |
| | (12 μm) | $\beta_{eff}>1.293$ | 1.55011 | 0 | 0 |
| | | | TC4 – Aged and fresh anvils | | |
| unmodified | N/IWC | $\beta_{eff}$ | 2.71399e+09 | -5.47770e+09 | 2.75779e+09 |
| unmodified | $1/D_e$ | $\beta_{eff}$ | -0.0744685 | 0.0589313 | 0.0203374 |
| unmodified | $2/Q_{abs,eff}$ | $\beta_{eff}<1.61$ | 5.41265 | -5.01213 | 1.55646 |
| | (12 μm) | $\beta_{eff}>1.61$ | 1.37763 | 0 | 0 |
| = 0 | N/IWC | $\beta_{eff}$ | 1.42952e+09 | -3.14430e+09 | 1.70038e+09 |
| = 0 | $1/D_e$ | $\beta_{eff}<1.5$ | -0.396886 | 0.550041 | -0.154148 |
| | | $\beta_{eff}>1.5$ | -0.0500520 | 0.0875957 | 0 |
| = 0 | $2/Q_{abs,eff}$ | $\beta_{eff}<1.319$ | 11.2409 | -14.8504 | 5.62970 |
| | (12 μm) | $\beta_{eff}>1.319$ | 1.44756 | 0 | 0 |

5 $N(D)_1$ unmodified:  SPARTICUS: if $\beta_{eff} < 1.031$, then x=1.031; TC4: if $\beta_{eff} < 1.04085$, then x=1.04085
  $N(D)_1=0$:   SPARTICUS: if $\beta_{eff} < 1.03078$, then x=1.03078; TC4: if $\beta_{eff} < 1.04410$, then x=1.04410

**Table 2. Fraction of samples with IIR $\beta_{eff}$ larger than the N/IWC and $D_e$ sensitivity limit, per season in 2013, over ocean and over land.**

| Fraction Samples | DJF | | MAM | | JJA | | SON | |
|---|---|---|---|---|---|---|---|---|
| | $\beta_{eff} > 1.031$ | $\beta_{eff} > 1.044$ | $\beta_{eff} > 1.031$ | $\beta_{eff} > 1.044$ | $\beta_{eff} > 1.031$ | $\beta_{eff} > 1.044$ | $\beta_{eff} > 1.031$ | $\beta_{eff} > 1.044$ |
| Sea | 0.86 | 0.82 | 0.85 | 0.80 | 0.84 | 0.78 | 0.85 | 0.79 |
| Land | 0.83 | 0.79 | 0.81 | 0.77 | 0.79 | 0.75 | 0.84 | 0.80 |





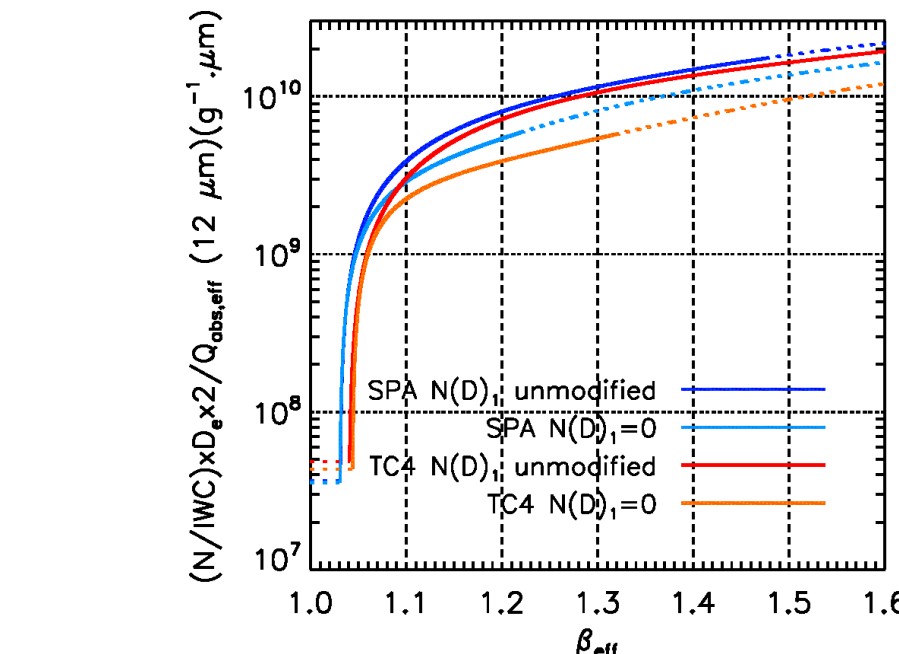

**Figure 6: Comparison of ice particle number concentration N derived from CALIPSO $\beta_{eff}$ using the four formulations of the retrieval scheme, derived from SPARTICUS using the $N(D)_1$ unmodified (navy blue) and $N(D)_1=0$ (light blue) assumptions, and from TC4 using the unmodified (red) and $N(D)_1=0$ (orange) assumptions. The dashed lines are where the curve-fit equations are extrapolated (see Sect. 3 and Table 1).**

SPARTICUS $N(D)_1$ unmodified and TC4 $N(D)_1=0$ formulations, which differ by about a factor 2 in retrieved N for $\beta_{eff} > 1.08$.

## 3.2 Relationship between $\beta_{eff}$, $\alpha_{ext}$, IWC, and N

As seen from Eq. (6), (7) and (8), $\beta_{eff}$ and $\alpha_{ext}$ are the two key parameters retrieved from the CALIPSO IIR to derive N/IWC, IWC, and finally N. The interrelationship between $\beta_{eff}$, $\alpha_{ext}$, IWC, and N is illustrated in Fig. 7 (top row) for the SPARTICUS relationships using the unmodified $N(D)_1$ assumption, which also shows the range encountered for these properties in the selected cloud population. The red dashed lines are where N = 100 L$^{-1}$, 500 L$^{-1}$ and 1000 L$^{-1}$. The pink dashed lines are where IWC = 0.5 mg g$^{-3}$, 5 mg g$^{-3}$, or 30 mg g$^{-3}$. The horizontal red dotted lines for $\beta_{eff} <$ 1.031 (or $D_e > 83$ µm) indicate where the retrieval is not sensitive to N/IWC. For $\beta_{eff} < 1.031$, N/IWC is set to its limiting (minimum) value so that N is a priori overestimated in these conditions, but typically smaller than 100 L$^{-1}$. For $\beta_{eff} < 1.031$, $D_e$ is set to 83 µm, as denoted by the horizontal pink lines, and IWC is a priori underestimated for these conditions. For our data selection, $\alpha_{ext}$ is mostly between 0.05 km$^{-1}$ and 5 km$^{-1}$. Large values of N (> 500 L$^{-1}$) result from larger values of $\beta_{eff}$ (yielding smaller $D_e$ and much larger N/IWC) and sufficiently large values of $\alpha_{ext}$ so that IWC is sufficiently large for these small values of $D_e$. Low values of N (< 100 L$^{-1}$) can be retrieved for small



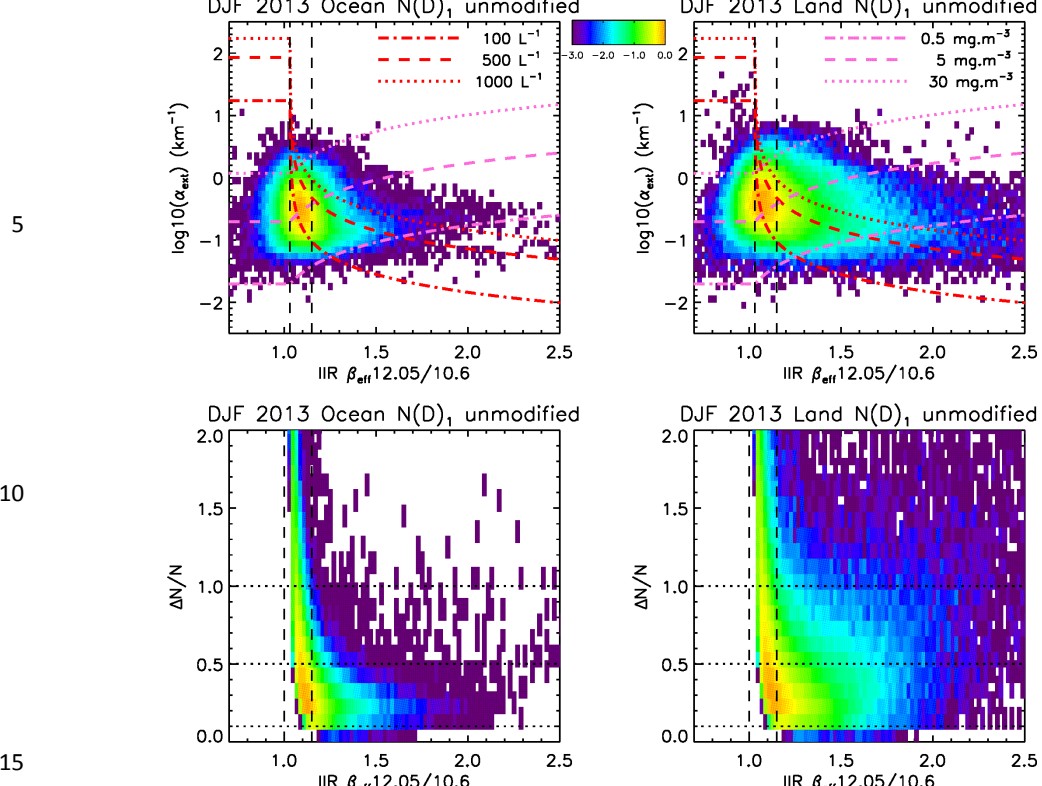

**Figure 7: Top:** The interrelationship between $\beta_{eff}$ (X-axis), layer extinction coefficient $\alpha_{ext}$ (km$^{-1}$) (Y-axis, log10 scale), ice water content IWC, and ice particle number concentration N for the SPARTICUS N(D)$_1$ unmodified assumption. The red dashed lines are where N is equal to 100, 500, or 1000 L$^{-1}$. The pink dashed lines are where IWC is equal to 0.5, 5, or 30 mg.m$^{-3}$. **Bottom:** 2D-distribution of $\beta_{eff}$ (X-axis) and relative uncertainty estimate $\Delta N/N$. The color bar gives the log of number of samples normalized to the maximum value. Relative uncertainty tends to be considerably smaller at larger $\beta_{eff}$ values. Left: ocean; right: land; all latitudes; based on December 2013, January and February 2014.

values of $\beta_{eff}$, yet larger than the low limit of 1.031, only if $\alpha_{ext}$ is sufficiently small. When $\alpha_{ext}$ is smaller than 0.05 km$^{-1}$, as could be encountered in the case of OD smaller than 0.3 (removed from the cloud sampling), finding N smaller than 100 L$^{-1}$ is very likely.

## 3.3 Retrieval uncertainties

After re-writing Eq. (8) as a function of $\beta_{eff}$ and $\tau_{abs}$(12.05 μm) using the regression curves shown in Fig. 6, the uncertainty $\Delta N$ is computed by propagating the errors in $\beta_{eff}$ (see Sect. 2.2.4) and in $\tau_{abs}$(12.05 μm), assuming a negligible error in $\Delta Z_{eq}$ (Eq. (7)) and in the relationships. More details about the uncertainty analysis and the equations used to compute $\Delta N$ can be found in the appendix.





Figure 7 (bottom row) shows $\Delta N/N$ against $\beta_{eff}$ for the same samples as in Fig. 7 (top row). $\Delta N/N$ decreases as $\beta_{eff}$ increases, reflecting that the technique is sensitive to small crystals. $\Delta N/N$ is found most of the time < 0.50 for $\beta_{eff}$ > 1.15, but increases up to more than 2.0 as $\beta_{eff}$ approaches the sensitivity limit. For a given value of $\beta_{eff}$, the variability of $\Delta N/N$ is due to the variability of $\Delta\beta_{eff}/\beta_{eff}$ and of $\Delta\alpha_{ext}/\alpha_{ext}$. $\Delta N/N$ is larger over land because of a larger

estimated uncertainty in $T_{BG}$ and also because the radiative contrast is sometimes relatively weak. $\Delta\beta_{eff}/\beta_{eff}$ is mostly due to random measurement errors, because systematic errors associated with the retrieval of $\tau_{abs}(12.05\ \mu m)$ and $\tau_{abs}(10.6\ \mu m)$ tend to cancel when these are ratioed to calculate $\beta_{eff}$. The uncertainty in $T_{BG}$ contributes more importantly to $\Delta\alpha_{ext}/\alpha_{ext}$ at the smallest emissivities. Uncertainty in $T_{BB}$ is not a major contributor for semi-transparent clouds of small to medium emissivity.

**4. Comparison with in situ cirrus cloud measurements**

Because our analysis is applied to IIR $\beta_{eff}$ using relationships with in situ $\beta_{eff}$ established from the SPARTICUS and TC4 field campaigns, a quantitative comparison of IIR and in situ $\beta_{eff}$ from these two campaigns is first discussed. Secondly, retrieved N/IWC are compared against in situ N/IWC measured from many field campaigns using the dataset of Krämer et al. (2009), where N and IWC are measured independently. The last step of this evaluation is to

compare retrieved and in situ $D_e$, IWC and N.

**4.1 Comparing IIR and in situ $\beta_{eff}$ during SPARTICUS and TC4 - Impact of the smallest size bin in PSD measurements**

For SPARTICUS, the CALIPSO retrievals were restricted to the relevant domain (latitude range 31°N-42°N and longitude range 90°W-103°W) and they were obtained from January through April of 2010 during daytime since

this period contained only synoptic cirrus based on the SPARTICUS flights. For TC4, the CALIPSO retrievals were restricted to the TC4 spatial and temporal domain (latitude range 5°S-15°N and longitude range 80°W-90°W) over oceans, in July and August 2007. CALIPSO retrievals are cloud layer average properties, and we use $T_c$ (see Sect. 2.2.3) as our best characterization of the representative cloud temperature. Finally, for each campaign, in situ $\beta_{eff}$ is computed for both $N(D)_1$ assumptions.

Comparisons with SPARTICUS are reported in Fig. 8a. The upper left panel shows the number of IIR pixels and in situ PSDs per 5°C temperature-bin. As seen in the upper right panel of Fig. 8a, the difference between the temperatures at cloud base ($T_{base}$) and cloud top ($T_{top}$) is smaller than 20°C for most of the clouds selected in the CALIPSO retrievals. Furthermore, $T_c$ tends to be located in the upper part of the cloud layer, at 20 % to 60 % from the top, and in-between, on average at 43%, where the maximum number of IIR observations are found (see dashed

black lines). This means that IIR $\beta_{eff}$ is a weighted measure near the middle of the cloud layer, slightly towards the upper part. A 5°C interval is considered in the analysis in accordance with the dispersion observed in the



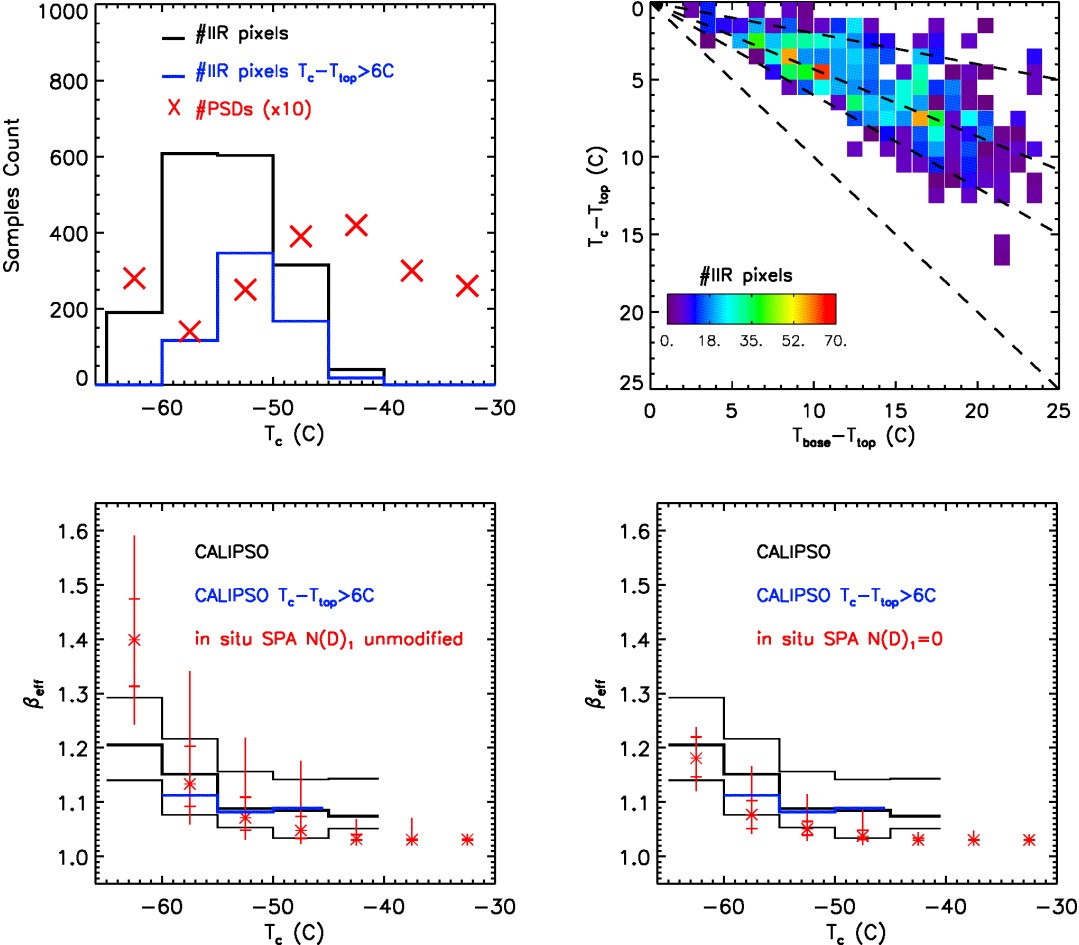

**Figure 8a: Statistical comparison of CALIPSO IIR $\beta_{eff}$ and and in situ $\beta_{eff}$ (synoptic cirrus) during SPARTICUS (sampled January-April 2010). Top left:** The red X symbols indicate the number of PSDs used in the SPARTICUS analysis (multiplied by 10 for clarity of presentation) within 5°C in situ temperature intervals, while the black histogram indicates the number of CALIPSO IIR pixels in each $T_c$ temperature interval, and the blue histogram indicates the number of IIR pixels where $T_c - T_{top}$ is larger than 6°C. **Top right:** 2D-histograms of $T_c - T_{top}$ and $T_{base} - T_{top}$ for the IIR pixels. The dashed lines from top to bottom are 20%, 43% (average value) and 60% from cloud top, with $T_c = T_{base}$ for the lowest curve. **Bottom:** Temperature dependence of in situ $\beta_{eff}$ derived assuming $N(D)_1$ unmodified (left) and $N(D)_1=0$ (right). The vertical lines give the measurement range, the horizontal bars give the 25th and 75th percentile values, and the asterisks give the median value. CALIPSO IIR median $\beta_{eff}$ is given by the thick black histogram, with thin black histograms giving the 25th and 75th percentile values. The thick blue histogram is CALIPSO IIR median $\beta_{eff}$ where $T_c - T_{top} > 6$°C.

temperature distributions (Fig. 8a top right). Because IIR pixels are required to correspond to clouds of $T_{base}$ less than -38°C, $T_c$ is mostly colder than -45°C, and the number of samples between -45°C and -40°C is relatively small (Fig. 8a, upper left panel). It is also seen that $T_c$-$T_{top}$ is smaller than 6°C for the majority of the IIR pixels overall, and for all pixels with $T_c < $ -60°C. The temperature dependence of median IIR $\beta_{eff}$ and of the 25th and 75th percentile values is given by the black histograms in the lower panels of Fig. 8a. The thick blue histogram shows median $\beta_{eff}$





when $T_c$-$T_{top}$ is larger than 6°C between -60°C and -45°C where the number of samples is sufficient (>20) for a meaningful analysis. This smaller median $\beta_{eff}$ (in blue) illustrates the sensitivity of $\beta_{eff}$ to distance from cloud top and cloud depth and indicates a smaller fraction of small ice crystals in the PSD for larger distances.

The temperature dependence of IIR $\beta_{eff}$ is compared with the dependence of in situ $\beta_{eff}$ derived by taking $N(D)_1$

unmodified (lower left panel) and $N(D)_1$=0 (lower right panel). The red asterisks denote the median in situ values, the red horizontal bars indicate the 25th and 75th percentile values and the vertical red lines indicate the range of values. Median IIR $\beta_{eff}$ and in situ $\beta_{eff}$ exhibit a similar temperature variation. A few issues should be kept in mind when interpreting these comparisons. First, regarding the unmodified version of $N(D)_1$ for T < -60 °C, the relatively high median in situ values for $\beta_{eff}$ and N (ranging from 513 to 2081 L$^{-1}$) come from two flights on 28 April 2010 that

sampled the ridge-crest cirrus mentioned above, raising some concern whether this single cirrus event was representative at these temperatures. A non-representative cirrus event could explain the larger median in situ $\beta_{eff}$ of 1.4 relative to the median IIR $\beta_{eff}$ of 1.2 for T < -60°C. Moreover, $N(D)_1$ for these April 28th PSD samples contributed 78% of the total N on average. But this would not violate our understanding of cloud physics if the RH$_i$ was near 100%. That is, high N having very small crystal sizes can exist for long periods when RH$_i$ ~ 100% since

little ice crystal growth or sublimation can occur then. Second, aircraft sampling at warmer temperatures may not be representative of satellite layer average retrieval at $T_c$ generally located in the upper part of the cloud if the cirrus layer is relatively deep with aircraft sampling relatively low in the cloud. For such conditions, the sampled ice particles would have relatively long growth times through vapor deposition and aggregation, producing relatively large ice particles and lower $\beta_{eff}$ and N in the lower cloud. This may have been the case for T > -50 °C. This point

is illustrated by the in situ measurements and modeling study of Mitchell et al. (1996) where a Lagrangian spiral descent through a cirrus layer was simulated with a steady-state snow growth model for vapor deposition and aggregation. Aggregation in the lower cloud was predicted to reduce N by ~ 60%. The larger dispersion of the IIR $\beta_{eff}$ distribution seen through the difference between the 25th and the 75th percentile values (thin black histograms) can be explained by the uncertainties reported in Table 3.

**Table 3: IIR median $\beta_{eff}$ (see Fig. 8a) and estimated uncertainty $\Delta\beta_{eff}$ during SPARTICUS. See Sect. 2.2.4 and appendix for details.**

| $T_c$ (°C) | Samples Count | Median $\beta_{eff}$ | Median $\Delta\beta_{eff}$ Measurement | Median $\Delta\beta_{eff}$ Background | Median $\Delta\beta_{eff}$ Blackbody | Median $\Delta\beta_{eff}$ All |
|---|---|---|---|---|---|---|
| -62.5 | 190 | 1.206 | 0.048 | 0.062 | 0.0018 | 0.079 |
| -57.5 | 608 | 1.151 | 0.039 | 0.037 | 0.0020 | 0.057 |
| -52.5 | 603 | 1.088 | 0.034 | 0.023 | 0.0027 | 0.044 |
| -47.5 | 315 | 1.085 | 0.040 | 0.026 | 0.0024 | 0.050 |
| -42.5 | 40 | 1.074 | 0.045 | 0.030 | 0.0029 | 0.054 |





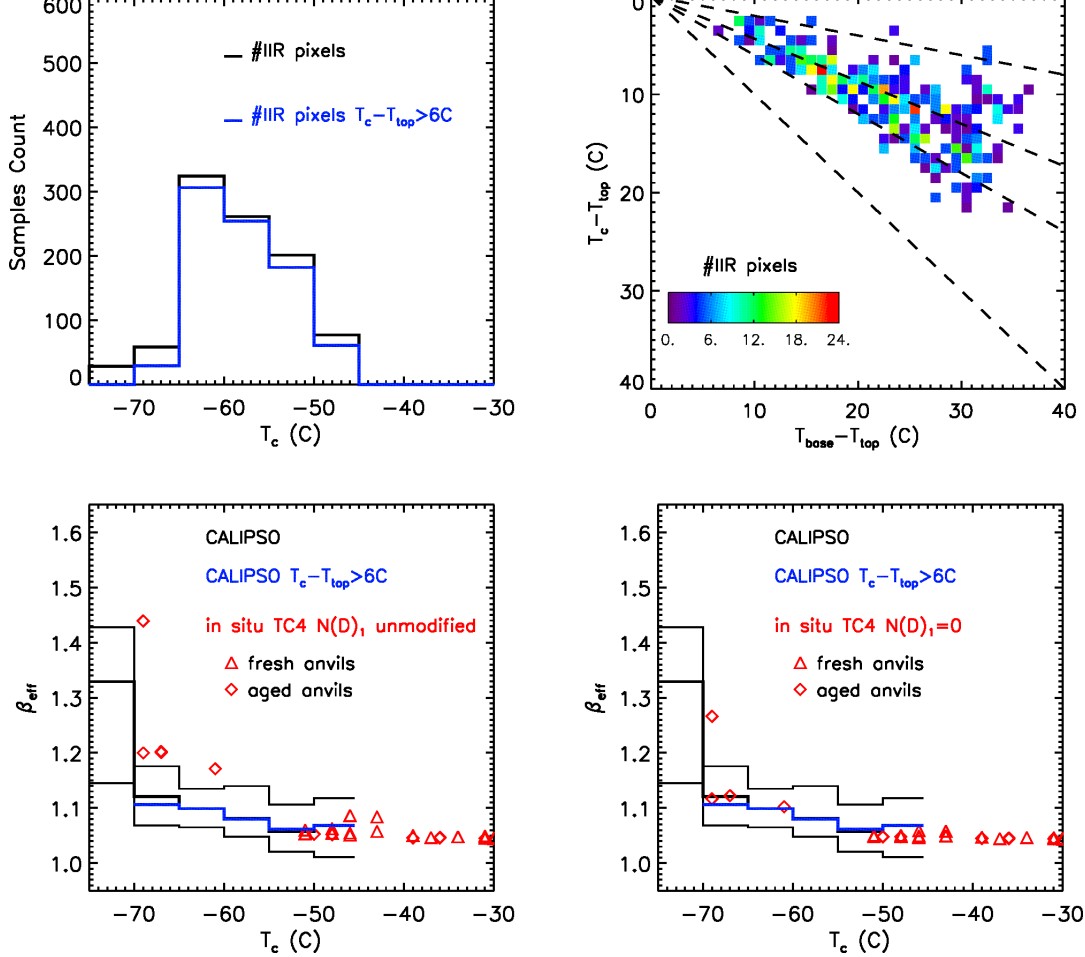

**Figure 8b: Statistical comparison of CALIPSO IIR β$_{eff}$ and and in situ β$_{eff}$ during TC4. Top left: The black histogram indicates the number of CALIPSO IIR pixels in each T$_c$ temperature interval, and the blue histogram indicates the number of IIR pixels where T$_c$ - T$_{top}$ is larger than 6°C. Top right: 2D-histograms of T$_c$ - T$_{top}$ and T$_{base}$ - T$_{top}$ for the IIR pixels. The dashed lines from top to bottom are 20%, 43% and 60% from cloud top, with T$_c$ = T$_{base}$ for the lowest curve, for comparison with SPARTICUS data. Bottom: Temperature dependence of in situ β$_{eff}$ derived assuming N(D)$_1$ unmodified (left) and N(D)$_1$=0 (right) for fresh (red triangles) and aged (red diamonds) anvils. CALIPSO IIR median β$_{eff}$ is given by the thick black histogram, with thin black histograms giving the 25$^{th}$ and 75$^{th}$ percentile values. The thick blue histogram is CALIPSO IIR median β$_{eff}$ where T$_c$ - T$_{top}$ > 6°C.**

Following the same approach as for SPARTICUS, Figs. 8b compares CALIPSO retrievals and TC4 in situ data. The IIR representative temperature is again close to mid-cloud. For temperatures between -69°C and -45°C, where CALIPSO and in situ β$_{eff}$ can be compared, most of the CALIPSO selected cirrus clouds have T$_c$-T$_{top}$ larger than 6°C (Fig. 8b, upper left panel) in contrast to SPARTICUS cirrus. The in situ and IIR β$_{eff}$ are in better agreement





when in situ $\beta_{eff}$ is computed without the first size bin ($N(D)_1$=0), especially at the coldest temperatures. The largest in situ $\beta_{eff}$ at -69°C is in fair agreement with IIR $\beta_{eff}$ in the neighboring temperature range between -75°C and -70°C.

To conclude, we find that despite the a priori different range of cloud geometrical depths (because TC4 data are from aged and fresh anvils), CALIPSO and in situ $\beta_{eff}$ are in better agreement for TC4 when the latter are computed
using $N(D)_1 = 0$, most of the time within 0.01-0.02.

### 4.2 Comparison of N/IWC with the Krämer cirrus dataset

Krämer et al. (2009) compiled coincident in situ measurements of N and IWC from 5 field campaigns (10 flights) between 68 °N and 21 °S latitude where N was measured by the FSSP probe and IWC was directly measured by various probes as described in Schiller et al. (2008). They report mass-weighted ice particle size $R_{ice}$ derived from
in situ measurements of IWC/N assuming ice spheres at bulk density (0.92 g cm$^{-3}$). Since $R_{ice}$ can be inverted to yield in situ measurements of N/IWC, this offers the opportunity to evaluate the representativeness of the four N/IWC-$\beta_{eff}$ relationships derived from the SPARTICUS and TC4 campaigns. Krämer et al. (2009) estimated that the FSSP measurements accounted for at least 80% (but typically > 90%) of the total N in a PSD. These measurements were made at T < 240 K where PSD tend to be relatively narrow and ice particle shattering upstream
of particle detection (i.e. the sample volume) is less of a problem (de Reus et al., 2009; Lawson et al., 2008). Moreover, the FSSPs used did not use a flow-straightening shroud in front of the inlet; a practice that will reduce the amount of shattering. The complete data set of in situ IWCs reported in Krämer et al. (2009) extends beyond the 10 field campaigns mentioned above, and this complete IWC data set is also described in Schiller et al. (2008).

Since this retrieval is sensitive to the smallest ice crystal sizes, it has the advantage of being sensitive to ice
nucleation processes, but this also poses certain challenges. For example, the comparison of retrieved and measured N/IWC and N in cirrus clouds is necessarily ambiguous due to (1) the uncertainty in PSD probe measurements at the smallest sizes in a PSD [assuming the probe is capable of measuring N between roughly 5 µm and 50 µm], (2) the PSD size range used to create the retrieval relationships relative to the PSD size range of the measurements used to test the retrieval, (3) the size range of the retrieved PSD (which is unknown), (4) in situ measurements in optically
thin layers below the retrieval limit of the IIR for this study, and (5) the comparison of retrieved layer-averaged quantities to localized aircraft measurements, as discussed earlier. Regarding (2), since this retrieval was developed from 2D-S probe in situ measurements, ideally it should be validated against 2D-S probe in situ measurements. Comparing with the Krämer et al. (2009) measurements introduces some ambiguity since the smallest size-bin of the 2D-S is from 5-15 µm whereas the Krämer et al. (2009) N measurements are based on the FSSP 100/300 that
sampled particles in the size range 3.0–30/0.6–40 µm diameter, and ice crystals larger than this size range were not recorded. Moreover, the amount of additional uncertainty in the FSSP measurements due to the possibility of





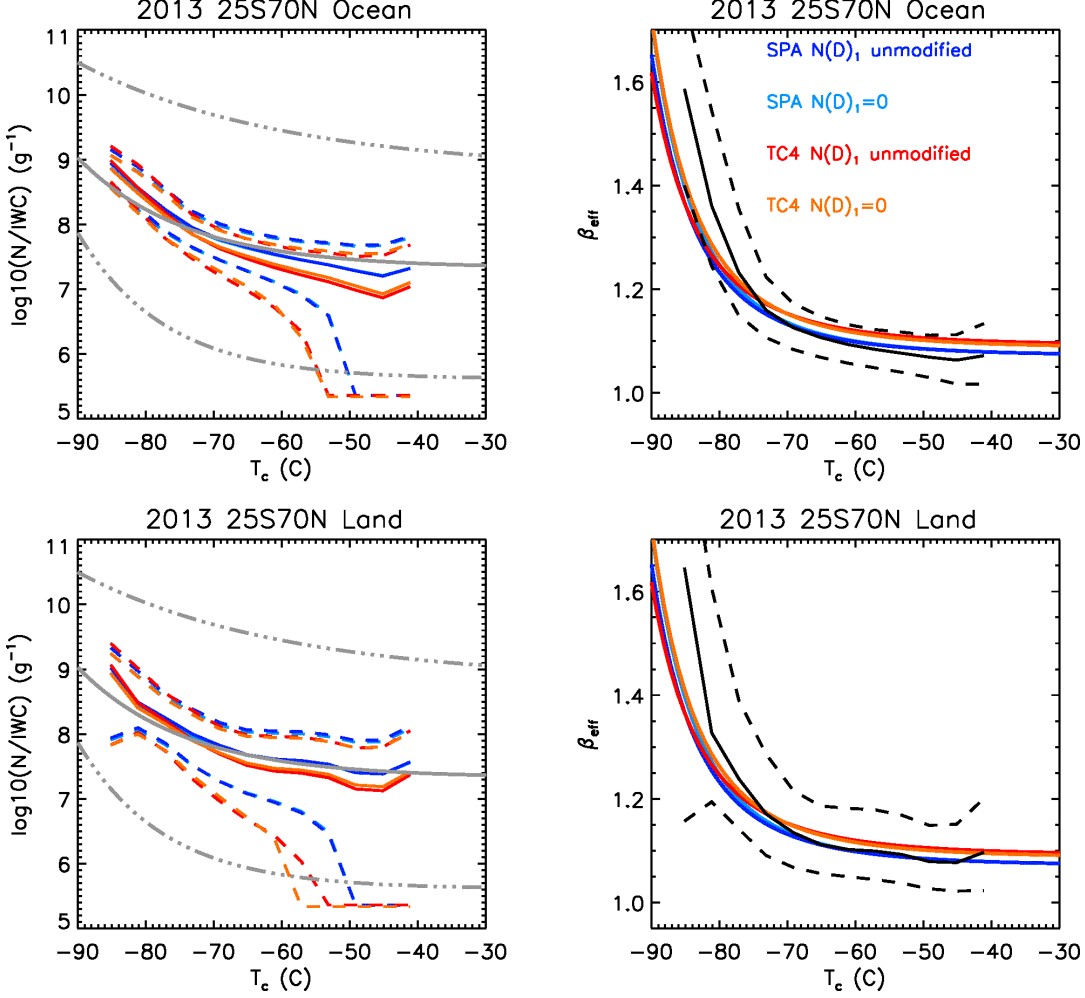

**Fig. 9.** **Left:** Comparisons of the median CALIPSO IIR N/IWC (g⁻¹) for the four formulations with in situ measurements from Krämer et al. (2009) shown by the grey curves; top and bottom being minimum and maximum values and middle grey solid curve being the middle value. Solid curves are median values while dashed curves indicate the $25^{th}$ and $75^{th}$ percentile values. **Right:** Comparisons of CALIPSO IIR $\beta_{eff}$ shown by the black curves (solid curve are median values while dashed curves indicate the $25^{th}$ and $75^{th}$ percentile values) with the four in situ $\beta_{eff}$ inferred from in situ N/IWC (from Krämer et al. (2009) using the four formulations. The navy and light blue curves correspond to the SPARTICUS formulations for the unmodified $N(D)_1$ assumption and the $N(D)_1 = 0$ assumption, respectively. The red and orange curves are using the TC4 formulations for the $N(D)_1$ unmodified and $N(D)_1 = 0$ assumptions, respectively. The CALIPSO IIR retrievals are for the approximate latitude range (25 °S to 70 °N) of the in situ data, over oceans (top) and over land (bottom).





shattering was not quantified. In order to perform first comparisons, we thus used these in situ measurements of IWC/N and inferred $\beta_{eff}$ using the four N/IWC vs. $\beta_{eff}$ relationships previously obtained for SPARTICUS and TC4.

Although the cirrus cloud measurements in Krämer et al. (2009) occurred over both land and ocean, no distinction was made in this regard. But since CALIPSO IIR $\beta_{eff}$ uncertainties are greater over land, Fig. 9 separates in situ and

satellite retrievals of N/IWC and $\beta_{eff}$ over ocean (top) and land (bottom). CALIPSO values are averaged over all seasons for 2013 and over the latitude range roughly corresponding to the field measurements (70°N to 25°S). Temperature intervals are 4°C.

Shown in the left panels is N/IWC vs. $T_c$. The N/IWC curve fits describing the in situ measurements of Krämer et al. (2009) are shown by the grey curves, and correspond to the maximum, minimum and middle (i.e. mid-point)

value of a cloud property as a function of temperature. They are compared with corresponding retrieved median values, based on our four formulations: SPARTICUS unmodified $N(D)_1$ (solid navy blue), SPARTICUS $N(D)_1 = 0$ (solid light blue), TC4 unmodified $N(D)_1$ (solid red), and TC4 $N(D)_1 = 0$ (solid orange), all derived from IIR $\beta_{eff}$ shown in the right panels (black curves). The dashed curves give the 25[th] and 75[th] percentile retrieval values. Using our four formulations, in situ N/IWC is converted into four in situ $\beta_{eff}$ plotted in the right panels for comparison with

IIR $\beta_{eff}$ in black. Comparing both N/IWC and $\beta_{eff}$ allows visualizing the non-linear relationship between N/IWC and $\beta_{eff}$.

Note that the Krämer et al. (2009) data used in Fig. 9 contain several non-zero bins between 5 and 15 microns (i.e. the 1[st] size-bin of the 2DS probe). Thus, the in situ PSD do not conform with the $N(D)_1 = 0$ assumption. However, as discussed above, the retrieved N/IWC (left panels of Fig. 9) is weakly sensitive to the $N(D)_1$ assumption. Given

the above ambiguities and uncertainties, the agreement between the median retrieved and in situ N/IWC is noticeable, especially for both SPARTICUS relationships over land. Both CALIPSO IIR and in situ median $\beta_{eff}$ are smaller than about 1.25 for temperatures greater than 203 K (-70 °C), in agreement with CALIPSO IIR $\beta_{eff}$ retrieved during SPARTICUS (Fig. 8a) and during TC4 (Fig. 8b).

### 4.3  Comparisons of $D_e$, IWC and N with the SPARTICUS, TC4 and Krämer datasets

The in situ and retrieved SPARTICUS cirrus cloud properties, namely $D_e$, $\alpha_{ext}$, IWC and N, are shown in Figs. 10a and 10b. They are both based on all size-bins of the 2D-S probe [$N(D)_1$ unmodified] in Fig. 10a, while these same properties were calculated using the assumption that $N(D)_1 = 0$ in Fig. 10b. The data are presented using the same convention as for $\beta_{eff}$ in Fig. 8a. For in situ data, the $N(D)_1$ assumption changes mostly N and the smallest $D_e$, but has a weaker impact on $\alpha_{ext}$ and IWC. For IIR, the changes result from the changes in the relationships with $\beta_{eff}$

(Figs. 2-6). Using $N(D)_1 = 0$ instead of $N(D)_1$ unmodified increases notably the smallest in situ $D_e$ and always



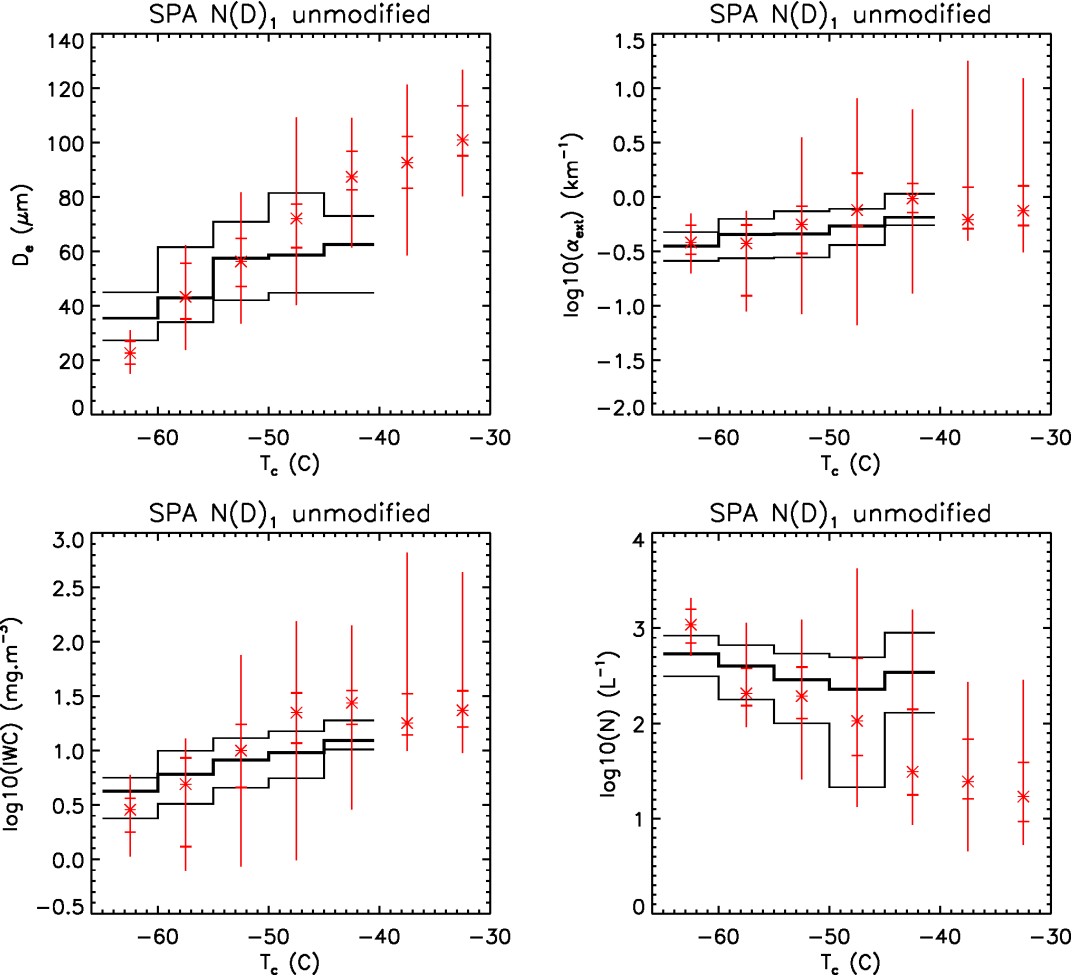

**Figure 10a. SPARTICUS in situ measurements for synoptic cirrus (sampled January-April 2010) are in red, showing the temperature dependence of effective diameter $D_e$ (µm), extinction coefficient $\alpha_{ext}$ (km$^{-1}$), ice water content IWC (mg. m$^{-3}$) and ice particle number concentration N(L$^{-1}$), and correspond to PSD measured within 5°C temperature intervals based on unmodified N(D)$_1$. The vertical lines give the measurement range, the horizontal bars give the 25$^{th}$ and 75$^{th}$ percentile values, and the asterisks give the median value. Corresponding CALIPSO IIR retrieved properties using the SPARTICUS unmodified relationships are given by the thick black histograms, with thin black histograms giving the 25$^{th}$ and 75$^{th}$ percentile values.**

decreases IIR $D_e$. The differences between in situ and IIR $D_e$ increase with temperature as $\beta_{eff}$ decreases and begins losing sensitivity to $D_e$ at warmer temperatures. Median in situ and IIR layer average $\alpha_{ext}$, which are both weakly sensitive to the N(D)$_1$ assumption, are within a factor less than 2. The notable larger variability of the in situ data is explained by the cloud local sampling in contrast to IIR average values. Reflecting the changes in $D_e$, IIR IWC is smaller using N(D)$_1$ = 0. Finally, using N(D)$_1$ = 0 reduces in situ N by a factor of 3 on average, but the change in the various relationships with $\beta_{eff}$ is such (Fig. 6) that N derived from IIR $\beta_{eff}$ is reduced by less than 35 %. The



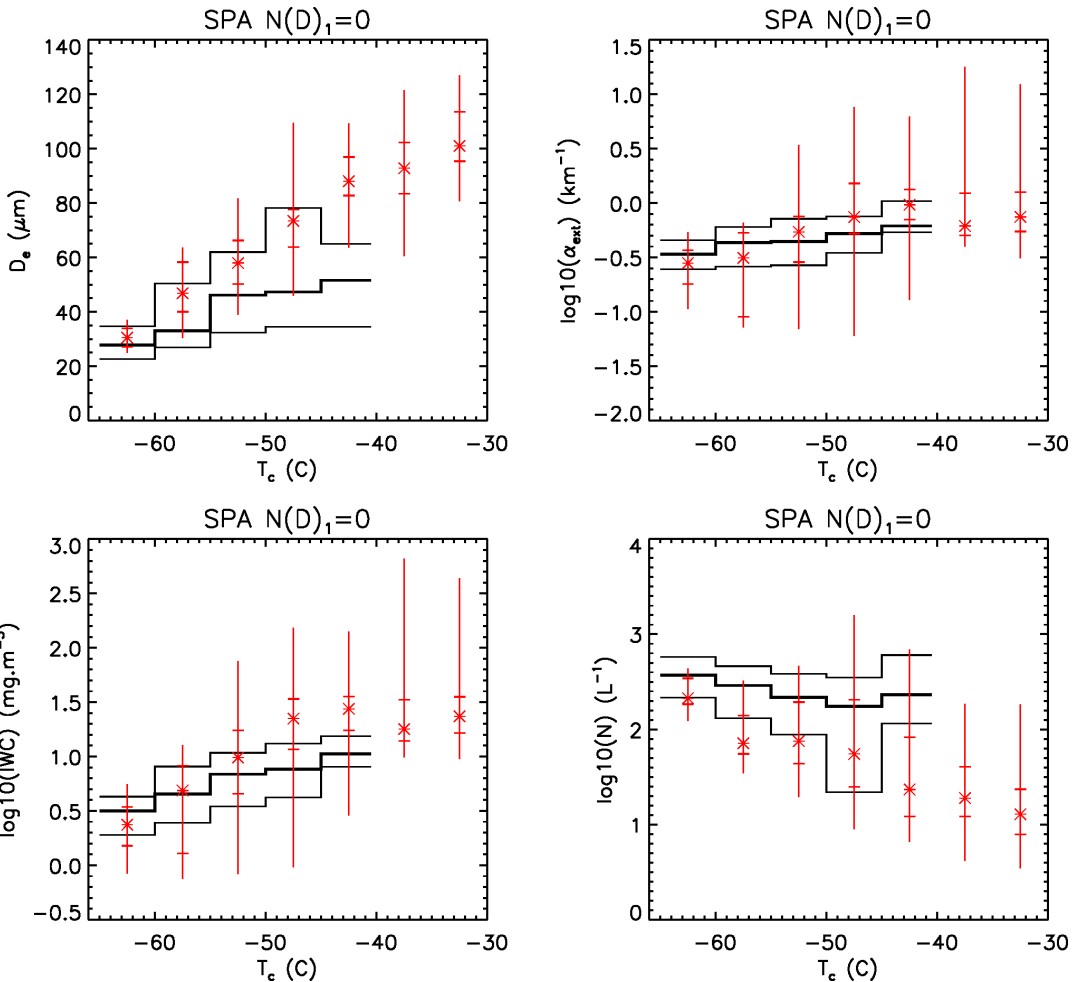

**Figure 10b. Same as Fig. 10a, except for the N(D)$_1$ = 0 assumption.**

comparison between in situ and layer average IIR N, which is driven by the comparison between the respective $\beta_{eff}$, is more favorable overall assuming N(D)$_1$ unmodified. The larger median IIR N values at $T_c$ > -45°C as compared

5    to the in situ median values is partly due to the fact that IIR $\beta_{eff}$ is larger than the in situ SPARTICUS low limit of 1.03 and possibly to a larger uncertainty $\Delta N$ as $\beta_{eff}$ approaches the sensitivity limit. As discussed earlier, IIR retrievals are layer average quantities with sensitivity to the upper part of the cloud layers, whereas in situ measurements can be in the lower part of a relatively deep cloud.

For TC4, CALIPSO and in situ D$_e$ (Figs, 10c and 10d, upper left panels) are in good agreement using the N(D)$_1$ = 0

10    assumption (Fig. 10d). The differences in cloud ODs (relative to SPARTICUS) is made evident when comparing the extinction coefficients. CALIPSO and in situ $\alpha_{ext,}$ are of the same order of magnitude for aged anvils (except at





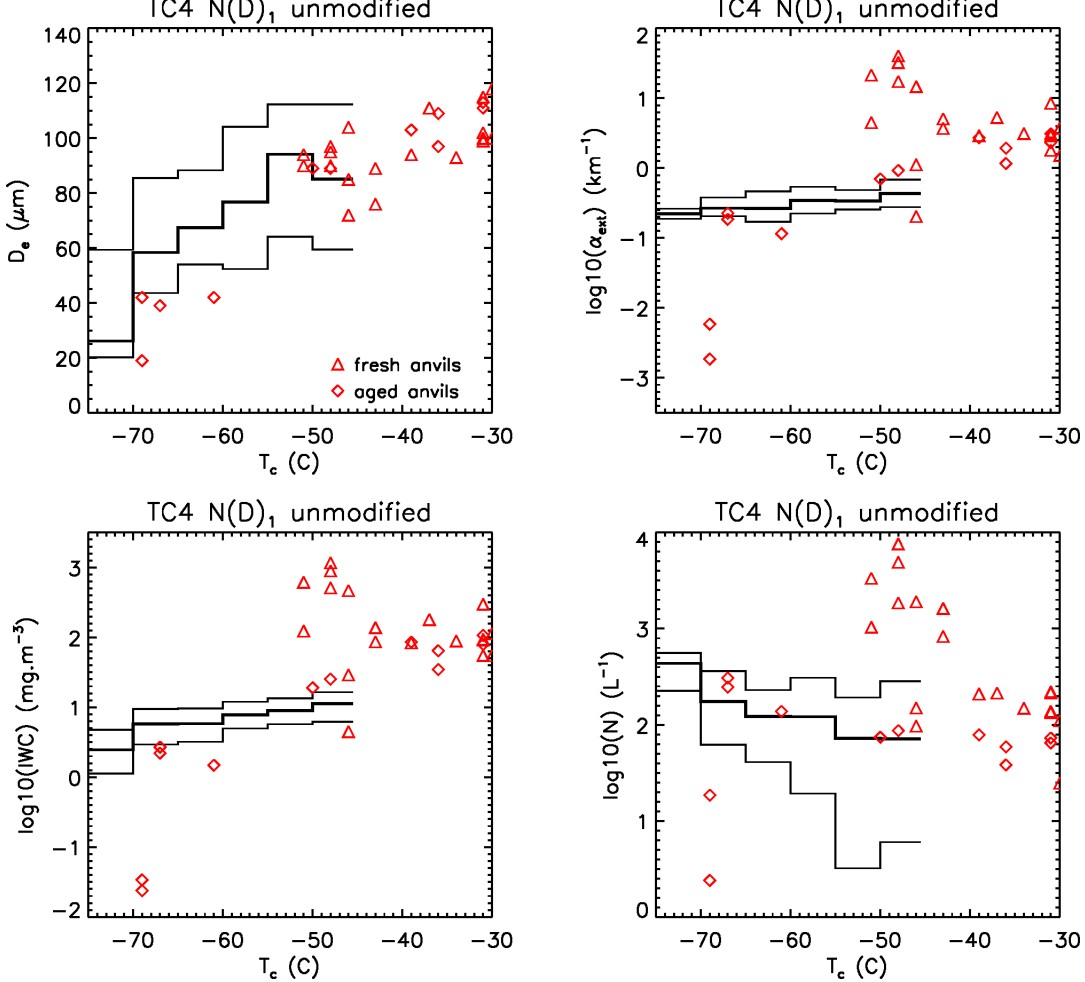

**Figure 10c. Temperature dependence of effective diameter $D_e$ (µm), extinction coefficient $\alpha_{ext}$ (km⁻¹), ice water content IWC (mg. m⁻³) and ice particle number concentration N (L⁻¹) during TC4 based on unmodified $N(D)_1$. In situ TC4 measurements in red are for fresh (triangles) and aged (diamonds) anvils. Corresponding CALIPSO IIR retrieved median properties using the TC4 relationships are given by the thick black histograms, with thin black histograms giving the 25th and 75th percentile values.**

-69°C). However, the in situ $\alpha_{ext}$ larger than 10 km⁻¹ between -52°C and -46°C are from fresh anvils sampled closer to the convective core and thus likely attenuate the CALIOP laser beam, and are therefore excluded from our cloud selection. For these fresh anvils, in situ N is unambiguously larger than N retrieved from CALIPSO in this temperature range, for both $N(D)_1$ assumptions. Otherwise, CALIPSO and in situ N are typically within a factor 2.

If future research produces convincing evidence that the $N(D)_1 = 0$ assumption is more realistic than the unmodified $N(D)_1$ assumption, then, based solely on the SPARTICUS data, the unmodified assumption may overestimate N by



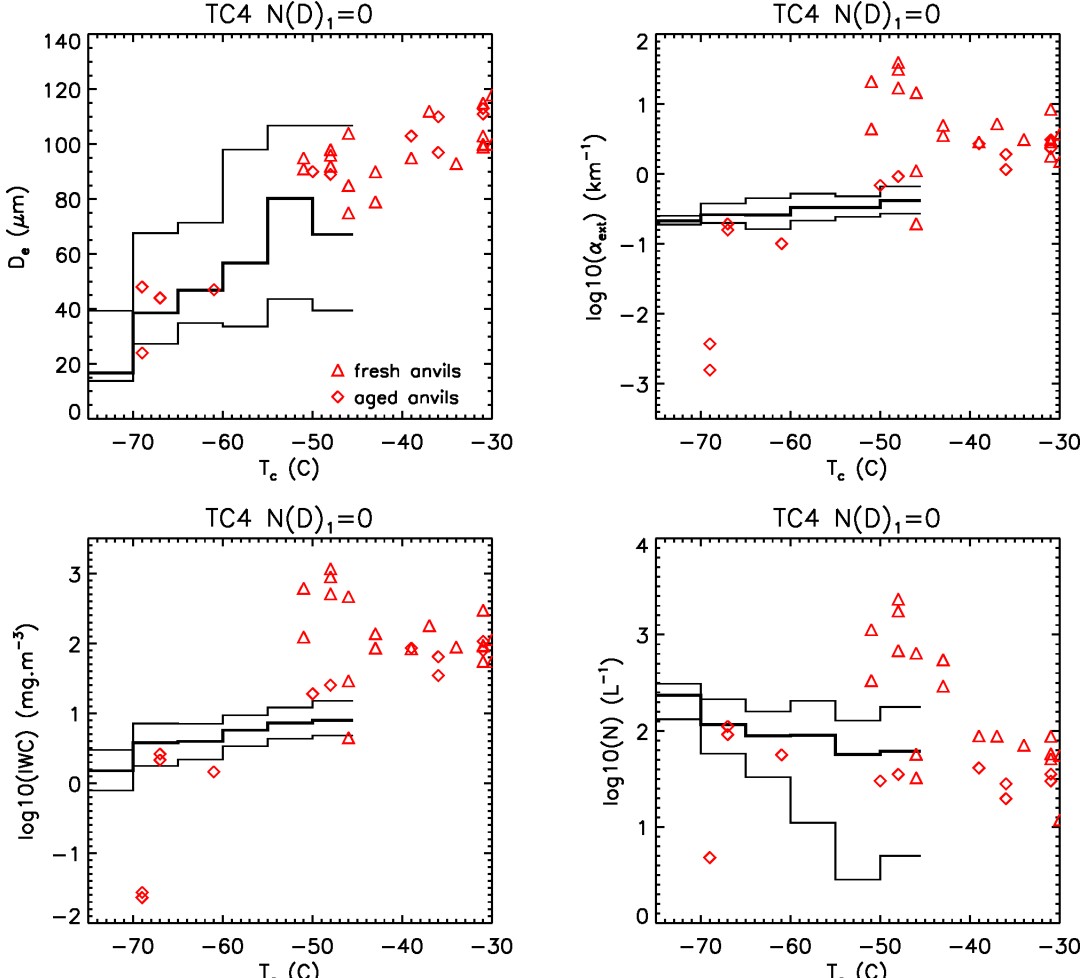

**Figure 10d. Same as Fig. 10c, except for the N(D)₁ = 0 assumption.**

about a factor of 3 and underestimate $D_e$ by up to ~ 1/3 for most cirrus clouds. Comparisons with IIR $\beta_{eff}$ could guide this analysis, keeping in mind that comparing in situ and layer average quantities can be challenging. Overall,

5    it is not yet clear which retrieval assumption yields the best agreement with in situ data.

Figure 10e compares retrievals of IWC and N with corresponding in situ values from the Krämer dataset. Both the retrieved and in situ N exhibit little temperature dependence at T > 200 K. Retrieved median N over ocean appear to be slightly lower than middle in situ values by a factor of 1.5 to 3 at -48°C, depending on the formulation used.

Below -75°C, retrieved N can be much larger. The divergence between the retrieved median and in situ middle

10    value for N for T < -75°C may be due to the in situ sampled cirrus often having mean layer extinction coefficients





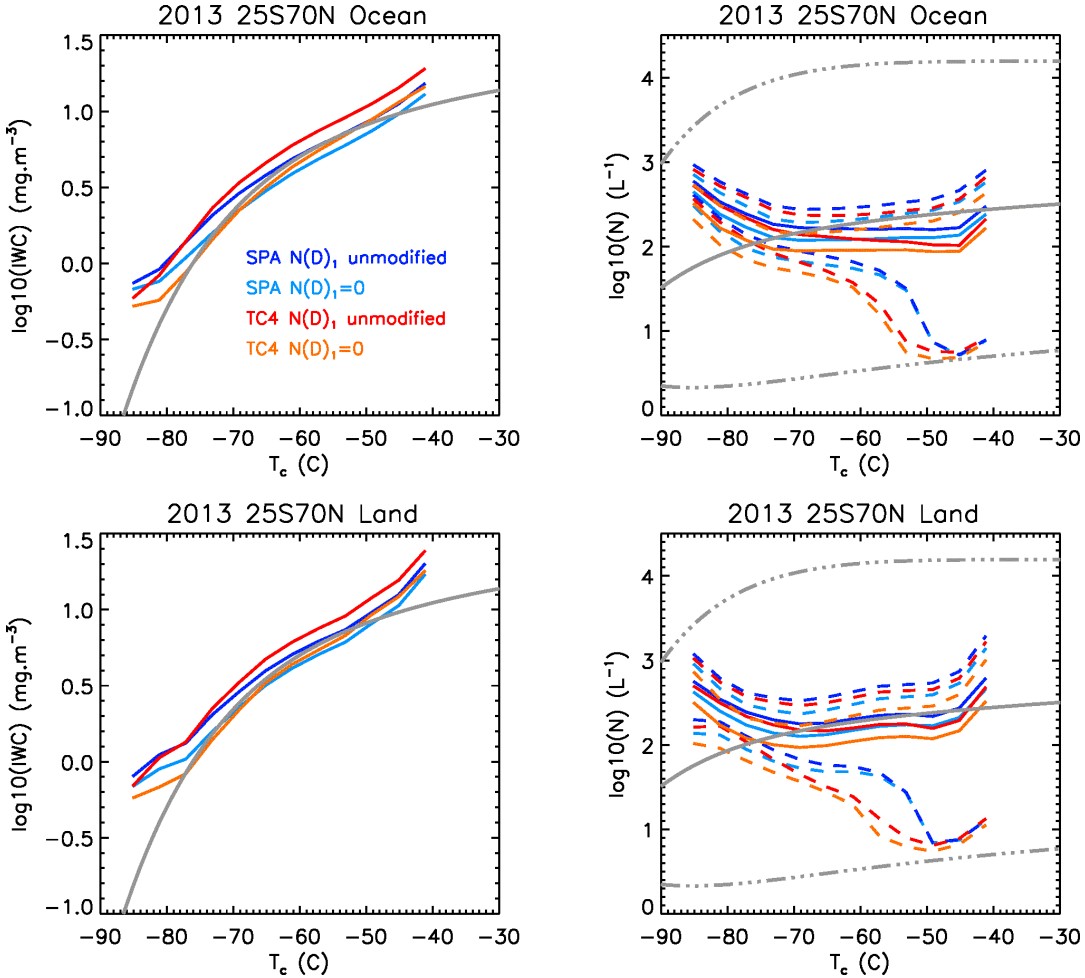

**Figure 10e : Same as Fig. 9, but for IWC (left, mg m⁻³) and N (right, L⁻¹).**

smaller than the IIR retrieval limit of about 0.05 km⁻¹ (see Fig. 7), resulting in the removal of ODs below ~ 0.3 from

the sampling statistics. TTL cirrus having OD < 0.3 are extensive in the tropics and have been characterized by

5    lower N (e.g. Jensen et al., 2013b; Spichtinger and Krämer, 2013; Woods et al., 2018). Retrieved median IWC is

very consistent with in situ middle values, with the $N(D)_1 = 0$ assumption yielding better agreement for T < -70°C.

In general, comparisons between the retrieved and in situ measured cloud properties during SPARTICUS and TC4

appear favorable despite the uncertainties involved. For any version of the four CALIPSO retrievals (Fig. 6),

relative differences in layer average N and $D_e$ in relation to different seasons and latitude zones should be

10    meaningful. From these relative differences, mechanistic inferences can be made and hypotheses explaining these

inferences can be postulated, keeping in mind the unique sensitivity of the technique to small crystals through $\beta_{eff.}$



## 5. Retrieval results

### 5.1 Frequency of occurrence of selected cirrus samples

As presented in Sect. 3, the sampled 1-km² IIR pixels are those for which the atmospheric column contains a single semi-transparent cloud layer of OD roughly between 0.3 and 3, of base temperature < 235 K, with a radiative

contrast between surface and the cloud of at least 20 K. This greatly limits the percentage of cirrus clouds sampled during this study (relative to all cirrus clouds).

Cirrus clouds of OD between 0.3 and 3 are geographically widespread across all latitudes and are also in an OD range that makes them radiatively important (Hong and Liu, 2015). Frequency of occurrence is defined as the number of cirrus cloud pixels sampled divided by the number of available IIR pixels. To clarify, a cirrus cloud

extending 20 km horizontally along a portion of the lidar track is counted 20 times whereas a cirrus cloud extending 5 km along this track is counted only 5 times.

Two years of CALIPSO IIR data are considered: 2008 (Dec. 2007 to Nov. 2008) and 2013 (March 2013 to Feb. 2014). It is noted that the version of the GMAO Met data used in the CALIPSO products is not the same in 2008 and in 2013. In 2008, it was GMAO GEOS 5.1 until Sept 2008 and GMAO GEOS 5.2 for Oct 2008 and Nov 2008.

In 2013, it is GMAO GEOS FP-IT for the whole period. Retrievals for each month of each year for all latitudes have been analyzed and organized into seasons, with winter as December, January, February (DJF); spring as March, April, May (MAM); summer as June, July, August (JJA); fall as September, October, November (SON). Frequency of occurrence is reported in Table 4 for each season and each 30° latitude zone, and for the entire planet during 2008 and 2013. The selection criteria result in very few sampled pixels relative to the number of available

IIR pixels, making the frequency of occurrence generally less than 2%. Thus what is most important in this analysis is not the actual frequency value but the relative differences in these values with respect to season and latitude. It is seen that despite our cloud subsampling, the geographical distribution of the occurrence frequencies is consistent with previous findings for ice clouds (T < 0°C) of OD between 0.3 and 3 (Hong and Liu, 2015). The greatest occurrence frequencies are in the tropics (i.e. 30° S-30° N). The occurrence frequency during Arctic (i. e. 60° N-82°

N latitude zone) winter is more than twice the frequency of other Arctic seasons. In the Antarctic (i.e. the 60° S-82° S latitude zone), frequency of occurrence is greatest in the spring and second-greatest during winter, in agreement with previous studies (Nazaryan et al., 2008; Hong and Liu, 2015). This is important since at high latitudes, the net radiative effect of ice clouds is strongest during the "cold season" where solar zenith angles are relatively low and ice cloud coverage is relatively high (Hong and Liu, 2015). Therefore, the cirrus cloud formation mechanism that

governs cirrus microphysical properties will have the greatest net cloud radiative effect (CRE) at high latitudes during winter and spring for the Antarctic and during winter for the Arctic.



**Table 4. Sampled cirrus cloud frequency of occurrence for each 30-degree latitude zone, and also for the entire globe (last line) during 2008 and 2013.**

| Occurrence of selected conditions (%) during 2008 (Dec 2007 to Nov 2008) | | | | |
|---|---|---|---|---|
| | DJF | MAM | JJA | SON |
| 60N-82N | 0.49 | 0.2 | 0.21 | 0.22 |
| 30N-60N | 0.74 | 0.96 | 0.60 | 0.73 |
| 0N-30N | 1.58 | 1.90 | 2.02 | 1.70 |
| 30S-0S | 1.58 | 1.46 | 0.75 | 1.07 |
| 60S-30S | 0.25 | 0.39 | 0.47 | 0.36 |
| 82S-60S | 0.16 | 0.19 | 0.31 | 0.72 |
| Full globe | 0.81 | 0.86 | 0.73 | 0.80 |

| Occurrence of selected conditions (%) during 2013 (March 2013 to Feb 2014) | | | | |
|---|---|---|---|---|
| | DJF | MAM | JJA | SON |
| 60N-82N | 0.61 | 0.31 | 0.16 | 0.24 |
| 30N-60N | 0.90 | 0.98 | 0.56 | 0.65 |
| 0N-30N | 1.43 | 1.82 | 1.86 | 1.76 |
| 30S-0S | 1.58 | 1.47 | 0.79 | 1.05 |
| 60S-30S | 0.30 | 0.36 | 0.46 | 0.35 |
| 82S-60S | 0.09 | 0.20 | 0.42 | 0.61 |
| Full globe | 0.82 | 0.86 | 0.71 | 0.78 |

## 5.2 Latitude, altitude and seasonal dependence of $\beta_{eff}$ and N

IIR $\beta_{eff}$ is a measure of the fraction of small ice crystals in the PSD (e.g. how narrow the PSD is) and is an important constraint for our retrievals. Figure 11 shows the latitude and altitude dependence of median $\beta_{eff}$ (left) and of the number of selected samples (right), where altitude is the cloud representative altitude, $Z_c$, defined as the centroid

10   altitude of the IIR weighting profile (Sect. 2.2.3). The plots are during 2008 and 2013, with from top to bottom DJF over oceans and over land, and JJA over oceans and land.



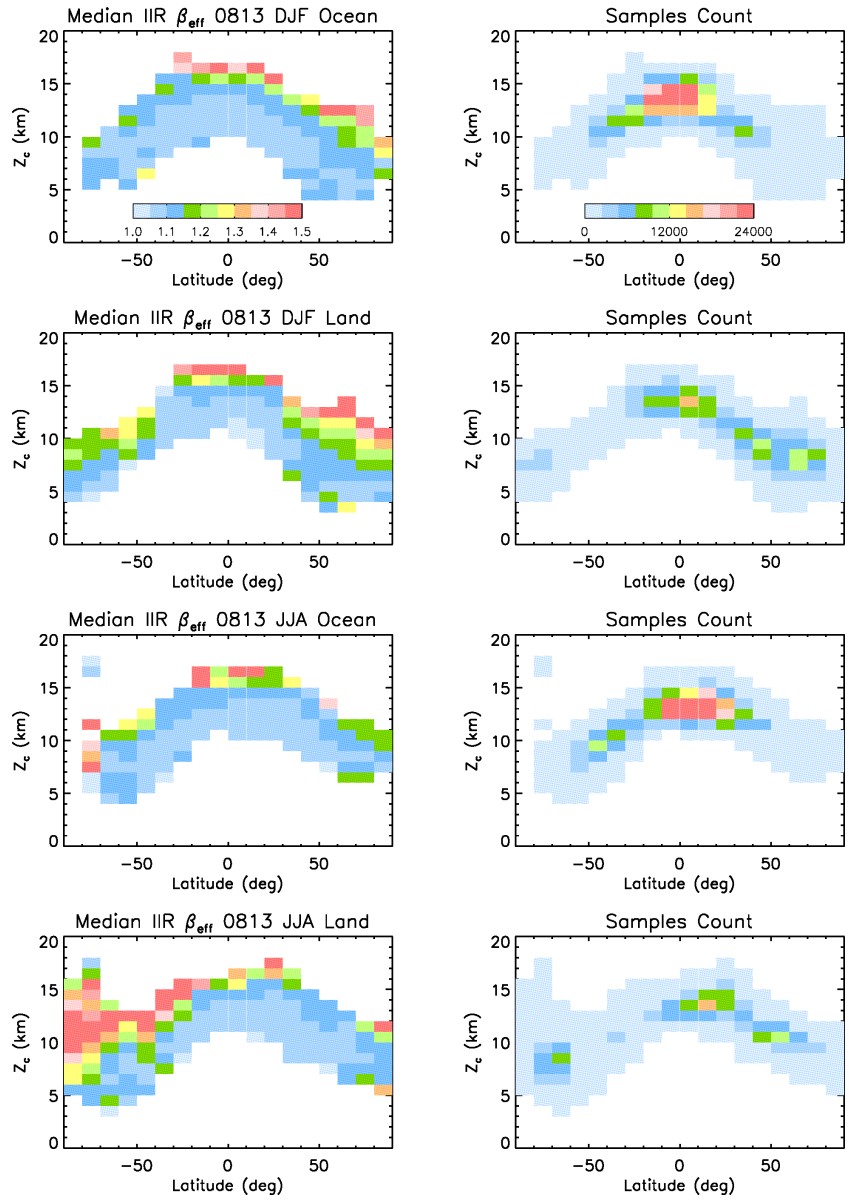

**Figure 11. Median IIR $\beta_{eff}$ (left) and samples count (right) vs. latitude and representative cloud altitude, $Z_c$, during 2008 and 2013. Panels from top to bottom are for DJF over oceans, DJF over land, JJA over oceans, and JJA over land.**

The majority of the sampled cirrus are in the tropical areas, between 20 °S and 10 °N in DJF and between 10 °S and

5  30°N in JJA. They are associated with anvil cirrus from deep convection and TTL cirrus. At mid- and high latitude in the Northern Hemisphere (NH), the sampled cirrus clouds tend to form at higher altitudes in summer than in winter. Over Antarctica during winter (JJA), the samples exhibiting centroid altitudes between 10-11 and 18 km





correspond to polar stratospheric clouds (PSCs) which are not found in summer, even though their base is in the troposphere per our data selection. The tropopause here is not well defined, allowing a continuum to exist between tropospheric cirrus and PSCs. Since there is less land in the SH mid-latitudes, the sample counts are fewer there. In the southern oceans, there tends to be more samples during winter (JJA).

Overall, median $\beta_{eff}$ decreases with decreasing altitude. It is larger than 1.2 (i.e. $D_e < 25$-40 µm, cf Figs. 3 and 4) at the top of the sounded atmosphere, prevailingly in the winter hemisphere. The lowest values of $\beta_{eff}$ ($< 1.1$) tend to be in the lower range of altitudes, and are abundant in the tropics and at mid-latitude in the NH in JJA.

Fig. 12a shows the latitude and altitude dependence of median N (left) and median $\Delta N/N$ based on the SPARTICUS data with $N(D)_1$ unmodified relationships, and Fig. 12b shows median N based on the three other formulations, with
from left to right SPARTICUS $N(D)_1=0$, TC4 $N(D)_1$ unmodified, and TC4 $N(D)_1=0$. The difference between the four formulations varies with $\beta_{eff}$ as shown in Fig. 6. As in Fig. 11, the panels from top to bottom are for DJF over oceans and land, and JJA over oceans and land. The number of samples can be seen in Fig. 11. Median $\Delta N/N$, which varies strongly with $\beta_{eff}$ (Fig. 7), is typically smaller than 50 % when $\beta_{eff}$ is larger, but often larger than 100 % at the lowest altitudes.

Median N is the lowest in the tropics (i.e. 20 °S-20 °N), and retrieved values are depending on the assumptions used, but are smaller than about 150 particles per liter. Over Antarctica during winter (JJA), the samples exhibiting centroid altitudes between 10-11 and 18 km corresponding to PSCs have lower N than the cirrus around 9-11 km for both JJA and DJF.

There is a tendency for N to increase at the lowest altitudes, and this occurs more outside the tropics. This may be
due to cirrus cloud layers corresponding to the warmest temperatures to be relatively shallow in geometrical thickness, because cirrus cloud base cannot be warmer than 235 K (as per our selection criteria). In geometrically thin cirrus, there is less dilution of the mean N due to ice sedimentation (Jensen et al., 2013a). Also, in thicker cirrus, sedimenting ice particles from above tend to quench hom in the mid-to-lower cloud by decreasing the $RH_i$, preventing the threshold $RH_i$ needed for hom. This can decrease N considerably (Spichtinger and Geirens, 2009a,
b). This effect would tend to produce higher N in geometrically thin cirrus relative to the thicker cirrus sampled at colder temperatures. Moreover, aggregation has less time to decrease N in thin cirrus.

### 5.3   Dependence of N on distance below cloud top

Our retrievals are now examined against both $T_c$ and $T_c$-$T_{Top}$ to estimate the impact of the distance from cloud top. The N retrievals are shown in Fig. 13 using the SPARTICUS $N(D)_1$unmodified assumption in the tropics (0-30°)





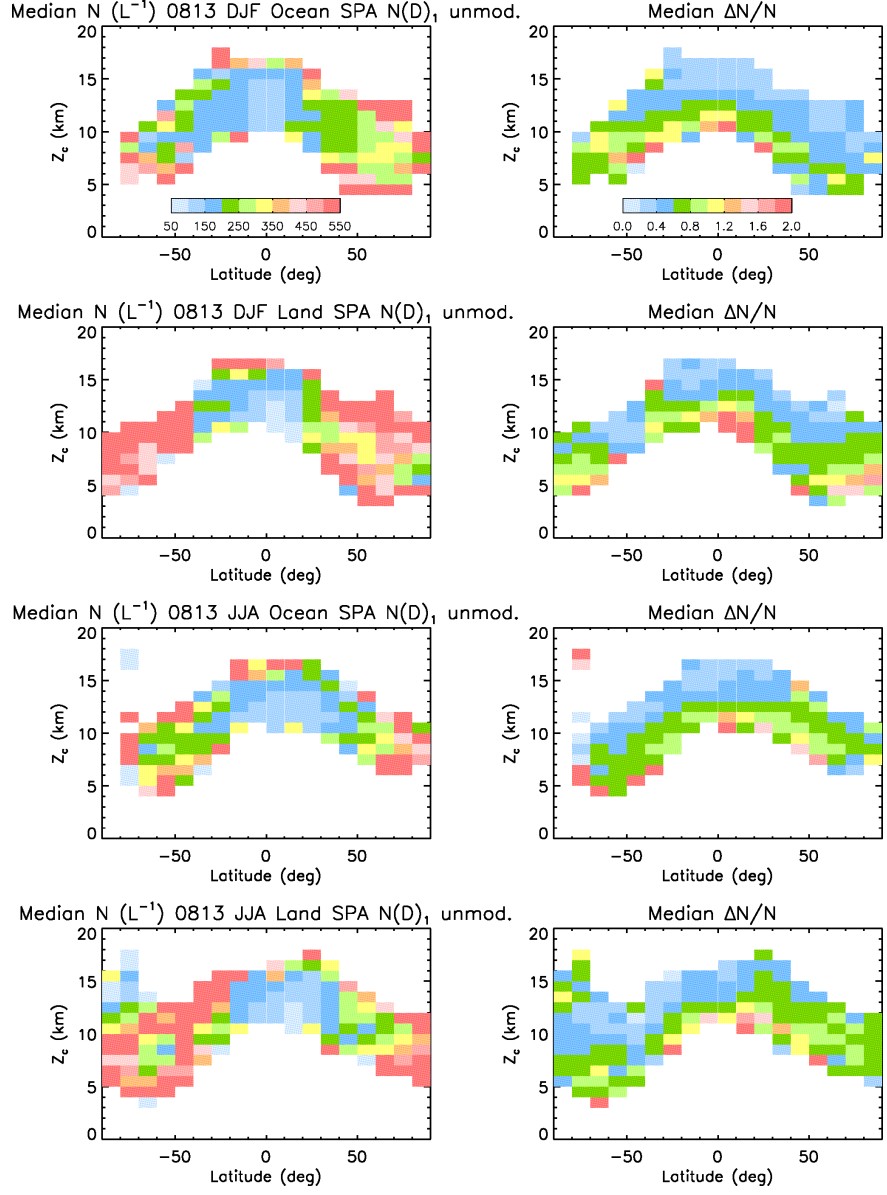

**Figure 12a: Median ice particle number concentration N (L⁻¹) retrieved from the SPARTICUS relationships assuming N(D)₁ unmodified (left) and associated relative uncertainty ΔN/N (right) vs. latitude and representative cloud altitude, Z$_c$, during 2008 and 2013. Panels from top to bottom are for DJF over oceans, DJF over land, JJA over oceans, and JJA over land.**

and at mid- (30-60°) and high (60-82°) latitudes in the winter and summer seasons (using both hemispheres) by

distinguishing retrievals over oceans and over land. The associated number of samples is given in Fig. 14. Fig. 13

shows a strong dependence of N on $T_c$-$T_{top}$, with large N (> 500 L⁻¹) seen near the top of the clouds, when $T_c$-$T_{top}$ is





**Figure 12b: Median ice particle number concentration N (L⁻¹) vs. latitude and representative cloud altitude, $Z_c$, during 2008 and 2013 using three formulations: SPARTICUS N(D)₁=0 (left), TC4 N(D)₁ unmodified (center), and TC4 N(D)₁=0 (right). Panels from top to bottom are for DJF over oceans, DJF over land, JJA over oceans, and JJA over land.**

smaller than about 5 °C. In contrast, the N dependence on $T_c$ is very weak (consistent with Figs. 10a-e). As seen in Figs. 13 and 14, the $T_c$-$T_{top}$ difference tends to be larger in the tropics than at mid- and high latitudes. The strong N dependence on $T_c - T_{top}$ appears to support the lower updraft cirrus cloud simulations of Spichtinger and Gierens (2009a, b) and the in situ measurements of Diao et al. (2015), although other possible explanations for this



**Figure 13.** Median retrieved ice particle number concentrations N (L$^{-1}$) using the SPARTICUS N(D)$_1$ unmodified formulation related to the representative cloud temperature $T_c$ and $T_c$ - $T_{top}$ at 0 °-30 ° (TRO, left), 30 °-60° (MID, center), and 60 °-82 ° (HIGH, right) during 2008 and 2013. Panels from top to bottom are for winter over oceans, winter over land, summer over oceans, and summer over land.

dependence cannot be ruled out. The strong dependence of N on $T_c$-$T_{top}$ (Fig. 13) appears weakest at high latitudes and at mid-latitudes during winter over land, with N being relatively high. If N near cloud top is due to hom as predicted by Spichtinger and Gierens (2009a, b), then hom appears more active near cloud top. Other attributes of Fig. 14 have been discussed in relation to Fig. 12.





**Figure 14.** Samples count vs. the representative cloud temperature $T_c$ and $T_c$ - $T_{top}$ at 0 °-30 ° (TRO, left), 30 °-60° (MID, center), and 60 °-82 ° (HIGH, right) during 2008 and 2013. Panels from top to bottom are for winter over oceans, winter over land, summer over oceans, and summer over land.

## 5.4 Effective diameter

The dependence of median $D_e$ on the representative temperature $T_c$ is shown in Fig. 15 for each latitude zone (tropics, mid- and high latitudes), with profiles for summer and winter for each zone (based on both hemispheres), over oceans and land. The analysis is using the SPARTICUS unmodified $N(D)_1$ (top) and the TC4 $N(D)_1 = 0$




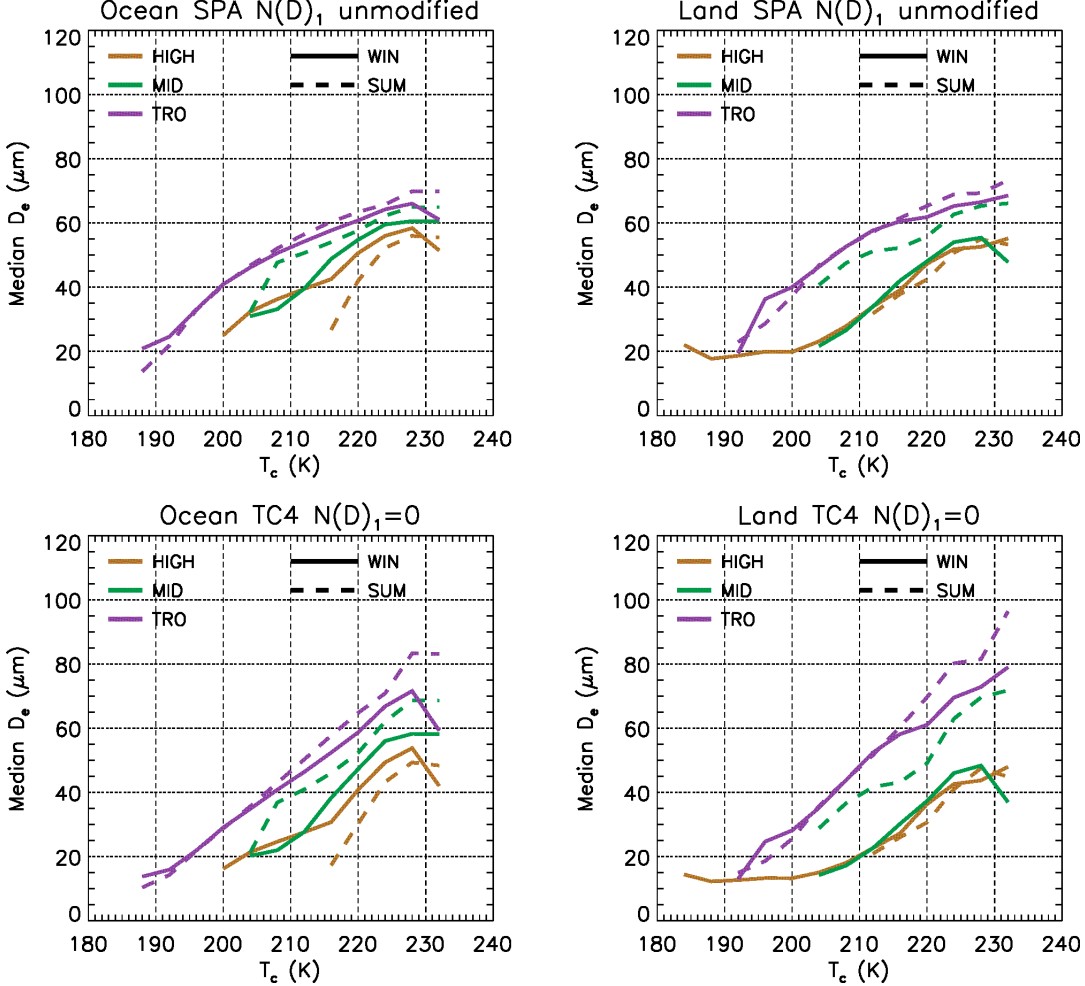

**Figure 15. Temperature dependence of the retrieved median effective diameter $D_e$ (μm) in winter (solid) and summer (dashed) during 2008 and 2013, based on the SPARTICUS N(D)$_1$ unmodified (upper panels) and the TC4 N(D)$_1$ = 0 (lower panels) formulations. Latitude zones are denoted by colors: purple: 0 °-30 ° (TRO); green: 30 °-60 ° (MID), brown: 60 °-82° (HIGH).**

(bottom) formulations. The TC4 curve fits yield a larger range of $D_e$ values than the SPARTICUS ones. At least 100 samples contributed to each data-point, and data from both years (2008 and 2013) were combined to generate these profiles. Consistent with the reported $\beta_{eff}$ -altitude relationships (Fig. 11), the $D_e$–temperature relationships show that $D_e$ for a given temperature and season is generally largest in the tropics, intermediate at mid-latitudes and smallest in the high latitudes. The profiles exhibit a considerable latitudinal and seasonal dependence, with latitudinal differences up to ∼ 40 μm for a given temperature. Seasonal differences may be up to about 20 μm for a given temperature at mid-latitude over land. Combined with the latitudinal and seasonal dependence of selected





**Figure 16: Median retrieved $D_e$ (μm) using the SPARTICUS $N(D)_1$ unmodified formulation vs. the representative cloud temperature $T_c$ and $T_c - T_{top}$ at 0 °-30 ° (TRO, left), 30 °-60° (MID, center), and 60 °-82 ° (HIGH, right) during 2008 and 2013. Panels from top to bottom are for winter over oceans, winter over land, summer over oceans, and summer over land.**

cirrus cloud frequency of occurrence (e.g. Table 4), these $D_e$ differences are likely to produce substantial variations in cirrus cloud net radiative forcing relative to a constant $D_e$ profile assumption.

Following the same reasoning as previously for N, Fig. 16 shows the dependence of median $D_e$ (using the SPARTICUS $N(D)_1$ =0 unmodified assumption) on $T_c$ and $T_c - T_{top}$, for the same latitude ranges and seasons as in





Figs. 13-15. Unlike N (Fig. 13), median $D_e$ depends strongly on $T_c$ with a somewhat weaker dependence on $T_c$-$T_{top}$. For instance, in the tropics, small median $D_e$ (< 50 µm) are found at any $T_c$ colder than about 205 K, but only near cloud top when $T_c$ is warmer than 205 K. At $T_c$ =220 K, median $D_e$ increases with $T_c$-$T_{top}$ from less than 50 µm near cloud top up to the upper retrieval limit around 80 µm at $T_c$-$T_{top}$ = 35 K.

## 6 Discussion and perspectives

### 6.1 Global and seasonal distribution of N

In the tropics, most cirrus are anvil cirrus and there is little difference in N between land and ocean, nor is there much seasonal dependence. A seasonal variation is seen in the NH at mid-latitude, more over land than over oceans, with larger N (and smaller $D_e$) during the winter than during the summer season (see Fig. 12a). This behavior is less
evident over land in the Southern Hemisphere (SH) where mid-latitude land mass is relatively small, but over the southern oceans where ice nuclei concentrations are relatively low (Vergara-Temprado, 2018), N is larger (and $D_e$ smaller) during winter. At high latitudes, N is relatively high and $D_e$ relatively low during both seasons. These observations appear generally consistent with the modeling results of Storelvmo and Herger (2014), where the Community Atmosphere Model version 5 (CAM5) was used to predict the global distribution of mineral dust at 200
hPa by season and the fraction of ice crystals produced by heterogeneous and homogeneous ice nucleation (henceforth het and hom). Dust concentrations in both hemispheres were minimal at high latitudes, especially during DJF. If dust is the main source of ice nuclei (e.g. Cziczo et al., 2013) and more ice crystals are produced through hom when dust concentrations are relatively low (e.g. Haag et al., 2003), then the above observations appear consistent with the predicted latitude dependence of dust concentrations. That is, lower ice nuclei concentrations
may result in higher $RH_i$ sufficient for hom to occur since the limited numbers of ice crystals produced via het may not exhibit sufficient surface area to draw down the $RH_i$ and prevent hom from occurring. While hom does not always result in relatively high N (e.g. Spichtinger and Krämer, 2013), hom can produce much higher N than het within sustained appreciable updrafts and is generally associated with higher N (e.g. Barahona and Nenes, 2008). Other observational studies indicating that hom is often important in determining the microphysics of cirrus clouds
are Mitchell et al. (2016), Zhao et al. (2018), Sourdeval et al. (2018) and Gryspeerdt et al. (2018).

Atmospheric dynamics may also help to explain the results in Fig. 12a. The tropical troposphere is well mixed due to deep convection, whereas in the mid-latitudes during winter, deep mixing is more limited. This reduced mixing should reduce the transport of ice nuclei to cirrus cloud levels (relative to the tropics). Snow cover during winter may also limit dust transport. Indeed, N is considerably higher during winter over land in the NH mid-latitudes, and
a similar but somewhat weaker seasonal relationship exists over oceans in the NH and SH mid-latitudes.



In general, we find higher N over mountainous regions, and mountain-induced waves may be responsible for higher N in the mid-latitudes over North America, especially during the winter season (having higher winds). The Andes Mountains in South America provide abrupt orography for mountain-induced waves that provide high updrafts favoring hom (Jiang et al., 2002; Hoffmann et al., 2016). This abrupt orography combined with minimal mineral

dust concentrations in this region, especially during DJF (Storelvmo and Herger, 2014), may explain the higher N over the Andes. Lower dust concentrations (Vergara-Temprado, 2018) may also be responsible for the higher N over the southern oceans.

### 6.2   Dependence of N on distance from cloud top

The temperature dependence of N on the cloud representative temperature $T_c$, as shown in Figs. 10a-b-c-d-e, is an

important relationship for understanding the physics of cirrus clouds. However, the in-cloud dependence of N on height below cloud top may also be important, as shown in Fig. 13. It was also shown with the SPARTICUS case study (Fig. 8a) that $\beta_{eff}$ can be sensitive to the $T_c$-$T_{top}$ temperature difference. The explicit synoptic cirrus cloud model of Spichtinger and Gierens (2009a, b) that describes the impact of aerosols on different types of nucleation mechanism indicates that the $T_c$-$T_{top}$ difference can be an important metric for understanding physical processes.

Using a constant vertical velocity of 0.05 m s$^{-1}$ in a modeled cirrostratus cloud, they show that the portion of an ice supersaturated region (ISSR) reaching the hom RHi threshold initiates cloud formation via hom, and though new ice crystal growth reduces the RHi, the RHi is not reduced much at this level since the rapid growth allows the ice crystals to fall more rapidly to lower levels. In this way a continuous supply of new ice crystals is maintained via hom near cloud top. As the ice crystals descend to lower levels and grow, they prevent the RHi from reaching the

hom threshold and N decreases substantially (since N near cloud top is spread vertically over the whole depth of cloud). In this way ice sedimentation plays a crucial role in the development and evolution of the cirrus cloud microphysical structure. These modeling results are also supported by the in situ measurements reported in Diao et al. (2015), which show the altitude dependence of the four evolution phases for synoptic cirrus clouds, which are ice nucleation, early ice crystal growth, later growth and sedimentation/ sublimation. The ice nucleation phase

generally occurs near the tropopause and cloud top while the other phases (e.g. growth of ice crystals) occur primarily below the ice nucleation layer. The above findings are also broadly consistent with those of Jensen et al. (2012; 2013a) that show N depends strongly on the ice sedimentation and entrainment/dilution processes.

### 6.3   Representativeness of these results

As mentioned, the cirrus clouds sampled in this study are optically thick (0.3 < OD < 3.0) and contribute a relatively

small fraction of total cirrus cloud coverage according to some cirrus cloud climatologies (e.g. Kienast-Sjögren et al., 2016; Goldfarb et al., 2001; Immler and Schrems, 2002). Thus it is possible that the cirrus cloud properties retrieved here are not representative of cirrus clouds in general. However, these are ground-based lidar studies





conducted at point locations whereas the global cirrus cloud climatology in Hong and Liu (2015) is based on the CALIPSO and CloudSat satellites. In that satellite study, where the ice cloud OD ranged from < 0.03 to > 20, the frequency of occurrence for cirrus having 0.3 < OD < 3.0 is much larger than these ground-based climatologies indicate, and this OD category appears to have the highest frequency of occurrence (see Fig. 7 of Hong and Liu,

2015). Moreover, in Kienast-Sjögren et al. (2016) where cirrus having OD > 0.3 comprised a very small fraction of the cirrus sampled, cirrus having OD > 0.3 still accounted for about half of the cloud net radiative forcing for cirrus cloud overcast conditions. If the Hong and Liu study is more realistic, then cirrus having 0.3 < OD < 3.0 should strongly dominate the overall cirrus cloud net radiative forcing.

## 6.4   Comparisons with other studies

Figure 17 shows the geographical distribution of median N based on the years 2008 and 2013, with N retrieved using the SPARTICUS unmodified (left) and TC4 $N(D)_1 = 0$ formulations (right), which correspond to the upper and lower N bounds, respectively. Figure 17 is for layers of representative temperature, $T_c$, between 218 and 228 K so that these results can be directly compared with the retrieved N values in Gryspeerdt et al. (2018). When averaging over all temperature levels sampled (not shown), our results are still comparable to those reported in Fig. 17.

These results are qualitatively similar to those reported in Fig. 1 of Gryspeerdt et al. (2018), including higher N over mountainous regions and over the southern oceans, with minimal N over the tropics. Sourdeval et al. (2018) have developed a new satellite retrieval for N regarding ice clouds, based on the CloudSat radar and the CALIPSO lidar (built upon the so-called DARDAR retrieval and referred to as the DARDAR-LIM scheme; LIM standing for Leipzig Institute for Meteorology). This retrieval uses the operational DARDAR retrieval products of IWC and $N_0^*$

(a PSD normalization factor that is a function of the IWC and $D_m$, where $D_m$ is the ice particle mean volume diameter, defined as the PSD moment ratio $M_4/M_3$), and also the "universal" normalized PSD described in Delanoë et al. (2005), to estimate N. Although their retrieval scheme is based on different physics than our scheme, these schemes yield similar results. However, there are important differences. Since our scheme is based on in situ cirrus cloud PSDs, especially the smallest ice crystal sizes, it naturally suffers from uncertainties endemic to the PSD

probes and uncertainties in how PSD shape may vary between regions (i.e. field campaigns). When $\beta_{eff}$ is greater than about 1.06, N retrievals vary by about a factor of 2 (Fig. 6), but N uncertainties are greater at smaller $\beta_{eff}$ where N/IWC and $D_e$ lose sensitivity to $\beta_{eff}$. This limits our ability to retrieve at relatively low N. On the other hand, the DARDAR-LIM scheme estimates N using the normalized universal PSD described in Delanoë et al. (2005). This normalized PSD has the form $N(D_{eq}) = N_0 D_{eq}^{\alpha} \exp(-k D_{eq}^{\beta})$, where $N_0$ and k are constrained by the radar and lidar

measurements, $\alpha$ and $\beta$ are fixed constants, and $D_{eq}$ is the melted equivalent diameter of an ice particle. Whereas $\beta_{eff}$ determines the PSD fraction of small (D < 50 µm) ice crystals in our scheme, this fraction is determined by $\alpha$ and k in the DARDAR-LIM scheme. Moreover, $\alpha$ depends on environmental conditions (Herzegh and Hobbs, 1985), with



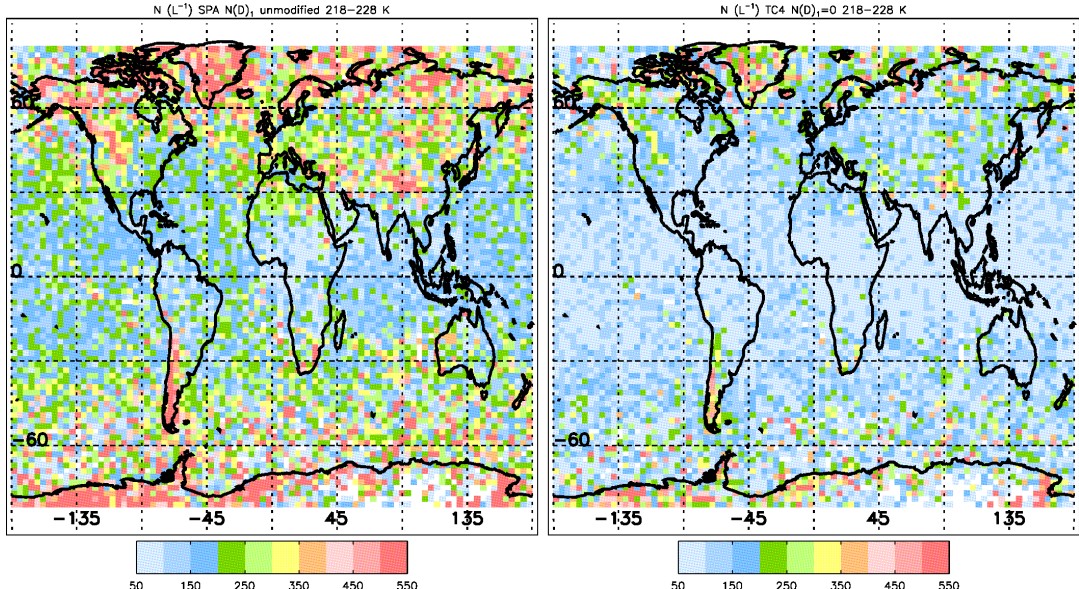

**Figure 17: Geographical distribution of median ice particle number concentration N (L$^{-1}$) during 2008 and 2013 where the layer representative temperature, T$_c$, is between 218 and 228 K. The retrievals are using two formulations: SPARTICUS N(D)$_1$ unmodified (left) and TC4 N(D)$_1$ = 0 (right).**

lower α values (which promote higher N) associated with higher updrafts and ice crystal production rates (such as might be found in cirrus clouds over mountainous terrain). Furthermore, radar measurements are not sensitive to small particles (Deng et al., 2010, 2013), in contrast to the lidar and to the IIR, and DARDAR-LIM retrievals using the lidar-only mode rely strongly on a priori relationships to retrieve N. In a general sense, in the case of clouds detected only by the radar or only by the lidar, the DARDAR algorithm does not benefit from the synergism

between these two measurements, making the a priori relationships essential to obtain closure.

It is evident that N varies by about a factor of 4 in Fig. 1 of Gryspeerdt et al. (2018) and by about a factor of 7 in Fig. 17 of this study. The greater N range of this study might be partly due to the use of β$_{eff}$ without the constraint of a parameterized PSD.

The DARDAR-LIM retrieval also shows that N has a strong temperature dependence, with N increasing with

decreasing temperature T. This dependence was not observed in the field campaigns analyzed in Krämer et al. (2009), nor is it predicted by our retrieval for T > 200 K (Fig. 10e).

The dependence of N on distance from cloud top was also evaluated in Fig. 5 of Gryspeerdt et al. (2018). A decrease in N with increasing distance from cloud top was evident in that study for cloud top distance more than ~



0.5 km (based on ice particle sizes > 5 μm), although the extent of decrease appears less than often observed in Fig. 14. A comparison is complicated by the different scales used (linear vs. log) and different data treatments.

The regional variation in median N might be greater than indicated by Fig. 17. Regarding TC4 and Fig. 8b, the $N(D)_1 = 0$ assumption yielded the best agreement between in situ and retrieved $\beta_{eff}$, whereas a similar comparison

with SPARTICUS data (Fig. 8a) may arguably favor the unmodified $N(D)_1$ assumption (since PSD sampled at T < -60 °C were from a single cirrus event). Thus it is possible that the TC4 $N(D)_1 = 0$ formulation of our retrieval scheme is most representative for tropical cirrus and that the SPARTICUS $N(D)_1$ unmodified version is most appropriate for mid-to-high latitudes. This would double the N range estimated from the color-bar legend in Fig. 17, with N varying by a factor of about 14 (note that N << 50 L$^{-1}$ is within the dynamic retrieval range of the TC4 $N(D)_1$

$= 0$ scheme, which is evident from Fig. 6 and Eq. 8).

Ice cloud CALIPSO-CloudSat retrievals of $D_e$ are reported in Fig. 12 of Hong and Liu (2015) against temperature in terms of season and latitude zone for ODs ranging from < 0.03 to > 20. Keeping in mind the different cloud sampling (in particular OD range and our cloud base temperature being colder than 235 K), comparing their seasonal $D_e$ changes with those in this study is not straight-forward. Nevertheless, mean $D_e$ values are comparable,

in particular for the TC4 $N(D)_1 = 0$ assumption at temperatures colder than 235 K (the upper limit in this study). Their $D_e$ in the tropics at 190 K is about 40 μm compared to 20 μm in this study. At 220 K, their $D_e$ around 70 μm is in fair agreement with our $D_e$ found between 60 and 70 μm. Heymsfield et al. (2014, their Fig. 11) report small $D_e$ around 20 μm at temperature colder than -72°C, in agreement with this study, but they find a steeper increase of $D_e$ with temperature. Detailed comparisons with other work, which are beyond the scope of this paper, should account

not only for temperature, but also for distance from cloud top.

### 6.5   A possible link between high latitude cirrus and mid-latitude weather

The retrieval results in Table 4 indicate that at high latitudes there tends to be the greatest cirrus cloud coverage during winter in the Arctic and during spring (SON) in the Antarctic (where relatively high N and small $D_e$ occur throughout the year in both regions, as shown in Figs. 12, 15 and 17). While this study only considers a subset of

cirrus clouds and two years of retrievals, our findings on the seasonal dependence of cirrus cloud coverage are consistent with other satellite cirrus cloud studies that consider a broader range of conditions over longer periods (Nazaryan et al., 2008; Hong and Liu, 2015). Independent of the macro- and microphysical cirrus cloud attributes found in this study, at high latitudes there are important seasonal changes to the cirrus cloud shortwave (SW) and longwave (LW) radiative forcing due to a changing solar zenith angle. The SW and LW components almost cancel

during summer, but during winter, the LW component strongly prevails, producing a strong net warming at the top of atmosphere (TOA) and at the surface (Hong and Liu, 2015; Storelvmo et al., 2014). This indicates that the strongest net radiative forcing by cirrus clouds on Arctic (Antarctic) climate occurs during winter (spring). This




seasonal cycle of the solar zenith angle combined with the unique macro- and microphysical properties of Arctic cirrus during winter suggests that wintertime Arctic cirrus may have a significant warming effect on Arctic climate. A satellite remote sensing study of ice clouds (T < 0°C) by Hong and Liu (2015) found that at high latitudes, ice cloud net radiative forcing at the TOA and at the surface during the cold season is > 2 W m$^{-2}$ for a cirrus cloud OD

of 1.5. Since the most severe effects of global warming occur at high latitudes, it is critical to understand the factors controlling the macro- and microphysical properties of high latitude cirrus clouds.

A potential link to mid-latitude winter weather is the possible impact of the winter Arctic cirrus on the meridional (north-south) temperature gradient between the Arctic and mid-latitudes. The cirrus-induced winter warming described above will occur throughout the troposphere (Chen et al., 2000; Hong and Liu, 2015), and will thus act to

reduce this temperature gradient in the upper troposphere (UT). While it is not clear how this would impact weather, some type of impact is likely if the warming is significant, and several possible scenarios are described in Cohen et al. (2014) and Barnes and Screen (2015). While many papers have been published recently regarding potential effects of Arctic Amplification (henceforth AA; the observation that the mean Arctic temperature rise due to greenhouse gases is at least a factor of two greater relative to the adjacent mid-latitudes) on mid-latitude weather,

it is important to note that the AA related to sea ice loss and associated sea surface temperature increases primarily affects temperatures between the surface and 700 hPa (Screen et al., 2012), while heating due to winter cirrus would strongly affect the UT. A theoretical link between AA and the jet-stream is found in the thermal wind balance, which states that a reduced meridional temperature gradient tends to produce a reduced vertical gradient in the zonal-wind field, depending on other factors like changes in surface winds, storm tracks and the tropopause height

(Barnes and Screen, 2015). Thus, AA could lead to a weaker jet-stream having more amplified Rossby waves and associated extreme weather events as hypothesized by Francis and Vavrus (2012; 2015), but it is currently not clear whether such a phenomenon is occurring or will be occurring (Barnes and Screen, 2015).

As described in Barnes and Screen (2015), GCM simulations from the fifth Coupled Model Intercomparison Project (CMIP5) show that while the lower troposphere during Arctic winter is projected to warm substantially by 2100, this

is not happening in the Arctic UT where little warming is projected. Moreover, in the tropics the models predict the strongest warming in 2100 occurs in the UT. These effects decrease the meridional temperature gradient at low levels and increase the temperature gradient in the UT. These low- and high-level gradients have competing effects on the jet-stream, with a decreasing low-level gradient acting to weaken the jet-stream and shift it towards the equator, while an increasing UT gradient acts to strengthen the jet-stream and shift it poleward (Barnes and Screen,

2015). An interesting question to ask here is whether the CMIP5 GCMs adequately describe the changes in winter Arctic cirrus that satellite remote sensing studies observe. If they do not, and the winter heating from Arctic cirrus clouds is underestimated in the models, then the meridional UT temperature gradient may be overestimated during winter. If this were the case, then increasing Arctic cirrus coverage during winter in the models would tend to



weaken the simulated jet-stream and shift it further towards the equator. Future GCM research should determine whether predicted cirrus cloud coverage and microphysics is consistent with the results from satellite studies such as this one, and strive for consistency with these remote observations. Then it could be determined whether the UT heating from the winter Arctic cirrus could be a significant factor affecting the simulated NH mid-latitude

circulation.

A related question is whether wintertime Arctic cirrus are increasing, causing a change in jet-stream behavior. Poleward transport of heat and moisture is a fundamental attribute of the Earth's climate system, and the atmosphere of a warming climate can hold more water vapor (Dufour et al., 2016). While mid-latitude frontal systems are a primary component of this poleward transport, extreme transport events that can be described as water vapor

intrusions (WVI) may account for 28% of the total moisture transport into the Arctic (Woods et al., 2013). During WVIs, low level winds from below the Arctic Circle (66 °N) penetrate deep into the Arctic mostly during winter (Johansson et al., 2017). These WVI destabilize the Arctic boundary layer, and can increase both low- and high-cloud coverage over the Arctic, with cirrus clouds increasing mostly in winter by 15-30%. Both Screen et al. (2012) and Francis and Vavrus (2015) found evidence of increased remote energy transport into the Arctic, especially after

2000. This occurred mostly during the fall through enhanced planetary wave amplitudes (Francis and Vavrus, 2015). This transport may have contributed to the observed increase of Arctic cirrus clouds during winter (Table 4). These remote effects may enhance Arctic winter cirrus and their associated heating rates, which may affect jet-stream dynamics.

### 7    Summary and conclusions

A new satellite remote sensing method was developed to retrieve the ice particle number concentration N within cirrus clouds, along with effective diameter $D_e$, ice water content (IWC) and path (IWP), and visible optical depth (OD). This was made possible by exploiting the fact that most of the cloud emissivity difference between the split-window channels at 11 and 12 µm is due to wave resonance absorption, a process sensitive to the smallest ice crystals that dominate N (Mitchell et al., 2010). Due to this process, a tight relationship between N/IWC and $\beta_{eff}$

was obtained. This relationship, and a similar tight relationship between $D_e$ and $\beta_{eff}$, are the unique aspects of this retrieval and make it self-consistent through the shared dependence on $\beta_{eff}$. Although the retrieval is restricted to single-layer cirrus cloud ODs between about 0.3 and 3.0 (which excludes most TTL cirrus), this OD range is likely to be the most radiatively significant range due to the lower cirrus cloud frequency of occurrence at higher ODs and a much lower cirrus cloud mean emissivity at the lower ODs (Hong and Liu, 2015). In other words, for the sampled

single layer clouds, the cirrus clouds that the IIR senses best in the window channels will also have the most influence on the Earth's longwave radiation budget.



A two-year global and seasonal analysis of these CALIPSO observations reveals that N depends on the latitude zone, season and surface type (land vs. ocean). In the relatively pristine high latitudes, N was relatively high and $D_e$ was relatively low, suggesting that homogeneous ice nucleation may be an important process at these latitudes. In the tropics, N was lowest and $D_e$ was largest on average, relative to the high- and mid-latitudes, with little seasonal

dependence. There was considerable seasonal dependence regarding median N and $D_e$ in the mid-latitudes, with N being higher and $D_e$ being smaller during winter (for a given temperature), especially over land in the NH and over ocean in the SH.

The objective of this paper was not to determine absolute magnitudes for the retrieved quantities, but rather to show how they vary in terms of temperature, cloud thickness, latitude, season and topography, using any of the four

formulations of this retrieval. For a given formulation, these relative differences were similar and were not sensitive to the retrieval formulation used. The N retrieval described herein is an advancement of the work described in Mitchell et al. (2016), which supports many of the findings in Gryspeerdt et al. (2018), including the dependence of homogeneous ice nucleation in cirrus clouds on topography and latitude. Consistent with other satellite studies on ice clouds, the optically thicker cirrus clouds we studied exhibited a strong seasonal dependence in the Arctic in

regards to frequency of occurrence, with cirrus during winter at least twice as likely to occur relative to other seasons. This might possibly have a significant effect on jet stream dynamics.

Future cirrus cloud field campaigns designed to sample small ice crystals in regions not representative of the TC4 and SPARTICUS domains (such as at high latitudes or over mid-latitude oceans) may further improve upon this retrieval framework. It may also be beneficial if more information relevant to remote sensing synergism is taken

during in situ observations.

**Appendix: Retrieval uncertainty analysis**

We begin this analysis with our retrieval equation for the ice particle number concentration, N:

$$N = \frac{\rho_i}{3} \times \frac{\left[ \frac{2}{Q_{abs,eff}} (12\,\mu m) \right] \cdot \tau_{abs}(12.05\,\mu m)}{\Delta z_{eq}} \times D_e \times \left( \frac{N}{IWC} \right) \qquad (A1)$$

with $\rho_i = 0.917 \times 10^6$ g m$^{-3}$. The quantities N/IWC, $D_e$, and 2/$Q_{abs,eff}$(12 μm) are retrieved from $\beta_{eff}$ using the

regression curves given in Table 1 for the four formulations. By writing x= $\beta_{eff}$, they can be expressed as

$$\left( \frac{N}{IWC} \right) (g^{-1}) = 10^9 (a2.x^2 + a1x + a0) \qquad (A2)$$



$$D_e\ (\mu m) = (b2.x^2 + b1.x + b0)^{-1} \tag{A3}$$

$$2/Q_{abs,eff}(12\ \mu m) = c2.x^2 + c1.x + c0 \tag{A4}$$

Equation. (A1) can be re-written as:

$$N(L^{-1}) = f(x) \times \alpha_{abs}(km^{-1}) \tag{A5}$$

with

$$f(x) = 10^{-6}\left(2\!\!\Big/_{Q_{abs,eff}}(12\mu m)\right) \cdot \left(\frac{N}{IWC}\right)(g^{-1}) \times D_e(\mu m) \times \frac{1}{3.27} = \frac{a2.x^2 + a1x + a0}{b2x^2 + b1x + b0} \cdot \times \frac{c2.x^2 + c1x + c0}{1} \times \frac{10^3}{3.27} \tag{A6}$$

and

$$\alpha_{abs} = \frac{\tau_{abs}(12.05\,\mu m)}{\Delta z_{eq}} \tag{A7}$$

Assuming a negligible error in $\Delta Z_{eq}$, and writing $\tau_{abs}(12.05\ \mu m)$ as $\tau_{12}$ and $\tau_{abs}(10.6\ \mu m)$ as $\tau_{10}$ for more clarity, so

that $x = \tau_{12}/\tau_{10}$, the derivative of N can be written:

$$\frac{dN}{N} = \frac{1}{f}\frac{\partial f}{\partial x} \cdot x \cdot \left(\frac{d\tau_{12}}{\tau_{12}} - \frac{d\tau_{10}}{\tau_{10}}\right) + \frac{d\tau_{12}}{\tau_{12}} \tag{A8}$$

In Eq. (A8), the derivative of $x = \beta_{eff}$ is:

$$dx = d\beta_{eff} = x \cdot \left(\frac{d\tau_{12}}{\tau_{12}} - \frac{d\tau_{10}}{\tau_{10}}\right) \tag{A9}$$

Errors in $\tau_{12}$ and in $\tau_{10}$ are computed by propagating errors in i) the measured brightness temperatures $T_m$, ii) the

background brightness temperatures $T_{BG}$, and iii) the blackbody brightness temperatures $T_{BB}$ (Garnier et al., 2015).

The uncertainties in $T_{m10}$ at 10.6 μm and in $T_{m12}$ at 12.05 μm are random errors set to 0.3 K according to the IIR

performance assessment established by the Centre National d'Etudes Spatiales (CNES) assuming no systematic bias

in the calibration. They are statistically independent.

Because the same cloud temperature is used to compute $\tau_{12}$ and $\tau_{10}$, the uncertainty $\Delta T_{BB}$ is the same at 10.6 and at

12.05 μm. A random error of +/-2K is estimated to include errors in the atmospheric model.



After correcting for systematic biases based on differences between observations and computations (BTDoc) in cloud-free conditions, the random error $\Delta T_{BG}$ in $T_{BG}$ is set from the standard deviation of the resulting distributions of BTDoc. Over ocean, nighttime and daytime standard deviations at 12.05 μm are similar, and found smaller than over land, where the deviations tend to be larger during daytime than at night. For simplicity, $\Delta T_{BG}$ at 12.05 μm is

set to ± 1K over ocean, and to ± 3K over land for both night and day. Standard distributions of BTDoc(10.6 μm) - BTDoc(12.05 μm) indicate whether the errors in $T_{BG}$ at 10.6 and 12.05 μm are canceling out or not, after accounting for the contribution from the observations, which is estimated to √2x0.3 = 0.45 K. Standard deviations of [BTDoc(10.6 μm) - BTDoc(12.05 μm)] are found smaller than 0.5 K over ocean and over land during nighttime, which indicates that the errors $\Delta T_{BG}$ in $T_{BG}$ at 12.05 μm and at 10.6 μm can be considered identical. They are found

locally up to 1 K during daytime over land, which could reflect a variability of the 10.6-12.05 difference in surface emissivity, but also the presence of residual clouds.

Finally, the relative uncertainty $\Delta N/N$ is written as:

$$\left(\frac{\Delta N}{N}\right)^2 = \left[\frac{1}{f}\frac{\partial f}{\partial x} \cdot x \cdot \left(\frac{\partial \tau_{12}}{\tau_{12} \cdot \partial T_{BG}} - \frac{\partial \tau_{10}}{\tau_{10} \cdot \partial T_{BG}}\right) + \frac{\partial \tau_{12}}{\tau_{12} \cdot \partial T_{BG}}\right]^2 \cdot \Delta T_{BG}^2 + \left[\frac{1}{f}\frac{\partial f}{\partial x} \cdot x \cdot \left(\frac{\partial \tau_{12}}{\tau_{12} \cdot \partial T_{BB}} - \frac{\partial \tau_{10}}{\tau_{10} \cdot \partial T_{BB}}\right) + \frac{\partial \tau_{12}}{\tau_{12} \cdot \partial T_{BB}}\right]^2 \cdot \Delta T_{BB}^2$$

$$+ \left[\left(\frac{1}{f}\frac{\partial f}{\partial x} \cdot x + 1\right) \cdot \frac{\partial \tau_{12}}{\tau_{12} \cdot \partial T_{m12}}\right]^2 \cdot \Delta T_{m12}^2 + \left[\left(\frac{1}{f}\frac{\partial f}{\partial x} \cdot x\right) \cdot \frac{\partial \tau_{10}}{\tau_{10} \cdot \partial T_{m10}}\right]^2 \cdot \Delta T_{m10}^2 \qquad (A10)$$

**Competing interests:** All the authors declare that they have no conflict of interest.

**Acknowledgements**: This research was supported primarily by NASA grant NNX16AM11G and the NASA

CALIPSO project. Earlier funding was provided by the Office of Science (BER), US Department of Energy, and additional support was obtained through the CNES Eeclat project. Dr. Melody Avery is deeply acknowledged for fruitful discussions and her invaluable comments at every stage in the progress of this work. Dr. Martina Krämer is gratefully acknowledged for providing us with the curve fits that describe her 2009 data set of cirrus cloud in situ data. CALIPSO products are available at the Atmospheric Science Data Center of the NASA Langley Research

Center and at the AERIS/ICARE Data and Services Center in Lille (France).



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
