# Peer review of "CALIPSO (IIR-CALIOP) Retrievals of Cirrus Cloud Ice"

_Atmospheric Chemistry and Physics, 2018_

## Referee Comment (RC1) · Anonymous Referee #1 · 8 Aug 2018

Review of 'CALIPSO (IIR-CALIOP) Retrievals of Cirrus Cloud Ice Particle Concentrations' by Mitchell et al.

This manuscript describes and evaluates a new method dedicated to retrieve the ice particle number concentration N from satellite measurements. The method presented here infers N by using the sensitivity of a couple of thermal infrared channels to the concentration in small particles. It is applicable to cirrus with an optical depth between 0.3 and 3 and a base temperature $T < 235°C$. The method notably depends on several relationships between the cloud effective absorption coefficient ratio $\beta_{\text{eff}}$ and diverse ice cloud optical and microphysical properties, that the authors deduce from in situ measurements. Its results are first evaluated through comparisons to in situ measurements of N and two years of global retrievals are then analyzed. Uncertainties on the retrievals are also discussed. The authors show that, despite large uncertainties on the absolute N values, the spatial variability observed in global distributions are consistent with expectations based on modeling, in situ and/or remote sensing studies.

I find this retrieval method, which provides a new yet simple way to interpret and extract information from thermal infrared measurements, very innovative and interesting. There are of course shortcomings, such as the very low ice cloud sampling after filtering or the strong dependence on relations obtained from limited in-situ data, but the authors properly discuss most of these in detail and account for them in their discussions/conclusions. This study can be seen as preliminary results of a method that could easily be improved in the future (by including more in situ data in its framework). This type of work is necessary considering that very few methods today allow for retrieving N from satellite despite the importance of this parameter to study cloud processes, understand aerosol-cloud interactions or evaluate models. The manuscript is well written but I find its length to be a serious issue. This study indeed covers a thorough technical description of the method, very detailed evaluation against in situ measurements, an analysis of global retrievals, comparisons to existing methods and even discussions on possible perspectives/applications. 51 pages from the abstract to the appendix, and 17 figures that are often multi-panel, is (even in an ACDP format) too much for this type of paper. I strongly encourage the authors to shorten the manuscript to not overwhelm the readers, who may then miss some important points. Overall, I therefore recommend for publication of these results, providing revisions following the comments listed below.

General comments:

1. A major concern with the current manuscript is its length. The authors cover a very wide range of information and results that are all interesting but their amount is, in my opinion, counterproductive. It should be kept in mind that this study will be of high interest to diverse communities, as it contains are interesting aspects for remote sensing, in situ measurements, modeling or ice microphysics. Based on the current version I suspect that readers will only focus on their section of interest and miss important results. One idea could be to shorten section 2, since a big part of it has already been described in Mitchell et al. [2016]. The results related to the $N(D)_1 = 0$ analyses are very important and should be mentioned in the paper but several of them could be moved to supplementary materials in order to lighten the discussions and the figures. Other analyses that are not focused on N (the novelty of the paper), such as section 5.4, could also be moved to supplementary materials. This would allow to remove some technical discussions and focus the paper on result analyses, which are also better suited for a publication in ACP.

2. The authors acknowledge in the manuscript that this new retrieval method is not expected to provide accurate absolute values of N. Only the spatial variability of this parameter is well represented. The accuracy of the absolute N is discussed throughout sections 3 and 4 but would it be possible to summarize a final estimate of the uncertainties on N in the conclusion? By this I mean not only the $\Delta N/N$ but also combining what has been learned from the in situ evaluation. For instance, would it be a factor of 2, 5, one order of magnitude? Also, in what conditions can optimal retrievals be expected? This type of information would be very useful to future users of this dataset, especially since this paper will serve as a reference..

3. A clear limitation of the method is its necessity to filter out a lot of data based on different cloud conditions. This, as shown in table 4, leads to samples that are only representative of up to a 2% frequency of occurrence. Is there a particular reason for not using more data? The method seems to be rather straightforward to apply on CALIPSO satellite products, and some of the co-authors should be familiar with such operational treatments. It therefore shouldn't take long to process 10 years of data and greatly improve the statistical significance of the results shown in this study (especially in sections 5 and 6).

4. Concerning the results shown in section 4 (comparisons to in situ measurements), more direct comparisons between SPARTICUS measurements and CALIPSO could be obtained by looking at the numerous co-incident overpasses between the satellite and the aircraft. This would avoid issues related to non-representative events in the in situ data by comparison to a long-term regional dataset, as noted in page 22. Have you tried looking at these exact satellite overpasses only?

5. Perhaps I have missed this information but I am under the impression that the $D_{eff}$ retrieved by IIR is not used in this method. How different would be the results if this operational retrieval was used instead of the in situ relationship? Is there a reason for not doing this?

Specific comments:

1. Section 2.2: Computing IIR radiances, or converting between optical properties, requires some assumptions on the shape of ice crystals and of the ice particle size distribution. Dubuisson et al. [2008] for instance showed that the IIR brightness temperatures are sensitive to both parameters. Garnier et al. [2012] showed that converting the IIR effective optical depths into absorption optical depths can also depends on ice particle shape assumptions. Do you have an idea if these assumptions have an impact on your N retrievals? Are they accounted for in the uncertainties described in the appendix? Also, are the ice crystal shapes used in the MADA method to compute the in situ relationships consistent with the shapes assumed in the treatment of CALIPSO measurements, and if not is there an impact on the N retrieval and their uncertainties?

2. p. 6 l. 22: By single-layered cloud do you refer to the absence of other ice clouds or also to the absence of a liquid or mixed-phase layer underneath? IIR seems to be able to deal quite well with multi (ice+liquid) layer conditions by adjusting the background radiance. For instance Fig. 10 in Sourdeval et al. [2016] shows that IIR is robust to multi-layer conditions. It would be worth checking if removing this filter makes a big difference in your dataset. Otherwise, not excluding multi-layer scenes would clearly help to increase the global statistical representativity of N retrievals.

3. Sec. 2.4: Can error bars be added to the points in Fig. 2-4? It seems like these instrumental and computational errors could easily encompass the changes due to the choice of including the first size bin or not.

4. Fig. 9: Could you comment on the very wide spread (75-25% percentiles) in the Krämer et al N/IWC vs Tc figure by comparison to what is noted for the CALIPSO data? Is there no spread plotted for the Krämer dataset on the right plots or is it too small to be seen? Also please indicate in the caption if the CALIPSO retrievals correspond to the entire 2008 and 2013 periods.

5. Fig. 10e: As a remark, the slightly negative relation between N and T (N decreasing towards low temperature) indicated in the Krämer et al. [2009] study is not found anymore in revised version of the dataset (not yet published but seen in recent conference presentations by M. Krämer et al). This should strengthen the statement p. 31 l. 4 that differences are likely to be due to different cloud sampling.

6. Sec. 5.4: Is there an explanation to the fact that Re has a very different dependence on Tc and Tc - Ttop by comparison to what was previously shown for N?

7. Section 6.4: p. 45 l. 11-13: Another possible explanation to the differences in absolute numbers could be that DARDAR-LIM ignores the concentrations of ice particles smaller than $5\,\mu$m. p. 45 l. 14-16: As mentioned before, the slight decrease of N towards low T as noted by Krämer et al. [2009] is not found in the most recent version of their dataset, which is not anymore inconsistent with the relation shown in Gryspeerdt et al. [2018]. An increase of N towards low T is also consistent with what is shown in Figures 10ab of this manuscript. Overall, comparisons between the N retrievals presented in this study and DARDAR-LIM are very difficult as both methods are based on different approaches and difference instruments. DARDAR-LIM also retrieves vertical profiles of N whereas IIR retrievals correspond to weighted N average values from cloud top. Nevertheless, it is quite remarkable that despite all these differences the two dataset show such similar results. This clearly strengthens the confidence in both satellite products.

8. Fig. 17: It would be interesting to see the spatial distribution of the frequency of occurrence of retrievals and of the distance from cloud top (in terms of temperature) corresponding to this figure.

9. p. 49 l. 9-10: The "four formulations" have not yet been mentioned in the conclusion. It would be useful to briefly describe again in what they differ.

Technical corrections:

1. p. 2 l. 13 and 16: it would be better to explicitly refer to "ice clouds" instead of "clouds"

2. p. 11 l. 17: "limit is 1.031"

3. p. 43 l. 15: "m.s$^{-1}$ in"

**References**

P. Dubuisson, V. Giraud, J. Pelon, B. Cadet, and P. Yang. Sensitivity of thermal infrared radiation at the top of the atmosphere and the surface to ice cloud microphysics. *J. Appl. Meteor. and Clim.*, 47 (10):2545–2560, 2008. doi: 10.1175/2008JAMC1805.1.

A. Garnier, J. Pelon, P. Dubuisson, M. Faivre, O. Chomette, N. Pascal, and D. P. Kratz. Retrieval of cloud properties using CALIPSO imaging infrared radiometer. Part I: Effective emissivity and optical depth. *J. Appl. Meteor. and Clim.*, 51(7):1407–1425, 2012. doi: 10.1175/JAMC-D-11-0220.1.

E. Gryspeerdt, O. Sourdeval, J. Quaas, J. Delanoë, and P. Kühne. Ice crystal number concentration estimates from lidar-radar satellite retrievals. part 2: Controls on the ice crystal number concentration. *Atmos. Chem. Phys. Discuss.*, 2018:1–25, 2018. doi: 10.5194/acp-2018-21.

M. Krämer, C. Schiller, A. Afchine, R. Bauer, I. Gensch, A. Mangold, S. Schlicht, N. Spelten, N. Sitnikov, S. Borrmann, M. de Reus, and P. Spichtinger. Ice supersaturations and cirrus cloud crystal numbers. *Atmos. Chem. Phys.*, 9(11):3505–3522, 2009. doi: 10.5194/acp-9-3505-2009.

D. L. Mitchell, A. Garnier, M. Avery, and E. Erfani. Calipso observations of the dependence of homo- and heterogeneous ice nucleation in cirrus clouds on latitude, season and surface condition. *Atmos. Chem. Phys. Discuss.*, 2016:1–60, 2016. doi: 10.5194/acp-2016-1062.

O. Sourdeval, L. C. Labonnote, A. J. Baran, J. Mülmenstädt, and G. Brogniez. A methodology for simultaneous retrieval of ice and liquid water cloud properties. Part 2: Near-global retrievals and evaluation against A-Train products. *Quart. J. Roy. Meteor. Soc.*, 142(701):3063–3081, 2016. doi: 10.1002/qj.2889.

---

## Referee Comment (RC2) · Anonymous Referee #2 · 21 Aug 2018

This is an excellent paper and should be published. My comments are not damning and may be ignored. This author group is at the forefront of IR remote sensing and microphysics. And this paper is a major addition to their portfolio. The manuscript by Mitchell et al. aimed at retrieving ice particle number concentrations in single-layer semitransparent cirrus clouds (optical depth between 0.3 and 3) from satellite observations. A relationship established between and N/IWC from field campaigns were applied to the CALIPSO CALIOP/IIR derived to estimate N, the number concentrations. The authors justified their method by comparing derived from satellite observations from CALIPSO and field campaigns, as well as comparing the results from other independent in-situ measurements. This retrieval method was then applied to two years of CALIPSO IIR data and findings from global and seasonal scales were discussed. I think this is a

well-written paper and the novel technique presented can be accepted for publication.

The manuscript is quite long. If this is an issue, one could consider splitting up the retrieval/radiative aspects and the interpretation of the results.

The following are my and a colleague's comments:

Section 2.2.0. How do you handle the lapse through the cloud. Cirrus can be extensive in their vertical dimension. Are you errors larger for geometrically thick cloud?

Section 2.2.0: You mentioned you used the 0.55 micron extinction derived mean cloud temperature. I would expect that the cloud weighting functions are a bit different for the 10.6 and 12 micron observations and these may also differ from each other. Does this matter? Or this effect absorbed in the Blackbody radiance calculation in 2.2.3?

Section 2.2.2: What does the bias look like between model and observations before correction?

Section 2.3: You reference Yang (2005) and make some mention of habits on page 10. Are allowing habit to be a free parameter or are discouraging people from using habits at all in prescribing properties from databases such as Yang's? This relevant information to the remote sensing community.

Figs. 2 and 3 and so on: Why are computed from the TC4 campaign mostly concentrated in regions less than 1.1?

Figs. 8a and 8b: In these figures, IIR median and in situ compare relatively well. Can the authors show an example scatterplot of the comparisons?

Section 5: The selection criteria resulted in less than 2% of qualified pixels. This makes me wonder if the selection requirement is relaxed to include more cirrus clouds, how much do the results in Figs. 11-16 change?

Figure 11. Your highest Beta_Eff occur at the highest clouds where your sample size is often relatively small. Is that an issue? Or is the "hom" effect.

Figure 17: Comparing the two figures, can the authors explain why some peak regions remain similar magnitude such as in southwest Southern America, but many weaken significantly for instance over the Arctic Ocean East of Greenland?

[Figure]

---

## Author Comment (AC2) · 27 Oct 2018

RESPONSES TO THE REFEREE #2 COMMENTS We thank the referee for his/her comments of this paper, and for constructive comments that have significantly improved this paper. We understand that this is a serious undertaking that requires considerable time and effort, and your efforts are appreciated!

In the pdf supplement, black font is used for the referee comments (RC) and blue font is used for author comments (AC) and the new text added to the paper. Also, many figures are included in our response, and these did not transfer with the text pasted below; please refer to the pdf supplement to view these figures, along with our entire response.

[Figure]

General comments:

RC The manuscript is quite long. If this is an issue, one could consider splitting up the retrieval/radiative aspects and the interpretation of the results.

AC We appreciate the referee's recognition of the substantial research effort that produced this manuscript. In regards to the manuscript length, we created another file titled Supplementary Materials that contains 12 figures and some of the text from the original manuscript, which shortened the paper by 8 pages.

Specific comments:

RC S1. Section 2.2.0. How do you handle the lapse through the cloud. Cirrus can be extensive in their vertical dimension. Are you errors larger for geometrically thick cloud?

AC S1. In Eq. (3), the lapse through the cloud is handled in the determination of the cloud blackbody radiance, RBB. The associated blackbody temperature, TBB, is determined using the approach detailed in Sect. 3 of Garnier et al. (2015) and summarized in Sect. 2.2.3 of this paper. First, we compute the IIR weighting profile in the cloud layer using the CALIOP extinction profile. The IIR weighting profile includes an attenuation term corresponding to the overlying infrared absorption optical depth (see Eq. (11) and Eq. (12) of Garnier et al. (2015)). Then, RBB is the weighted averaged blackbody radiance computed using this IIR weighting profile and the GMAO temperature profile in the layer. We actually forgot to precise that the GMAO temperature profile is used, and the 3rd sentence in Sect. 2.2.3 now reads (changes in bold):

"The CALIOP lidar 532 nm extinction profile in the cloud is used to determine an IIR weighting profile that is used, together with the **GMAO GEOS5 temperature profile**, to compute RBB as the weighted averaged blackbody radiance."

RC S2. Section 2.2.0: You mentioned you used the 0.55 micron extinction derived mean cloud temperature. I would expect that the cloud weighting functions are a bit

different for the 10.6 and 12 micron observations and these may also differ from each other. Does this matter? Or this effect absorbed in the Blackbody radiance calculation in 2.2.3?

AC S2. For this study, TBB is taken identical at 10.6 and 12 $\mu$m, following the same approach as in the Version 3 algorithm, but is improved compared to Version 3 by using the corrections detailed in Garnier et al. (2015) and summarized in Sect 2.2.3. Thus, TBB is computed at 12 $\mu$m by taking the ratio, r, between visible extinction optical depth and absorption optical depth at 12 $\mu$m, Tabs(12.05$\mu$m), equal to 2. We agree that TBB could have been estimated separately at 12 $\mu$m (noted TBB,12) and at 10.6 $\mu$m (noted TBB,10). Furthermore, taking the same ratio, r, for both channels means taking Tabs(12.05$\mu$m) =Tabs(10.6$\mu$m) or $\beta$eff=1, which is not consistent with our findings. In order to assess the error resulting from our simplified approach, we re-computed TBB,12(r12) and TBB,10(r10), with r12 and r10 not taken equal to 2, but computed respectively using Tabs(12.05$\mu$m) and Tabs(10.6$\mu$m) initially reported in the operational Version 3 products. The analysis was conducted over oceans in JJA 2013 between 82°S and 82°N. a) We find that the difference TBB,12(r12)-TBB,12(r12=2) (see Table 1 below) is smaller than 0.09 K on average with a mean absolute deviation smaller than 0.13 K. The resulting error is negligible compared to the assumed uncertainty of $\pm$ 2K in TBB,12(r12=2). b) We find that the difference TBB,10(r10) - TBB,12(r12) (see Table 2 below) is smaller than 0.12 K on average with a mean absolute deviation smaller than 0.07 K. c) Using TBB,10(r10) and TBB,12(r12) instead of the same temperature TBB=TBB,12 (r12=2) as in this study reduces $\beta$eff by less than 0.001 on average, with a mean absolute deviation smaller than 0.0007, which is negligible (see Table 3).

We added the following sentence at the end of the 1st paragraph in Sect. 2.2.3: "Computing TBB at 10.6 $\mu$m and at 12.05 $\mu$m yields temperatures that differ by less than 0.15 K on average, which has a negligible impact on $\beta$eff for our cloud selection."

Table 1: Analysis of the difference TBB,12(r12) - TBB,12(r12=2). Oceans, JJA 2013, 82°S-82°N. Temperature Tc (K) Samples Count Min Max Median Mean Standard deviation Mean absolute deviation 188 73 0.012 0.082 0.063 0.061 0.017 0.013 192 348 -0.038 0.501 0.056 0.071 0.066 0.044 196 2149 -0.685 0.486 0.076 0.087 0.085 0.056 200 6915 -0.719 1.879 0.083 0.084 0.129 0.081 204 16783 -1.098 3.048 0.063 0.061 0.160 0.100 208 28620 -1.246 2.181 0.048 0.043 0.182 0.118 212 39959 -3.193 3.299 0.022 0.011 0.199 0.123 216 45591 -1.374 3.680 0.004 -0.017 0.199 0.127 220 41081 -1.606 5.351 -0.015 -0.040 0.202 0.123 224 29724 -0.878 1.984 -0.021 -0.044 0.160 0.102 228 10377 -0.644 1.664 -0.018 -0.026 0.124 0.074 232 1084 -0.303 1.087 -0.001 -0.000 0.088 0.040

Table 2: Analysis of the difference $T_{BB,10}(r10)$ - $T_{BB,12}(r12)$. Oceans, JJA 2013, 82°S-82°N. Temperature $T_c$ (K) Samples Count Min Max Median Mean Standard deviation Mean absolute deviation 188 73 0.016 0.039 0.032 0.031 0.005 0.004 192 348 0.000 0.221 0.040 0.048 0.034 0.028 196 2149 -0.025 0.345 0.051 0.066 0.050 0.038 200 6915 -0.27 0.667 0.083 0.093 0.060 0.046 204 16783 -0.018 0.729 0.095 0.113 0.078 0.059 208 28620 -0.277 1.155 0.098 0.118 0.086 0.066 212 39959 -1.792 0.763 0.091 0.110 0.085 0.063 216 45591 -0.164 0.601 0.080 0.097 0.071 0.055 220 41081 -0.392 0.622 0.064 0.077 0.058 0.044 224 29724 -0.741 0.768 0.044 0.052 0.041 0.030 228 10377 -0.162 0.251 0.025 0.029 0.025 0.018 232 1084 -0.367 0.093 0.010 0.012 0.026 0.010

Table 3: Analysis of the difference between $\beta$eff computed using $T_{BB,10}(r10)$ and $T_{BB,12}(r12)$ and $\beta$eff from this study. Oceans, JJA 2013, 82°S-82°N. Temperature $T_c$ (K) Samples Count Min Max Median Mean Standard deviation Mean absolute deviation 188 73 -0.0001 -0.0000 -0.0000 -0.0000 0.0000 0.0000 192 348 -0.0005 0.0000 -0.0001 -0.0001 0.0001 0.0001 196 2149 -0.0041 0.0001 -0.0001 -0.0002 0.0003 0.0002 200 6915 -0.0048 0.0013 -0.0003 -0.0004 0.0004 0.0003 204 16783 -0.0081 0.0026 -0.0005 -0.0006 0.0006 0.0005 208 28620 -0.0215 0.0080 -0.0006 -0.0008 0.0009 0.0006 212 39959 -0.0220 0.0551 -0.0007 -0.0009 0.001 0.0007 216 45591 -0.0159 0.0075 -0.0007 -0.001 0.001 0.0007 220 41081 -0.0311 0.0129 -0.0007 -0.001 0.001 0.0007 224 29724 -0.0338 0.0319 -0.0006 -0.0008 0.0009 0.0006 228 10377

-0.0109 0.0069 -0.0004 -0.0005 0.0007 0.0004 232 1084 -0.0029 0.0176 -0.0002 -0.0002 0.0008 0.0002

RC S3. Section 2.2.2: What does the bias look like between model and observations before correction?

AC S3. Probability density functions (PDFs) of the differences between clear sky observations and model (BTDoc) at 12.05 $\mu$m before (night: light blue; day: orange) and after (night: navy blue; day: red) correction are shown below for six latitude bands over ocean (left) and over land (right) in January 2008. This figure is now Fig. S1a under Supplementary Materials.

Figure S1a: Probability density functions of the differences between clear sky observations and computations of brightness temperature (BTDoc) at 12.05 $\mu$m, before (night: light blue; day: orange) and after (night: navy blue; day: red) correction are shown below for six latitude bands over ocean (left) and over land (right) in January 2008. Note the different scales over ocean and over land.

Similarly, PDFs of the clear sky inter-channel differences [BTDoc(10.6 $\mu$m)– BTDoc(12.05 $\mu$m)] are shown below, and the figure is now Fig. S1b under Supplementary Materials.

Figure S1b: Same as Fig. S1a, but for the inter-channel difference between observations and computations, BTDoc(10.6 $\mu$m)-BTDoc(12.05 $\mu$m).

The following sentence has been added at the end of Sect. 2.2.2: "Figures S1a and S1b in Supplementary Materials show distributions of BTDoc before and after correction".

RC S4. Section 2.3: You reference Yang (2005) and make some mention of habits on page 10. Are allowing habit to be a free parameter or are discouraging people from using habits at all in prescribing properties from databases such as Yang's? This relevant information to the remote sensing community.

AC S4. We have added a paragraph describing the retrieval's relative insensitivity to ice particle shape at the end of Sect. 2.3. As described in this paragraph (shown below), there is no inferred "ice particle shape recipe" that the retrieval is based on. It would be difficult to apply a database such as Yang's to in situ data since ideally one would need to know the habit composition of the ice particle size distribution (PSD), where habit varies with size across the PSD. Such information is difficult to extract from in situ probe measurements, although it should be possible using the Cloud Particle Imager (CPI) and suitable image analysis software. The new paragraph reads as follows:

"Cirrus cloud emissivity and $\tau$abs depend on ice particle shape (Mitchell et al., 1996; Dubuisson et al. 2008). However, this retrieval should not be very sensitive to ice particle shape for several reasons, one being that $\beta$eff is directly retrieved from cloud radiances as per (2) and (3). Another reason is that no ice particle shape assumptions are made when calculating $\beta$eff from in situ measurements with the exception of the absorption contribution from tunneling (which was not sensitive to realistic shape changes, as described above). That is, the 2D-S probe in situ data include measurements and estimates for ice particle projected area and mass, respectively. MADA optical properties are calculated directly from these in situ area and mass values, thus largely avoiding the need for shape assumptions. Thirdly, this retrieval is most sensitive to the smaller ice particles in a PSD where the variance in ice particle shape is minimal (Baker and Lawson, 2006b; Lawson et al., 2006b; Woods et al., 2018). During the SPARTICUS campaign, many cirrus clouds were sampled so that biases in ice particle shape due to a specific cloud condition are less likely to occur."

RC S5. Figs. 2 and 3 and so on: Why are computed from the TC4 campaign mostly concentrated in regions less than 1.1?

AC S5. As mentioned in Sect. 2.3, PSD sampling times were longer during TC4 (relative to SPARTICUS) with fewer sampling days, resulting in fewer PSD samples. Of these, only aged anvil cirrus sampled on one day by the WB57 were sampled at T < -60‰C (Mitchell et al., 2011, JGR). TC4 cirrus sampled at warmer temperatures had

substantially broader PSDs that yield lower $\beta$eff values, typically < 1.10. New text has been added to the second-to-last paragraph in Sect. 2.3:

"There are much fewer TC4 points in Figs. 2 and 3 for $\beta$eff > 1.1 since the higher $\beta$eff values were obtained only for T < -60⁰C, which only occurred during a single flight."

RC S6. Figs. 8a and 8b: In these figures, IIR median and in situ compare relatively well. Can the authors show an example scatterplot of the comparisons?

AC S6. Because spatially and temporally coincident measurements are very rare for cloudy scenes meeting the IIR cloud selection criteria, these figures show statistical analyses of in situ data on one hand, and of IIR data on the other hand, and are not one-to-one comparisons. Therefore, we don't think that we can show a scatterplot of the comparisons. We added the following sentence at the end of the 1st paragraph in Section 4.1:

"Data analysis is performed on a statistical basis, as coincident in situ and satellite data only provide a very small dataset due to our data selection."

RC S7: Section 5: The selection criteria resulted in less than 2% of qualified pixels. This makes me wonder if the selection requirement is relaxed to include more cirrus clouds, how much do the results in Figs. 11-16 change?

AC S7. The rationale for selecting the relevant cloudy scenes for this study is presented in Sect. 2.2.1. We tried to clarify, and Sect. 2.2.1 now reads (changes are in bold): "Because IIR is a passive instrument, meaningful retrievals are possible for well identified scenes. This study is restricted to the cases where the atmospheric column contains one cirrus cloud layer. We also insure that the background radiance is only due to the surface (see Eq. (3)) allowing a more accurate computation than for cloudy scenes. The retrievals were applied only to single-layered semi-transparent cirrus clouds that do not fully attenuate the CALIOP laser beam, so that the cloud base is detected by the lidar. The cloud base is in the troposphere and its temperature is required to be colder than -38°C (235 K) to ensure that the cloud is entirely composed of ice. This is likely to exclude liquid-origin cirrus clouds from our data set (Luebke et al., 2016). When the column contains also a dense water cloud, the background radiance can be computed assuming that the water cloud is a blackbody. However, because systematic biases were made evident (Garnier et al., 2012), we chose to discard these cases, which reduces the number of selected samples by about 25 %. Because the relative uncertainties in $\tau$abs and in $\beta$eff increase very rapidly as cloud emissivity decreases (Garnier et al., 2013), the lidar layer-integrated attenuated backscatter (IAB) was chosen greater than 0.01 sr-1 to avoid very large uncertainties at the smallest visible optical depths (ODs). This resulted in an OD range of about 0.3 to 3.0. Similarly, clouds for which the radiative contrast RBG -RBB between the surface and the cloud is less than 20 K in brightness temperature units are discarded. IIR observations must be of good quality according to the quality flag reported in the IIR Level 2 product (Vaughan et al., 2017)."

Frequency of occurrence could indeed be increased by relaxing some selection criteria, but the difficulty is that the additional information could be obscured by large additional uncertainties. As an illustration, we reprocessed the dataset to include clouds of OD between 0.1 and 0.3. Fig. S5 under Supplementary Materials show the interrelationships between $\beta$eff, $\alpha$ext, IWC, and N as well as $\Delta$N/N for OD > 0.1, and is compared to Fig. S4 (former Fig. 7) obtained using the chosen threshold OD > 0.3. Figure S5 is copied below:

Figure S5: Same as Fig. S4 but sample selection criteria was changed to accept samples having OD > 0.1 approximately. Note the larger portion of samples having $\Delta$N/N > 1.

The following text has been added under Supplementary Materials: "This same analysis was repeated in Fig. S5, except the sample selection criteria for minimum OD was changed from 0.3 to 0.1. This increased the sample population considerably. The larger dispersion in $\beta$eff and in particular the larger portion of samples with $\beta$eff much

smaller than 1 (Fig. S5, top row) are due to large uncertainties at OD between 0.1 and 0.3, which also explain the larger portion of samples with $\Delta N/N$ >1 (Fig. S5, bottom row). More samples now correspond to lower values of $\alpha$ext, (down to 0.016 km-1), IWC, and N. "

And at the end of Sect. 3.2, we added: "We repeated this analysis except using an OD threshold of 0.1 (instead of 0.3; see also Sect. 5.1). Figure S5 shows this same analysis except that the sample selection criteria for minimum OD was changed from 0.30 to 0.10. In the lower row relating $\Delta N/N$ to $\beta$eff, the number of samples having $\Delta N/N$ > 1.0 has substantially increased over both ocean and land relative to Fig. S4 due to the lower OD threshold for sample selection, and more samples correspond to lower values of $\alpha$ext, IWC, and N."

Table 4 in Sect. 5.1 has been updated to show frequency of occurrence when OD>0.1. The following new text has been added at the end of the 3rd paragraph in Sect. 5.1: "Relaxing the lower OD threshold to 0.1 instead of 0.3 would increase the number of samples by about a factor of 2.5."

The equivalent of Fig. 9 (previous Fig. 12a) using a OD threshold of 0.1 instead of 0.3 is now Fig. S10 in the Supplementary Materials. This figure is shown below:

Fig. S10. Same as Fig. 9 in the main paper, but by relaxing the OD threshold to OD > 0.1.

This figure is now introduced at the end of Sect. 5.2 where the following text is added:

"Fig. S10 under Supplementary Materials, shows the same results as in Fig. 9. but by relaxing the OD threshold to OD > 0.1. Consistent with Fig. S5, median N is decreased, by a factor 1.5 on average, while $\Delta N/N$ is more than doubled." A similar statement was added to Summary and Conclusions at the end of the 4th paragraph.

As an illustration, but not included in the paper, below are the equivalent of previous Fig. 11 and Figs. 12b to 16 using an OD threshold of 0.1 instead of 0.3.

Same as Fig. 8 (previous Fig. 11), but with OD>0.1. The OD threshold does not change much IIR $\beta$eff over ocean. The changes are more notable over land, which could be due in part to larger additional errors than over ocean.

Same as Fig. S9 (previous Fig. 12b), but with OD> 0.1.

Same as Fig. 10 (previous Fig. 13), but with OD>0.1. The relative variations are similar in both figures, with N being smaller when samples of OD between 0.1 and 0.3 are included.

Same as Fig. S11 (previous Fig. 14), but with OD>0.1

Same as Fig. 11 (previous Fig. 15), but with OD>0.1. Consistent with Fig. 8, the changes in median De are more notable over land. Larger additional errors occur more over land than over oceans.

Same as Fig. S12 (previous Fig. 16), but with OD>0.1. Again, the changes in median De are more notable over land than over ocean.

RC S8. Figure 11. Your highest Beta_Eff occur at the highest clouds where your sample size is often relatively small. Is that an issue? Or is the "hom" effect.

AC S8. It's more likely the "hom effect". That is, we consistently see higher $\beta$eff at colder temperatures and higher altitudes with the in situ PSD from field campaigns. This is where hom can produce the highest N, resulting in the smallest crystals that produce the highest $\beta$eff. New text has been added at the end of the 3rd paragraph in Sect. 5.2:

"At the coldest temperatures (highest Zc), hom should be more frequent, resulting in smaller ice crystals and thus higher $\beta$eff."

RC S9. Figure 17: Comparing the two figures, can the authors explain why some peak regions remain similar magnitude such as in southwest Southern America, but many weaken significantly for instance over the Arctic Ocean East of Greenland?

AC S9. It is important to recognize that while the color legend in Fig. 12 (previously Fig. 17) changes color in increments of 50 L-1, the 1st color bin is from zero to 100 L-1 and the last color bin is from 500 L-1 to infinity. When comparing the two plots, regions corresponding to the highest N values may not change much due to this legend convention.

New text has been added to the end of the 1st paragraph in Sect. 6.4: "While the color legend in Fig. 12 changes color in increments of 50 L-1, the 1st color bin is from zero to 100 L-1 and the last color bin is from 500 L-1 to infinity. When comparing the two plots, regions corresponding to the highest N values may not change much due to this legend convention."

Please also note the supplement to this comment:
https://www.atmos-chem-phys-discuss.net/acp-2018-526/acp-2018-526-AC2-supplement.pdf

---

## Author Response (AR1)

**AUTHOR RESPONSES TO REFEREE #1 COMMENTS**

We thank the referee for his/her review of this manuscript, and for constructive comments that have significantly improved this paper.  We understand that this is a serious undertaking that requires considerable time and effort, and your efforts are appreciated!

In the pdf supplement, black font is used for the referee comments (RC) and blue font is used for author comments (AC) and new text added to the paper.

Following our responses is the MS Word version of our revised manuscript with all changes indicated via "Track Changes".

General comments:

RC G1. A major concern with the current manuscript is its length. The authors cover a very wide range of information and results that are all interesting but their amount is, in my opinion, counterproductive. It should be kept in mind that this study will be of high interest to diverse communities, as it contains are interesting aspects for remote sensing, in situ measurements, modeling or ice microphysics. Based on the current version I suspect that readers will only focus on their section of interest and miss important results. One idea could be to shorten section 2, since a big part of it has already been described in Mitchell et al. [2016]. The results related to the N(D)1 =0 analyses are very important and should be mentioned in the paper but several of them could be moved to supplementary materials in order to lighten the discussions and the figures. Other analyses that are not focused on N (the novelty of the paper), such as section 5.4, could also be moved to supplementary materials. This would allow to remove some technical discussions and focus the paper on result analyses, which are also better suited for a publication in ACP.

AC G1. Both referees commented on the paper's length being too long, although the content was appropriate.  To remedy this problem, we moved 11 of the original figures and two new figures to a new file titled "Supplementary Materials".  Some of the original text was also moved to Supplementary Materials.  This shortened the paper's length by about 8 pages.

RC G2. The authors acknowledge in the manuscript that this new retrieval method is not expected to provide accurate absolute values of N. Only the spatial variability of this parameter is well represented. The accuracy of the absolute N is discussed throughout sections 3 and 4 but would it be possible to summarize a final estimate of the uncertainties on N in the conclusion? By this I mean not only the $\Delta N/N$ but also combining what has been learned from the in situ evaluation. For instance, would it be a factor of 2, 5, one order of magnitude? Also, in what conditions can optimal retrievals be expected? This type of information would be very useful to future users of this dataset, especially since this paper will serve as a reference.

AC G2. In the 1st paragraph of the new Sect. 3.2 (previously Sect. 3.3), we write that ΔN is computed assuming a negligible error in the relationships.

We added the following sentence to clarify that: *"Errors in the relationships create additional systematic uncertainties, as discussed in Sect. 3.1."*

As suggested by the reviewer, the following text is added in the 2nd paragraph of Summary and Conclusions:

*"Four formulations of the retrieval were used, based on either the SPARTICUS or TC4 field campaigns, with the smallest size-bin of the 2D-S probe either assumed valid ($N(D)_1$ unmodified) or by assuming $N(D)_1=0$. The SPARTICUS unmodified $N(D)_1$ assumption gives the highest N values while the TC4 $N(D)_1 = 0$ assumption yielded the lowest N values. The N predicted from these two formulations differed by a factor of two (see Fig. 5), thus defining a possible systematic uncertainty in N. The random relative uncertainty, $\Delta N/N$, associated to a N retrieval is most of the time < 0.5 for small crystals ($D_e$ < 35-50 µm), but increases up to more than 2 at the sensitivity limit"*.

In addition, the definition of $\beta_{eff}$ is now repeated at the beginning of the conclusion and the sensitivity ranges of the retrievals is summarized at the end of the 1st paragraph of the conclusion as:
*"Perhaps the most unique aspect of this retrieval method is its sensitivity to small ice crystals via $\beta_{eff}$. The sensitivity ranges (N/IWC > ~ $10^7$ $g^{-1}$ & $D_e$ < 90-110 µm) are usually compatible with cirrus clouds (T < -38 °C) since PSDs tend to be narrower at these temperatures, containing relatively small ice particles."*

Finally, we note at the end of the 4th paragraph under Summary and Conclusions: "When the sample selection criteria were relaxed to accept samples having cloud OD > 0.1 (instead of OD > 0.3), median N was reduced by a factor of 1.5 on average while ΔN/N was more than doubled." This is also mentioned earlier in the paper at the end of Sect. 5.2.

RC G3. A clear limitation of the method is its necessity to filter out a lot of data based on different cloud conditions. This, as shown in table 4, leads to samples that are only representative of up to a 2% frequency of occurrence. Is there a particular reason for not using more data?

AC G3. The rationale for selecting the relevant cloudy scenes for this study is presented in Sect. 2.2.1. We tried to clarify, and Sect. 2.2.1 now reads (changes are in bold):

*"Because IIR is a passive instrument, meaningful retrievals are possible for well identified scenes. This study is restricted to the cases where the atmospheric column contains one cirrus cloud layer.  We also insure that the background radiance is only due to the surface (see Eq. (3)) allowing a more accurate computation than for cloudy scenes.*  The retrievals were applied only to single-layered semi-transparent cirrus clouds that do not fully attenuate the CALIOP laser beam, so that the cloud base is detected by the lidar.  The cloud base is in the troposphere and its temperature is required to be colder than -38°C (235 K) to ensure that the cloud is entirely composed of ice.  This is likely to exclude liquid-origin cirrus clouds from our data set (Luebke et al., 2016).  **When the column contains also a dense water cloud, the background radiance can be computed assuming that the water cloud is a blackbody. However, because systematic biases were made evident (Garnier et al., 2012), we chose to discard these cases, which reduces the number of selected samples by about 25 %.** Because the relative uncertainties in $\tau_{abs}$ and in $\beta_{eff}$ increase very rapidly as cloud emissivity decreases (Garnier et al., 2013), the lidar layer-integrated attenuated backscatter (IAB) was chosen greater than 0.01 sr$^{-1}$ to avoid very large uncertainties at the smallest visible optical depths (ODs).  This resulted in an OD range of about 0.3 to 3.0.  Similarly, clouds for which the radiative contrast $R_{BG}$ -$R_{BB}$ between the surface and the cloud is less than 20 K in brightness temperature units are discarded.  IIR observations must be of good quality according to the quality flag reported in the IIR Level 2 product (Vaughan et al., 2017)."*

Frequency of occurrence could be increased by relaxing some selection criteria, but the difficulty is that the additional information could be obscured by large additional uncertainties. Following reviewer #2's suggestion, we reprocessed the data set to include clouds of OD between 0.1 and 0.3. This increased the frequency of occurrence by a factor 2 to 3 (as now shown in Table 4), but this also increased the relative uncertainty in IIR $\beta_{eff}$ and in CALIPSO N. As noted under AC G2, we have reported how this affects N.

The method seems to be rather straightforward to apply on CALIPSO satellite products, and some of the co-authors should be familiar with such operational treatments. It therefore shouldn't take long to process 10 years of data and greatly improve the statistical significance of the results shown in this study (especially in sections 5 and 6).

The first paragraph of Sect. 2.2 has been shortened and modified to clarify that IIR $\beta_{eff}$ is based on the CALIPSO IIR Version 3 Level 2 operational product, but that the various improvements implemented for this study required to reprocess the dataset. Processing of other years will be interesting for instance to examine inter-annual variability, and will be carried out with the new version 4 of the operational products. The end of the first paragraph of Sect. 2.2 now reads:

*"IIR effective emissivity was reprocessed for this study to reduce possible biases, as described in the following sub-sections.  [……….]. These improvements will be implemented in the next version 4 of the IIR Level 2 products.  This is why we only focus in this study on a limited dataset*

*as a proof-of-concept. We will further develop a more statistically relevant analysis using the version 4 products."*

RC G4. Concerning the results shown in section 4 (comparisons to in situ measurements), more direct comparisons between SPARTICUS measurements and CALIPSO could be obtained by looking at the numerous co-incident overpasses between the satellite and the aircraft. This would avoid issues related to non-representative events in the in situ data by comparison to a long-term regional dataset, as noted in page 22. Have you tried looking at these exact satellite overpasses only?

AC G4. Yes, but unfortunately, we found that spatially and temporally coincident measurements are very rare for scenes meeting the IIR cloud selection criteria. We added the following sentence at the end of the 1$^{st}$ paragraph in Section 4.1:

*"Data analysis is performed on a statistical basis, as coincident in situ and satellite data only provide a very small dataset due to our data selection."*

RC G5. Perhaps I have missed this information but I am under the impression that the De retrieved by IIR is not used in this method. How different would be the results if this operational retrieval was used instead of the in situ relationship? Is there a reason for not doing this?

AC G5. The fundamental parameter retrieved from IIR is $\beta_{eff}$. In the CALIPSO retrieval equation of N (Eq. (8), Sect. 2.5), IIR $\beta_{eff}$ is related to N/IWC, $D_e$, and $2/Q_{abs,eff}$ using regression curves derived from aircraft PSD measurements (Sect. 2.3 and Sect. 2.4). Indeed, the three relationships used in Eq. (8) must be derived in a consistent manner from the same in situ PSDs.

IIR $D_e$ reported in the Version 3 data products is derived using the $\beta_{eff}$-$D_e$ relationships presented in Garnier et al. (2013) computed using different optical properties, with no size distribution, and by using both the 12.05/10.6 and 12.05/08.65 pairs of channels. Because the technique and the assumptions are different than in this study, we could not use $D_e$ from the Version 3 product for this study. Furthermore, IIR data have been reprocessed for this study in order to implement the improvement described in Sect. 2.2.2 and 2.2.3, yielding different $\beta_{eff}$ in this study than in the Version 3 products, especially over land.

We removed the comments about IIR $D_e$ in Sect. 2.2 to avoid confusion and added the following text to the end of the 2$^{nd}$ paragraph of Sect. 2.4:

*"It is noted that the $D_e$- $\beta_{eff}$ relationships used in the IIR Version 3 operational algorithm (Fig. 3a in Garnier et al., 2013) tend to yield smaller $D_e$ than the relationships derived from these PSD measurements, largely because they were computed with no size distribution."*

Specific comments:

RC S1. Section 2.2: Computing IIR radiances, or converting between optical properties, requires some assumptions on the shape of ice crystals and of the ice particle size distribution.
10   Dubuisson et al. [2008] for instance showed that the IIR brightness temperatures are sensitive to both parameters. Garnier et al. [2012] showed that converting the IIR effective optical depths into absorption optical depths can also depend on ice particle shape assumptions. Do you have an idea if these assumptions have an impact on your N retrievals? Are they accounted for in the uncertainties described in the appendix?

AC S1. We clarified the presentation of IIR $\beta_{eff}$ at the beginning of Sect. 2.2, which now reads:

*"We use the absorption optical depth $\tau_{abs}$(12.05 μm) and $\tau_{abs}$(10.6 μm) retrieved from the effective emissivity in CALIPSO IIR channels 12.05 μm and 10.6 μm. The retrieved optical depths are not purely due to absorption, but also include the effects of scattering. Thus, their ratio is*
20   *not exactly β, but the effective β or $\beta_{eff}$ written as*

    $\beta_{eff} = \tau_{abs}$*(12.05 μm)/$\tau_{abs}$(10.6 μm) ."*                                   *(1)"*

The weak dependence of the retrieval on ice particle shape is described in a new paragraph at the end of Sect. 2.3:

*"Cirrus cloud emissivity and $\tau_{abs}$ depend on ice particle shape (Mitchell et al., 1996a; Dubuisson et al. 2008).  However, this retrieval should not be very sensitive to ice particle shape for several reasons, one being that $\beta_{eff}$ is directly retrieved from cloud radiances as per (2) and (3).  Another reason is that no ice particle shape assumptions are made when calculating $\beta_{eff}$ from in situ*
30   *measurements with the exception of the absorption contribution from tunneling (which was not sensitive to realistic shape changes, as described above).  That is, the 2D-S probe in situ data include measurements and estimates for ice particle projected area and mass, respectively. MADA optical properties are calculated directly from these in situ area and mass values, thus largely avoiding the need for shape assumptions.  Thirdly, this retrieval is most sensitive to the*
35   *smaller ice particles in a PSD where the variance in ice particle shape is minimal (Baker and Lawson, 2006b; Lawson et al., 2006b; Woods et al., 2018).  During the SPARTICUS campaign, many cirrus clouds were sampled so that biases in ice particle shape due to a specific cloud condition are less likely to occur."*

The uncertainties in N described in the appendix are random errors computed assuming no error in the relationships. Errors in the relationships create additional systematic uncertainty (see AC G2), but the dependence on ice crystal shapes should be very weak, as stated above.

Also, are the ice crystal shapes used in the MADA method to compute the in situ relationships consistent with the shapes assumed in the treatment of CALIPSO measurements, and if not is there an impact on the N retrieval and their uncertainties?

10    CALIPSO IIR $\beta_{eff}$ is computed directly from the effective emissivity at 12.05 µm and 10.6 µm. This computation does not involve any assumption about the shapes of ice crystals. As such, CALIPSO IIR $\beta_{eff}$ can be seen as a direct observation. CALIPSO N is then retrieved from CALIPSO IIR $\beta_{eff}$ and the in situ relationships presented in this manuscript.

15    RC S2. p. 6 l. 22: By single-layered cloud do you refer to the absence of other ice clouds or also to the absence of a liquid or mixed-phase layer underneath? IIR seems to be able to deal quite well with multi (ice+liquid) layer conditions by adjusting the background radiance. For instance Fig. 10 in Sourdeval et al. [2016] shows that IIR is robust to multi-layer conditions. It would be worth checking if removing this filter makes a big difference in your dataset. Otherwise, not
20    excluding multi-layer scenes would clearly help to increase the global statistical representativity of N retrievals.

AC S2. The rationale for our cloud selection presented in Sect. 2.2.1 has been modified (see AC G3).
25    We confirm that the retrievals are applied when a cirrus cloud is the only cloud layer in the atmospheric column, that is when the background radiance is from the surface.
We could have also included scenes for which the cirrus cloud layer is over a low opaque water cloud that fully attenuates the CALIOP laser beam. Indeed, in this second case, the background radiance can be computed assuming that the low water cloud behaves as a blackbody emitting
30    at a temperature inferred from the water cloud altitude and GMAO GEOS temperature profiles. The expected accuracy in this second case was evaluated in Garnier et al. (2012) through comparisons of observed and computed blackbody brightness temperatures of low opaque warm clouds. The observations were colder than the blackbody computations by up to 9 K (their Fig. 10), suggesting significant biases in the cloud temperature inferred from the GMAO
35    GEOS profiles in case of strong temperature inversions at the top of the cloud.
We chose to discard these scenes for this study, which reduces the number of samples by about 25 %. These biases will be re-evaluated with the future version 4 of the IIR products which will use MERRA 2 profiles data.

RC S3. Sec. 2.4: Can error bars be added to the points in Fig. 2-4? It seems like these instrumental and computational errors could easily encompass the changes due to the choice of including the first size bin or not.

5      AC S3. The data points in Figs. 2-4 come from in situ PSD measurements.  There is only one PSD per point, and therefore it is not possible to assign an error bar from a statistical basis.  One could in principle propagate errors associated with the various stages of instrument data processing (which perhaps is what the referee is referring to), but uncertainties for these data processing stages have not been characterized in most cases.  For example, in "Laboratory and
10     Flight Tests of 2D Imaging Probes: Toward a Better Understanding of Instrument Performance and the Impact on Archived Data" by Gurganus and Lawson (2018, JTech), the 2D-S and the CIP probes are intercompared in various ways, but uncertainties are not estimated that could be applied to address this request.  This paper has now been referenced to give the reader additional knowledge of 2D-S performance characteristics.  As stated at the end of this paper,
15     "The research reported here does not offer quantitative corrections for the instrument errors that have been uncovered.  Instead, it describes conditions under which the uncertainties can exist and provides some qualitative estimates of magnitude. Corrections to the data will require more laboratory tests with upgraded facilities, flight tests, improved forward models of the instruments and numerical simulations."

RC S4. Fig. 9: Could you comment on the very wide spread (75-25% percentiles) in the Krämer et al N/IWC vs Tc figure by comparison to what is noted for the CALIPSO data? Is there no spread plotted for the Krämer dataset on the right plots or is it too small to be seen? Also please indicate in the caption if the CALIPSO retrievals correspond to the entire 2008 and 2013
25     periods.

AC S4. To shorten the paper, this figure has been moved to Supplementary Materials and is Fig. S6.  The caption states that "in situ measurements from Krämer et al. (2009)" are "shown by the grey curves; top and bottom being minimum and maximum values and middle grey solid
30     curve being the middle value."  Thus, these dashed Kramer curves are not the 25 and 75 percentile curves, but rather indicate the max/min values.  To reduce confusion, we have identified curves corresponding to the four formulations as colored.  We have indicated that percentiles corresponding to the 4 formulations are not shown in the right panel, and that the retrievals shown are from 2013.

The caption now reads: *"Left: Comparisons of the median CALIPSO IIR N/IWC (g$^{-1}$) for the four formulations (colored) with in situ measurements from Krämer et al. (2009) shown by the grey curves; top and bottom being minimum and maximum values and middle grey solid curve being the middle value.  Colored solid curves are median values while dashed curves indicate the 25$^{th}$*
40     *and 75$^{th}$ percentile values.  Right:  Comparisons of CALIPSO IIR β$_{eff}$ shown by the black curves (solid curve gives the median value while dashed curves indicate the 25$^{th}$ and 75$^{th}$ percentile*

*values) with the four (colored) in situ $\beta_{eff}$ inferred from in situ N/IWC (from Krämer et al. (2009) using the four formulations. Corresponding percentiles are not shown. The navy and light blue curves correspond to the SPARTICUS formulations for the unmodified $N(D)_1$ assumption and the $N(D)_1 = 0$ assumption, respectively. The red and orange curves are using the TC4 formulations for the $N(D)_1$ unmodified and $N(D)_1 = 0$ assumptions, respectively. The CALIPSO IIR retrievals are from 2013 and are for the approximate latitude range (25 °S to 70 °N) of the in situ data, over oceans (top) and over land (bottom)."*

RC S5. Fig. 10e: As a remark, the slightly negative relation between N and T (N decreasing towards low temperature) indicated in the Kramer et al. [2009] study is not found anymore in revised version of the dataset (not yet published but seen in recent conference presentations by M. Krämer et al). This should strengthen the statement p. 31 l. 4 that differences are likely to be due to different cloud sampling.

AC S5. We cannot comment on this since the data is not yet published.

RC S6. Sec. 5.4: Is there an explanation to the fact that Re has a very different dependence on Tc and Tc - Ttop by comparison to what was previously shown for N?

AC S6. In the new Fig. 10 (previously Fig. 13), there is a tendency for N to decrease from left-to-right as a function of $T_c - T_{top}$. In the new Fig. S12 (previously Fig. 16), $D_e$ tends to decrease with decreasing $T_c$ but remains quasi-constant (or less variable) for a given $T_c$ (i.e. less sensitive to $T_c - T_{top}$). $D_e$ is proportional to IWC/$A_{PSD}$ (as calculated for the in situ data and thus implemented in the retrieval regressions), where IWC = ice water content and $A_{PSD}$ = projected area of the ice particle size distribution (PSD). For a gamma function PSD of the form $N(D) = N_o D^v \exp(-\lambda D)$ and representing ice particle projected area as $A = \gamma D^\sigma$ and mass as $m = \alpha D^\beta$, $A_{PSD} = \gamma \Gamma(\sigma + v + 1) N/ \Gamma(v + 1) \lambda^\sigma$ and IWC = $\alpha \Gamma(\beta + v + 1) N/ \Gamma(v + 1) \lambda^\beta$ where $\Gamma$ denotes the gamma function. The IWC/$A_{PSD}$ ratio now shows that $D_e$ is independent of N, and $D_e$ is proportional to $\lambda^{\sigma-\beta}$ where $\lambda$ = PSD slope parameter. The difference $\sigma - \beta$ is slightly more than -1 (Mitchell 1996, JAS), showing that $D_e$ is strongly related to the PSD mean size [note that $\lambda = (v + 1)/D_{mean}$, where $D_{mean}$ = PSD mean maximum dimension]. The decrease in $D_e$ with decreasing $T_c$ is likely due to a decrease in IWC with decreasing $T_c$.

New text has been added to the end of the last paragraph in Sect. 5.4 to express these points:
*"This weaker dependence on $T_c - T_{top}$ relative to N is expected since $D_e$ is proportional to IWC/$A_{PSD}$. For a gamma function PSD and assuming power law relationships for ice particle area and mass, $D_e$ is proportional to $\lambda^{\sigma-\beta}$ where $\lambda$ = PSD slope parameter and $\sigma$ and $\beta$ are power law exponents for area and mass, respectively. The difference $\sigma - \beta$ is slightly more than -1, showing that $D_e$ is strongly related to the PSD mean size (inversely proportional to $\lambda$). The derivation also shows that $D_e$ is not dependent on N. The decrease in $D_e$ with decreasing $T_c$ is likely due to a decrease in IWC with decreasing $T_c$."*

RC S7.  Section 6.4: p. 45 l. 11-13: Another possible explanation to the differences in absolute numbers could be that DARDAR-LIM ignores the concentrations of ice particles smaller than 5 µm.  p. 45 l. 14-16: As mentioned before, the slight decrease of N towards low T as noted by Krämer et al. [2009] is not found in the most recent version of their dataset, which is not anymore inconsistent with the relation shown in Gryspeerdt et al. [2018]. An increase of N towards low T is also consistent with what is shown in Figures 10ab of this manuscript. Overall, comparisons between the N retrievals presented in this study and DARDAR-LIM are very difficult as both methods are based on different approaches and difference instruments. DARDAR-LIM also retrieves vertical profiles of N whereas IIR retrievals correspond to weighted N average values from cloud top. Nevertheless, it is quite remarkable that despite all these differences the two dataset show such similar results. This clearly strengthens the confidence in both satellite products.

AC S7. Section 6.4 has been changed significantly in view of the referee's suggestions.  We agree with the referee that absolute differences in N between the DARDAR-LIM and our CALIPSO IIR retrieval may be partly attributed to the fact that DARDAR-LIM (as per the ACP paper) does not account for ice crystals < 5 µm in maximum dimension D.  Fortunately, there are DARDAR-LIM results posted on the ACP website as part of the review process for the DARDAR-LIM paper that show N for D > 1 µm.  We have cited this work to show that using a D > 1 µm cutoff increases DARDAR-LIM N by a factor of about 1.7, making the CALIPSO IIR N retrievals in general agreement with the DARDAR-LIM N values.  New text has been added accordingly:

 "*The highest N values reported in Sourdeval et al. (2018a) and Gryspeerdt et al. (2018) are for ice particles larger than 5 µm.  These values would be higher by a factor of about 1.7 if all ice particles larger than 1 µm were considered (Sourdeval et al., 2018b).  This is important to note since the CALIPSO IIR retrieval has no cut-off size, and it is very sensitive to ice crystals in the 1 to 5 µm range (Fig. 1).  Keeping this in mind, the CALIPSO N retrievals are comparable in value to the DARDAR-LIM N retrievals.*"

Regarding the temperature dependence of N, although we cannot cite unpublished work, we appreciate this information and have removed the discussion about the temperature dependence of in situ N measurements (in relation to retrievals).

We agree that there is good qualitative agreement between our CALIPSO IIR and the DARDAR-LIM retrievals, and that this "strengthens the confidence in both satellite products."

RC S8. Fig. 17: It would be interesting to see the spatial distribution of the frequency of occurrence of retrievals and of the distance from cloud top (in terms of temperature) corresponding to this figure.

AC S8.  Maps of the number of samples (left) and of Median $T_c$-$T_{top}$ (right) associated with the new Fig. 12 (previously Fig. 17 ), with $T_c$ between 218 K and 228 K, are shown below:

[Figure]

These figures have been added to the Supplementary Materials section of the manuscript as Fig. S13.  New text has been added in the 1$^{st}$ paragraph in Sect. 6.4:

*"The geographical distributions of the number of samples and of median $T_c - T_{top}$ corresponding to Fig. 12 are shown in Fig. S13 under Supplementary Materials.  Median $T_c$-$T_{top}$ is up to 20 °C and is smaller at mid- to high latitudes than in the tropics."*

The 2$^{nd}$ paragraph in Sect. 6.4 starts now as: *"Results in Fig. 12 can be compared with the retrieved N values near cloud top at 223 K shown in Fig.1 of Gryspeerdt et al. (2018)."*

The reviewer's question led us to comment briefly on the typical relationship between $T_c$-$T_{top}$ and $T_{base}$-$T_{top}$. Thus, isolines of $T_{base}$-$T_{top}$ have been added to Fig. 10 and Fig. S11, and the text has been modified accordingly in Sect. 5.3 (changes in bold):

*"Our retrievals are now examined against both $T_c$ and $T_c$-$T_{top}$ to estimate the impact of the distance from cloud top. **These** N retrievals are shown in Fig. **10** using the SPARTICUS N(D)$_1$ unmodified assumption in the tropics (0-30°) and at mid- (30-60°) and high (60-82°) latitudes in the winter and summer seasons (using both hemispheres) by distinguishing retrievals over oceans and over land.  The associated number of samples is given in Fig. **S11**.  **Added to these figures are isolines of temperature differences between cloud base and cloud top ($T_{base}$-$T_{top}$). For most of the layers, $T_c$-$T_{top}$ represents 30 to 70% of $T_{base}$-$T_{top}$ (Fig. S11).  Fig. 10** shows a*

*strong dependence of N on $T_c$-$T_{top}$, with large N (> 500 $L^{-1}$) seen near the top of the* **geometrically thin clouds ($T_{base}$-$T_{top}$ < 15 °C)**, *when $T_c$-$T_{top}$ is smaller than about 5 °C."*

RC S9. p. 49 l. 9-10: The "four formulations" have not yet been mentioned in the conclusion. It would be useful to briefly describe again in what they differ.

AC S9.  The "four formulations" of the retrieval are now mentioned under "Summary and Conclusions" as follows:

*"Four formulations of the retrieval were used, based on either the SPARTICUS or TC4 field campaigns, with the smallest size-bin of the 2D-S probe either assumed valid (N(D)$_1$ unmodified) or by assuming N(D)$_1$=0. The SPARTICUS unmodified N(D)$_1$ assumption gives the highest N values while the TC4 N(D)$_1$ = 0 assumption yielded the lowest N values.  The N predicted from these two formulations differed by a factor of two (see Fig. 5), thus defining a possible systematic uncertainty in N."*

Technical corrections:
1. p. 2 l. 13 and 16: it would be better to explicitly refer to \ice clouds" instead of \clouds"
Done
2. p. 11 l. 17: \limit is 1.031"
Done
3. p. 43 l. 15: \m.s$^{-1}$ in"
Text removed

**AUTHOR RESPONSES TO REFEREE #2 COMMENTS**

We thank the referee for his/her comments of this paper, and for constructive comments that have significantly improved this paper.  We understand that this is a serious undertaking that requires considerable time and effort, and your efforts are appreciated!

In the text below, black font is used for the referee comments (RC) and blue font is used for author comments (AC) and new text added to the paper.

Following our responses is the MS Word version of our revised manuscript with all changes
10   indicated via "Track Changes".

General comments:

RC The manuscript is quite long. If this is an issue, one could consider splitting up the
15   retrieval/radiative aspects and the interpretation of the results.

AC We appreciate the referee's recognition of the substantial research effort that produced this manuscript.  In regards to the manuscript length, we created another file titled Supplementary Materials that contains 12 figures and some of the text from the original manuscript, which
20   shortened the paper by 8 pages.

Specific comments:

RC S1.  Section 2.2.0. How do you handle the lapse through the cloud. Cirrus can be extensive
25   in their vertical dimension. Are you errors larger for geometrically thick cloud?

AC S1.  In Eq. (3), the lapse through the cloud is handled in the determination of the cloud blackbody radiance, $R_{BB}$. The associated blackbody temperature, $T_{BB}$, is determined using the approach detailed in Sect. 3 of Garnier et al. (2015) and summarized in Sect. 2.2.3 of this paper.
30   First, we compute the IIR weighting profile in the cloud layer using the CALIOP extinction profile. The IIR weighting profile includes an attenuation term corresponding to the overlying infrared absorption optical depth (see Eq. (11) and Eq. (12) of Garnier et al. (2015)). Then, $R_{BB}$ is the weighted averaged blackbody radiance computed using this IIR weighting profile and the GMAO temperature profile in the layer. We actually forgot to precise that the GMAO
35   temperature profile is used, and the 3$^{rd}$ sentence in Sect. 2.2.3 now reads (changes in bold):

*"The CALIOP lidar 532 nm extinction profile in the cloud is used to determine an IIR weighting profile that is used, **together with the GMAO GEOS5 temperature profile,** to compute $R_{BB}$ as the weighted averaged blackbody radiance."*

RC S2. Section 2.2.0: You mentioned you used the 0.55 micron extinction derived mean cloud temperature. I would expect that the cloud weighting functions are a bit different for the 10.6 and 12 micron observations and these may also differ from each other. Does this matter? Or this effect absorbed in the Blackbody radiance calculation in 2.2.3?

AC S2. For this study, $T_{BB}$ is taken identical at 10.6 and 12 µm, following the same approach as in the Version 3 algorithm, but is improved compared to Version 3 by using the corrections detailed in Garnier et al. (2015) and summarized in Sect 2.2.3. Thus, $T_{BB}$ is computed at 12 µm by taking the ratio, r, between visible extinction optical depth and absorption optical depth at 12 µm, $\tau_{abs}$(12.05µm), equal to 2.

We agree that $T_{BB}$ could have been estimated separately at 12 µm (noted $T_{BB,12}$) and at 10.6 µm (noted $T_{BB,10}$). Furthermore, taking the same ratio, r, for both channels means taking $\tau_{abs}$(12.05µm) =$\tau_{abs}$(10.6µm) or $\beta_{eff}$=1, which is not consistent with our findings. In order to assess the error resulting from our simplified approach, we re-computed $T_{BB,12}(r_{12})$ and $T_{BB,10}(r_{10})$, with $r_{12}$ and $r_{10}$ not taken equal to 2, but computed respectively using $\tau_{abs}$(12.05µm) and $\tau_{abs}$(10.6µm) initially reported in the operational Version 3 products. The analysis was conducted over oceans in JJA 2013 between 82°S and 82°N.

a) We find that the difference $T_{BB,12}(r_{12})$-$T_{BB,12}(r_{12}=2)$ (see Table 1 below) is smaller than 0.09 K on average with a mean absolute deviation smaller than 0.13 K. The resulting error is negligible compared to the assumed uncertainty of ± 2K in $T_{BB,12}(r_{12}=2)$.

b) We find that the difference $T_{BB,10}(r_{10})$ - $T_{BB,12}(r_{12})$ (see Table 2 below) is smaller than 0.12 K on average with a mean absolute deviation smaller than 0.07 K.

c) Using $T_{BB,10}(r_{10})$ and $T_{BB,12}(r_{12})$ instead of the same temperature $T_{BB}=T_{BB,12}$ $(r_{12}=2)$ as in this study reduces $\beta_{eff}$ by less than 0.001 on average, with a mean absolute deviation smaller than 0.0007, which is negligible (see Table 3).

We added the following sentence at the end of the 1$^{st}$ paragraph in Sect. 2.2.3:

"*Computing $T_{BB}$ at 10.6 µm and at 12.05 µm yields temperatures that differ by less than 0.15 K on average, which has a negligible impact on $\beta_{eff}$ for our cloud selection.*"

Table 1: Analysis of the difference $T_{BB,12}(r_{12})$ - $T_{BB,12}(r_{12}=2)$. Oceans, JJA 2013, 82°S-82°N.

| Temperature $T_c$ (K) | Samples Count | Min | Max | Median | Mean | Standard deviation | Mean absolute deviation |
|---|---|---|---|---|---|---|---|
| 188 | 73 | 0.012 | 0.082 | 0.063 | 0.061 | 0.017 | 0.013 |
| 192 | 348 | -0.038 | 0.501 | 0.056 | 0.071 | 0.066 | 0.044 |
| 196 | 2149 | -0.685 | 0.486 | 0.076 | 0.087 | 0.085 | 0.056 |
| 200 | 6915 | -0.719 | 1.879 | 0.083 | 0.084 | 0.129 | 0.081 |

| | | | | | | | |
|---|---|---|---|---|---|---|---|
| 204 | 16783 | -1.098 | 3.048 | 0.063 | 0.061 | 0.160 | 0.100 |
| 208 | 28620 | -1.246 | 2.181 | 0.048 | 0.043 | 0.182 | 0.118 |
| 212 | 39959 | -3.193 | 3.299 | 0.022 | 0.011 | 0.199 | 0.123 |
| 216 | 45591 | -1.374 | 3.680 | 0.004 | -0.017 | 0.199 | 0.127 |
| 220 | 41081 | -1.606 | 5.351 | -0.015 | -0.040 | 0.202 | 0.123 |
| 224 | 29724 | -0.878 | 1.984 | -0.021 | -0.044 | 0.160 | 0.102 |
| 228 | 10377 | -0.644 | 1.664 | -0.018 | -0.026 | 0.124 | 0.074 |
| 232 | 1084 | -0.303 | 1.087 | -0.001 | -0.000 | 0.088 | 0.040 |

Table 2: Analysis of the difference $T_{BB,10}(r_{10})$ - $T_{BB,12}(r_{12})$. Oceans, JJA 2013, 82°S-82°N.

| Temperature $T_c$ (K) | Samples Count | Min | Max | Median | Mean | Standard deviation | Mean absolute deviation |
|---|---|---|---|---|---|---|---|
| 188 | 73 | 0.016 | 0.039 | 0.032 | 0.031 | 0.005 | 0.004 |
| 192 | 348 | 0.000 | 0.221 | 0.040 | 0.048 | 0.034 | 0.028 |
| 196 | 2149 | -0.025 | 0.345 | 0.051 | 0.066 | 0.050 | 0.038 |
| 200 | 6915 | -0.27 | 0.667 | 0.083 | 0.093 | 0.060 | 0.046 |
| 204 | 16783 | -0.018 | 0.729 | 0.095 | 0.113 | 0.078 | 0.059 |
| 208 | 28620 | -0.277 | 1.155 | 0.098 | 0.118 | 0.086 | 0.066 |
| 212 | 39959 | -1.792 | 0.763 | 0.091 | 0.110 | 0.085 | 0.063 |
| 216 | 45591 | -0.164 | 0.601 | 0.080 | 0.097 | 0.071 | 0.055 |
| 220 | 41081 | -0.392 | 0.622 | 0.064 | 0.077 | 0.058 | 0.044 |
| 224 | 29724 | -0.741 | 0.768 | 0.044 | 0.052 | 0.041 | 0.030 |
| 228 | 10377 | -0.162 | 0.251 | 0.025 | 0.029 | 0.025 | 0.018 |
| 232 | 1084 | -0.367 | 0.093 | 0.010 | 0.012 | 0.026 | 0.010 |

Table 3: Analysis of the difference between $\beta_{eff}$ computed using $T_{BB,10}(r_{10})$ and $T_{BB,12}(r_{12})$ and $\beta_{eff}$ from this study. Oceans, JJA 2013, 82°S-82°N.

| Temperature $T_c$ (K) | Samples Count | Min | Max | Median | Mean | Standard deviation | Mean absolute deviation |
|---|---|---|---|---|---|---|---|
| 188 | 73 | -0.0001 | -0.0000 | -0.0000 | -0.0000 | 0.0000 | 0.0000 |
| 192 | 348 | -0.0005 | 0.0000 | -0.0001 | -0.0001 | 0.0001 | 0.0001 |
| 196 | 2149 | -0.0041 | 0.0001 | -0.0001 | -0.0002 | 0.0003 | 0.0002 |
| 200 | 6915 | -0.0048 | 0.0013 | -0.0003 | -0.0004 | 0.0004 | 0.0003 |
| 204 | 16783 | -0.0081 | 0.0026 | -0.0005 | -0.0006 | 0.0006 | 0.0005 |
| 208 | 28620 | -0.0215 | 0.0080 | -0.0006 | -0.0008 | 0.0009 | 0.0006 |
| 212 | 39959 | -0.0220 | 0.0551 | -0.0007 | -0.0009 | 0.001 | 0.0007 |
| 216 | 45591 | -0.0159 | 0.0075 | -0.0007 | -0.001 | 0.001 | 0.0007 |
| 220 | 41081 | -0.0311 | 0.0129 | -0.0007 | -0.001 | 0.001 | 0.0007 |
| 224 | 29724 | -0.0338 | 0.0319 | -0.0006 | -0.0008 | 0.0009 | 0.0006 |
| 228 | 10377 | -0.0109 | 0.0069 | -0.0004 | -0.0005 | 0.0007 | 0.0004 |
| 232 | 1084 | -0.0029 | 0.0176 | -0.0002 | -0.0002 | 0.0008 | 0.0002 |

RC S3.  Section 2.2.2: What does the bias look like between model and observations before correction?

AC S3. Probability density functions (PDFs) of the differences between clear sky observations and model (BTDoc) at 12.05 µm before (night: light blue; day: orange) and after (night: navy blue; day: red) correction are shown below for six latitude bands over ocean (left) and over land (right) in January 2008. This figure is now Fig. S1a under Supplementary Materials.

[Figure]

**Figure S1a: Probability density functions of the differences between clear sky observations and computations of brightness temperature (BTDoc) at 12.05 µm, before (night: light blue; day: orange) and after (night: navy blue; day: red) correction are shown below for six latitude bands over ocean (left) and over land (right) in January 2008. Note the different scales over ocean and over land.**

Similarly, PDFs of the clear sky inter-channel differences [BTDoc(10.6 µm)– BTDoc(12.05 µm)] are shown below, and the figure is now Fig. S1b under Supplementary Materials.

[Figure]

**Figure S1b: Same as Fig. S1a, but for the inter-channel difference between observations and computations, BTDoc(10.6 μm)-BTDoc(12.05 μm).**

The following sentence has been added at the end of Sect. 2.2.2: "*Figures S1a and S1b in Supplementary Materials show distributions of BTDoc before and after correction*".

RC S4.  Section 2.3: You reference Yang (2005) and make some mention of habits on page 10. Are allowing habit to be a free parameter or are discouraging people from using habits at all in prescribing properties from databases such as Yang's? This relevant information to the remote sensing community.

AC S4.  We have added a paragraph describing the retrieval's relative insensitivity to ice particle shape at the end of Sect. 2.3.  As described in this paragraph (shown below), there is no inferred "ice particle shape recipe" that the retrieval is based on.  It would be difficult to apply a database such as Yang's to in situ data since ideally one would need to know the habit composition of the ice particle size distribution (PSD), where habit varies with size across the PSD.  Such information is difficult to extract from in situ probe measurements, although it should be possible using the Cloud Particle Imager (CPI) and suitable image analysis software. The new paragraph reads as follows:

*"Cirrus cloud emissivity and $\tau_{abs}$ depend on ice particle shape (Mitchell et al., 1996; Dubuisson et al. 2008).  However, this retrieval should not be very sensitive to ice particle shape for several reasons, one being that $\beta_{eff}$ is directly retrieved from cloud radiances as per (2) and (3).  Another reason is that no ice particle shape assumptions are made when calculating $\beta_{eff}$ from in situ measurements with the exception of the absorption contribution from tunneling (which was not sensitive to realistic shape changes, as described above).  That is, the 2D-S probe in situ data include measurements and estimates for ice particle projected area and mass, respectively. MADA optical properties are calculated directly from these in situ area and mass values, thus largely avoiding the need for shape assumptions.  Thirdly, this retrieval is most sensitive to the smaller ice particles in a PSD where the variance in ice particle shape is minimal (Baker and Lawson, 2006b; Lawson et al., 2006b; Woods et al., 2018).  During the SPARTICUS campaign, many cirrus clouds were sampled so that biases in ice particle shape due to a specific cloud condition are less likely to occur."*

RC S5.  Figs. 2 and 3 and so on: Why are computed from the TC4 campaign mostly concentrated in regions less than 1.1?

AC S5.  As mentioned in Sect. 2.3, PSD sampling times were longer during TC4 (relative to SPARTICUS) with fewer sampling days, resulting in fewer PSD samples.  Of these, only aged anvil cirrus sampled on one day by the WB57 were sampled at T < -60$^{o}$C (Mitchell et al., 2011, JGR). TC4 cirrus sampled at warmer temperatures had substantially broader PSDs that yield lower $\beta_{eff}$ values, typically < 1.10.  New text has been added to the second-to-last paragraph in Sect. 2.3:

*"There are much fewer TC4 points in Figs. 2 and 3 for $\beta_{eff}$ > 1.1 since the higher $\beta_{eff}$ values were obtained only for T < -60$^{o}$C, which only occurred during a single flight."*

RC S6.  Figs. 8a and 8b: In these figures, IIR median and in situ compare relatively well. Can the authors show an example scatterplot of the comparisons?

AC S6. Because spatially and temporally coincident measurements are very rare for cloudy scenes meeting the IIR cloud selection criteria, these figures show statistical analyses of in situ data on one hand, and of IIR data on the other hand, and are not one-to-one comparisons. Therefore, we don't think that we can show a scatterplot of the comparisons.
We added the following sentence at the end of the 1$^{st}$ paragraph in Section 4.1:

*"Data analysis is performed on a statistical basis, as coincident in situ and satellite data only provide a very small dataset due to our data selection."*

RC S7:  Section 5: The selection criteria resulted in less than 2% of qualified pixels. This makes me wonder if the selection requirement is relaxed to include more cirrus clouds, how much do the results in Figs. 11-16 change?

AC S7.  The rationale for selecting the relevant cloudy scenes for this study is presented in Sect. 2.2.1. We tried to clarify, and Sect. 2.2.1 now reads (changes are in bold):

*__"Because IIR is a passive instrument, meaningful retrievals are possible for well identified__*
5 *__scenes. This study is restricted to the cases where the atmospheric column contains one cirrus__*
*__cloud layer.  We also insure that the background radiance is only due to the surface (see Eq.__*
*__(3)) allowing a more accurate computation than for cloudy scenes.__*  *The retrievals were applied only to single-layered semi-transparent cirrus clouds that do not fully attenuate the CALIOP laser beam, so that the cloud base is detected by the lidar.  The cloud base is in the troposphere*
10 *and its temperature is required to be colder than -38°C (235 K) to ensure that the cloud is entirely composed of ice.  This is likely to exclude liquid-origin cirrus clouds from our data set (Luebke et al., 2016).*  *__When the column contains also a dense water cloud, the background__*
*__radiance can be computed assuming that the water cloud is a blackbody. However, because__*
*__systematic biases were made evident (Garnier et al., 2012), we chose to discard these cases,__*
15 *__which reduces the number of selected samples by about 25 %.__*  *Because the relative uncertainties in $\tau_{abs}$ and in $\beta_{eff}$ increase very rapidly as cloud emissivity decreases (Garnier et al., 2013), the lidar layer-integrated attenuated backscatter (IAB) was chosen greater than 0.01 $sr^{-1}$ to avoid very large uncertainties at the smallest visible optical depths (ODs).  This resulted in an OD range of about 0.3 to 3.0.  Similarly, clouds for which the radiative contrast $R_{BG}$ -$R_{BB}$ between the surface and the cloud is less than 20 K in brightness temperature units are discarded.  IIR*
20 *observations must be of good quality according to the quality flag reported in the IIR Level 2 product (Vaughan et al., 2017)."*

Frequency of occurrence could indeed be increased by relaxing some selection criteria, but the
25 difficulty is that the additional information could be obscured by large additional uncertainties. As an illustration, we reprocessed the dataset to include clouds of OD between 0.1 and 0.3. Fig. S5 under Supplementary Materials show the interrelationships between $\beta_{eff}$, $\alpha_{ext}$, IWC, and N as well as $\Delta N/N$ for OD > 0.1, and is compared to Fig. S4 (former Fig. 7) obtained using the chosen threshold OD > 0.3. Figure S5 is copied below:

[Figure]

*Figure S5: Same as Fig. S4 but sample selection criteria was changed to accept samples having OD > 0.1 approximately. Note the larger portion of samples having ΔN/N > 1.*

5    The following text has been added under Supplementary Materials:

*"This same analysis was repeated in Fig. S5, except the sample selection criteria for minimum OD was changed from 0.3 to 0.1. This increased the sample population considerably. The larger dispersion in $\beta_{eff}$ and in particular the larger portion of samples with $\beta_{eff}$ much smaller than 1 (Fig. S5, top row) are due to large uncertainties at OD between 0.1 and 0.3, which also explain*

10    *the larger portion of samples with ΔN/N >1 (Fig. S5, bottom row). More samples now correspond to lower values of $\alpha_{ext}$ (down to 0.016 $km^{-1}$), IWC, and N. "*

And at the end of Sect. 3.2, we added:

*"We repeated this analysis except using an OD threshold of 0.1 (instead of 0.3; see also Sect.*

15    *5.1). Figure S5 shows this same analysis except that the sample selection criteria for minimum OD was changed from 0.30 to 0.10. In the lower row relating ΔN/N to $\beta_{eff}$, the number of*

*samples having ΔN/N > 1.0 has substantially increased over both ocean and land relative to Fig.*
*S4 due to the lower OD threshold for sample selection, and more samples correspond to lower*
*values of $\alpha_{ext}$, IWC, and N."*

5    Table 4 in Sect. 5.1 has been updated to show frequency of occurrence when OD>0.1. The
     following new text has been added at the end of the 3rd paragraph in Sect. 5.1:
     *"Relaxing the lower OD threshold to 0.1 instead of 0.3 would increase the number of samples by*
     *about a factor of 2.5."*

10   The equivalent of Fig. 9 (previous Fig. 12a) using a OD threshold of 0.1 instead of 0.3 is now Fig.
     S10 in the Supplementary Materials. This figure is shown below:

[Figure]

**Fig. S10. Same as Fig. 9 in the main paper, but by relaxing the OD threshold to OD > 0.1.**

15   This figure is now introduced at the end of Sect. 5.2 where the following text is added:

     *"Fig. S10 under Supplementary Materials, shows the same results as in Fig. 9. but by relaxing*
     *the OD threshold to OD > 0.1. Consistent with Fig. S5, median N is decreased, by a factor 1.5 on*
     *average, while ΔN/N is more than doubled."*  A similar statement was added to Summary and
20   Conclusions at the end of the 4th paragraph.

As an illustration, but not included in the paper, below are the equivalent of previous Fig. 11 and Figs. 12b to 16 using an OD threshold of 0.1 instead of 0.3.

[Figure]

Same as Fig. 8 (previous Fig. 11), but with OD>0.1. The OD threshold does not change much IIR $\beta_{eff}$ over ocean. The changes are more notable over land, which could be due in part to larger additional errors than over ocean.

[Figure]

Same as Fig. S9 (previous Fig. 12b), but with OD> 0.1.

[Figure]

5    Same as Fig. 10 (previous Fig. 13), but with OD>0.1. The relative variations are similar in both figures, with N being smaller
     when samples of OD between 0.1 and 0.3 are included.

[Figure]

Same as Fig. S11 (previous Fig. 14), but with OD>0.1

5    Same as Fig. 11 (previous Fig. 15), but with OD>0.1. Consistent with Fig. 8, the changes in median $D_e$ are more notable over land.  Larger additional errors occur more over land than over oceans.

[Figure]

Same as Fig. S12 (previous Fig. 16), but with OD>0.1. Again, the changes in median $D_e$ are more notable over land than over ocean.

RC S8.  Figure 11. Your highest Beta_Eff occur at the highest clouds where your sample size is often relatively small. Is that an issue? Or is the "hom" effect.

AC S8.  It's more likely the "hom effect".  That is, we consistently see higher $\beta_{eff}$ at colder temperatures and higher altitudes with the in situ PSD from field campaigns.  This is where hom can produce the highest N, resulting in the smallest crystals that produce the highest $\beta_{eff}$.  New text has been added at the end of the 3$^{rd}$ paragraph in Sect. 5.2:

"At the coldest temperatures (highest $Z_c$), hom should be more frequent, resulting in smaller ice crystals and thus higher $\beta_{eff}$."

RC S9.  Figure 17: Comparing the two figures, can the authors explain why some peak regions remain similar magnitude such as in southwest Southern America, but many weaken significantly for instance over the Arctic Ocean East of Greenland?

AC S9.  It is important to recognize that while the color legend in Fig. 12 (previously 
[revised manuscript text omitted]

[Figure]

**Figure S1a: Probability density functions of the differences between clear sky observations and computations of brightness temperature (BTDoc) at 12.05 μm, before (night: light blue; day: orange) and after (night: navy blue; day: red) correction are shown below for six latitude bands over ocean (left) and over land (right) in January 2008. Note the different scales over ocean and over land.**

[Figure]

**Figure S1b: Same as Fig. S1a, but for the inter-channel difference between observations and computations, BTDoc(10.6 µm)-BTDoc(12.05 µm).**

**Impact of the smallest size bin in PSD measurements**

Jensen et al. (2013a) argues that the PSD first bin $N(D)_1$ as measured by the 2D-S probe is anomalously high since it tends to be considerably higher than the adjacent size-bins [$N(D)_2$ for example] and that these small ice crystals should rapidly grow or sublimate to larger or smaller sizes (> 15 μm or < 5 μm) due to the relative humidity with respect to ice, $RH_i$, being significantly different than ice saturation ($RH_i = 100\%$). Therefore, $N(D)_1 > N(D)_2$ would imply frequent ice nucleation events to sustain these higher $N(D)_1$ values, which appears unlikely. This argument provided an additional incentive to formulate this retrieval assuming $N(D)_1=0$.

However, there are also physical reasons that argue in favor of assuming that $N(D)_1$ is a valid measurement. For example, if strong competition for water vapor due to a relatively high small ice crystal concentration (e.g. due to a homogeneous ice nucleation event) rapidly reduces the $RH_i$ to ~ 100%, then this relatively high concentration may last for time periods comparable to the lifetime of the cirrus cloud. High ice crystal concentrations (~ 300 to 10,000 $L^{-1}$) associated with $RH_i$ ~ 100% were documented by aircraft measurements in the tropical tropopause layer (TTL), existing in layers ranging from meters to 0.4 km in depth (Jensen et al., 2013b). These layers were embedded within a deeper cirrus cloud having N typically less than 20 $L^{-1}$ (where $RH_i$ was higher). Evidence that $RH_i$ near 100% is common in cirrus clouds is shown in Figs. 6 and 7 of Krämer et al. (2009), where for the relationships most representative of cirrus clouds, the relaxation time τ for $RH_i$ to develop a quasi-steady state (i.e. dynamical equilibrium denoted $RH_{qsi}$) is on the order of 5-10 minutes. $RH_{qsi}$ is where $d(RH_i)/dt \approx 0$, where the rate of vapor uptake by ice approximately balances the rate of supersaturation development. Between the time of initial in-cloud supersaturation (corresponding to cloud formation) and τ, the cloud updraft w tends to be higher than the w occurring after $RH_{qsi}$ is attained. Since cirrus cloud lifetimes tend to be considerably longer than 5-10 minutes, w is relatively low with $RH_i$ ~ 100% for time t > τ. This may explain the relatively high frequencies of occurrence of $RH_i$ near 100% in Fig. 7 of Krämer et al. (2009). It can also be argued that cirrus clouds are formed by atmospheric wave activity, and that $RH_i$ near 100% results from averaging transient wave-induced fluctuations of $RH_i$. However, Fig. 1 in Krämer et al. (2009) shows water vapor concentrations being fairly constant with time over periods of 30 to 50 minutes, while also showing evidence of wave-induced fluctuations in $RH_i$ during another period.

Comparisons between the Fast Cloud Droplet Probe (FCDP) and the 2D-S probe during ATTREX when sampling tropical tropopause layer (TTL) cirrus clouds (Woods et al., 2018, Fig. 3) show that the smallest 2D-S size bin often, but not always, measured a lower $N(D)_1$ than did the FCDP. These clouds were not in close proximity to deep convection and were sustained over relatively long periods.

[Figure]

**Figure S2: Same as Fig. 2 (top) and Fig. 3 (bottom) in the main paper, but in this case the first size-bin of the PSD is not included (i.e. $N(D)_1=0$). The dashed lines in the lower panels are where the curve-fit equations giving $1/D_e$ in $\mu m^{-1}$ are extrapolated (see Table 1 in main paper).**

[Figure]

**Figure S3:** The $\beta_{eff}$ dependence of the term that converts the absorption optical depth $\tau_{abs}$ into visible optical depth in Eq. 7, based on PSDs from SPARTICUS synoptic cirrus (blue squares) and anvils (black squares), and TC4 aged (red diamonds) and fresh (black diamonds) anvils, where the first size-bin is included (top, $N(D)_1$ unmodified), or not included (bottom, $N(D)_1=0$). The larger (smaller) symbols denote PSDs measured at a temperature colder (warmer) than -38°C. The curve-fit equations are for SPARTICUS synoptic cirrus (blue) and for TC4 aged and fresh anvils (red). The dashed lines are where the curve-fit equations are extrapolated (see Table 1 in main paper).

**Relationship between $\beta_{eff}$, $\alpha_{ext}$, IWC, and N**

As seen from Eq. (6), (7) and (8), $\beta_{eff}$ and $\alpha_{ext}$ are the two key parameters retrieved from the CALIPSO IIR to derive N/IWC, IWC, and finally N. The interrelationship between $\beta_{eff}$, $\alpha_{ext}$, IWC, and N is illustrated in Fig. S4 (top row) for the SPARTICUS relationships using the unmodified $N(D)_1$ assumption, which also shows the range encountered for these properties in the selected cloud population. The red dashed lines are where N = 100 L$^{-1}$, 500 L$^{-1}$ and 1000 L$^{-1}$. The pink dashed lines are where IWC = 0.5 mg g$^{-3}$, 5 mg g$^{-3}$, or 30 mg g$^{-3}$. The horizontal red dotted lines for $\beta_{eff} < 1.031$ (or $D_e > 83$ μm) indicate where the retrieval is not sensitive to N/IWC. For $\beta_{eff} < 1.031$, N/IWC is set to its limiting (minimum) value so that N is a priori overestimated in these conditions, but typically smaller than 100 L$^{-1}$. For $\beta_{eff} < 1.031$, $D_e$ is set to 83 μm, as denoted by the horizontal pink lines, and IWC is a priori underestimated for these conditions. For our data selection, $\alpha_{ext}$ is mostly between 0.05 km$^{-1}$ and 5 km$^{-1}$. Large values of N ($> 500$ L$^{-1}$) result from larger values of $\beta_{eff}$ (yielding smaller $D_e$ and much larger N/IWC) and sufficiently large values of $\alpha_{ext}$ so that IWC is sufficiently large for these small values of $D_e$. Low values of N ($< 100$ L$^{-1}$) can be retrieved for small values of $\beta_{eff}$, yet larger than the low limit of 1.031, only if $\alpha_{ext}$ is sufficiently small.

This same analysis was repeated in Fig. S5, except the sample selection criteria for minimum OD was changed from 0.3 to 0.1. This increased the sample population considerably. The larger dispersion in $\beta_{eff}$ and in particular the

larger portion of samples with $\beta_{eff}$ much smaller than 1 (Fig. S5, top row) are due to large uncertainties at OD

between 0.1 and 0.3, which also explain the larger portion of samples with $\Delta N/N > 1$ (Fig. S5, bottom row). More

samples now correspond to lower values of $\alpha_{ext}$ (down to 0.016 km$^{-1}$), IWC, and N.

[Figure]

**Figure S4:** **Top: The interrelationship between $\beta_{eff}$ (X-axis), layer extinction coefficient $\alpha_{ext}$ (km$^{-1}$) (Y-axis, log10 scale), ice water content IWC, and ice particle number concentration N for the SPARTICUS N(D)$_1$ unmodified assumption. The red dashed lines are where N is equal to 100, 500, or 1000 L$^{-1}$. The pink dashed lines are where IWC is equal to 0.5, 5, or 30 mg.m$^{-3}$. Bottom: 2D-distribution of $\beta_{eff}$ (X-axis) and relative uncertainty estimate $\Delta N/N$. The color bar gives the log of number of samples normalized to the maximum value. Relative uncertainty tends to be considerably smaller at larger $\beta_{eff}$ values. Left: ocean; right: land; all latitudes; based on December 2013, January and February 2014.**

[Figure]

**Figure S5:** Same as Fig. S4 but sample selection criteria was changed to accept samples having OD > 0.1 approximately. Note the larger portion of samples having ΔN/N > 1.

**Comparison of N/IWC with the Krämer cirrus dataset**

[revised manuscript text omitted]

---

## Author Response (AR2)

**AUTHOR RESPONSES TO THE CO-EDITOR COMMENTS**

Dear Dr. Garrett,

Thank you for your comments, which have improved the paper.

5    In our responses to your comments below, your comments (CE) are in black font and blue font is used for both author comments (AC) and new text added to the paper.

Following our responses is the MS Word version of our revised manuscript with all changes indicated via "Track Changes", followed by the revised Supplementary Materials.

10   CE 1.  The reference to photon tunneling seems out of place. The Nussenwieg papers that are referenced discuss a wave phenomenon, and for the interactions of light with particles defined by length dimensions, a wave description would be most appropriate. The text on Atmospheric Radiation by Bohren and Clothiaux casts a very dim light on the concept of photon tunneling, and I request removal of any reference to those terms from the article. Wave resonance is fine, especially with a short
15   description of the mechanics of the physics.

AC 1:  As requested, all references to "photon tunneling" were removed and replaced with "wave resonance" (except on one occasion where the word tunneling was used once in reference to the work of Nussenzveig, who consistently uses that term), and tunneling factors or $T_e$ were replaced with resonance factors or $R_e$.  Perhaps one reason Nussenzveig uses the word tunneling is the close analogue
20   to tunneling in quantum mechanics with translational motion (which is sometimes explained via the Heisenberg Uncertainty Principle), and the fact that grazing rays beyond a particle's area cross-section can be absorbed through this process.  Moreover, Nussenzveig's treatment of tunneling is based on his Complex Angular Momentum theory where angular momentum is quantized.  As you know, quantized radiation as predicted by the photoelectric effect is commonly referred to as photons.  In Guimaraes and
25   Nussenzveig (1992, Opt. Commun.), the impact parameter b = $\lambda \Lambda / 2\pi$, where $\lambda$ represents the quantized angular momentum = $\ell + \frac{1}{2}$ and $\Lambda$ = wavelength.  Thus, b (related to the distance over which tunneling occurs) is quantized and depends only on $\Lambda$ for a given $\ell$.

Text was changed as follows in Sect. 2.1 to address this issue and to provide a short description of the process: "For wavelengths between 2.7 and 100 µm, the most critical process parameterized was wave
30   resonance, also referred to as tunneling (e.g. Nussenzveig, 1977; Guimaraes and Nussenzveig, 1992; Nussenzveig, 2002) as it accounts for the scattering/absorption of radiation beyond a particle's physical cross-sectional area.  It is sometimes not recognized that this process has two components; (1) a surface wave component contributing to scattering (Nussenzveig and Wiscombe, 1987) and (2) a trapped wave component "orbiting" or resonating within the particle, experiencing near-total internal reflections at
35   the critical angle (Guimaraes and Nussenzveig, 1992).  It is this second component that contributes to absorption in ice particles (Mitchell, 2000)."

Addressing this concern also produced small changes in text in Sect. 2.3 and 2.4.

CE 2.  Reviewer criticisms about the absence of consideration of shape assumption for small crystals do not seem to have been appropriately addressed. The response that "Thirdly, this retrieval is most sensitive to the smaller ice particles in a PSD where the variance in ice particle shape is minimal" seems at odds with observations of columns, plates, bullets, and even bullet rosettes at sizes <50 um. I do not see how frozen particles that grow through diffusional processes at cold temperatures can be "quasi-spherical", even if they look that way in coarse resolution imagery, or that they can all be similar shape.

AC 2.  The last paragraph in Sect. 2.3 on ice particle shape has been rewritten to address the above comments as follows:

"Cirrus cloud emissivity and $\tau_{abs}$ depend on ice particle shape (Mitchell et al., 1996a; Dubuisson et al. 2008).  However, two factors may reduce the dependence of this retrieval on ice particle shape, one being that $\beta_{eff}$ is directly retrieved from cloud radiances as per (2) and (3).  Another reason is that no ice particle shape assumptions are made when calculating $\beta_{eff}$ from in situ measurements with the exception of the absorption contribution from wave resonance (which was not sensitive to realistic shape changes, as described above).  That is, the 2D-S probe in situ data include measurements and estimates for ice particle projected area and mass, respectively, as noted above.  MADA optical properties are calculated directly from these in situ area and mass values, thus largely avoiding the need for shape assumptions.  In addition, this retrieval is most sensitive to the smaller ice particles in a PSD where ice particle shape is difficult to measure.  If certain ice nucleation mechanisms and environmental conditions promote some types of ice embryos over others (e.g. quasi-spherical vs. hexagonal columns/plates), there may be some dependence on ice crystal shape.  An analysis of small ice crystal shapes in cirrus clouds can be found in Baker and Lawson (2006b), Lawson et al. (2006b) and Woods et al. (2018).  During the SPARTICUS campaign, many cirrus clouds were sampled so that biases in ice particle shape are less likely to occur."

3. The figures are of low resolution. Fits on the figures should be expressed with the appropriate variables, with the appropriate number of significant digits, and in scientific not engineering notation.

AC 3.  The figure resolution was changed from 300 dpi to 600 dpi.  The curve fits in Figs. 2, 3, 4, S2, and S3 have been modified according to the co-editor's recommendation and Table 1 has been updated accordingly, giving the appropriate number of significant digits.

When necessary, changes were made to write the correct symbol for degree Celsius (°C), with a space between the value and the symbol (for instance "-38 °C").  A space was included before and after the mathematical operators "=", "<" or ">".  In Figs. S6 and 7c (Kramer comparisons), the range of latitude now appears as "25° S-70° N", with similar changes in Figs. S1 and S2.  The notation "mg.m$^{-3}$" in Figs. S4

and S5 was changed to "mg m$^{-3}$" (space and no dot).  Capitalization in the headers was changed whenever required.

The title for the Supplementary Materials was deleted since the ACP instructions state that: "Supplements will receive a title page added during the publication process including title ("Supplement of"), authors, and the correspondence email. Therefore, please avoid providing this information in the supplement."

We hope that we have adequately addressed your concerns, and we aspire to carry this research forwards.

Best regards,

David Mitchell

[revised manuscript text omitted]

**Impact of the smallest size bin in PSD measurements**

Jensen et al. (2013a) argues that the PSD first bin $N(D)_1$ as measured by the 2D-S probe is anomalously high since it tends to be considerably higher than the adjacent size-bins [$N(D)_2$ for example] and that these small ice crystals should rapidly grow or sublimate to larger or smaller sizes (> 15 μm or < 5 μm) due to the relative humidity with

5    respect to ice, $RH_i$, being significantly different than ice saturation ($RH_i = 100$ %). Therefore, $N(D)_1 > N(D)_2$ would imply frequent ice nucleation events to sustain these higher $N(D)_1$ values, which appears unlikely. This argument provided an additional incentive to formulate this retrieval assuming $N(D)_1 = 0$.

However, there are also physical reasons that argue in favor of assuming that $N(D)_1$ is a valid measurement. For example, if strong competition for water vapor due to a relatively high small ice crystal concentration (e.g. due to a

10   homogeneous ice nucleation event) rapidly reduces the $RH_i$ to ~ 100%, then this relatively high concentration may last for time periods comparable to the lifetime of the cirrus cloud. High ice crystal concentrations (~ 300 to 10,000 $L^{-1}$) associated with $RH_i$ ~ 100% were documented by aircraft measurements in the tropical tropopause layer (TTL), existing in layers ranging from meters to 0.4 km in depth (Jensen et al., 2013b). These layers were embedded within a deeper cirrus cloud having N typically less than 20 $L^{-1}$ (where $RH_i$ was higher). Evidence that $RH_i$ near 100% is

15   common in cirrus clouds is shown in Figs. 6 and 7 of Krämer et al. (2009), where for the relationships most representative of cirrus clouds, the relaxation time $\tau$ for $RH_i$ to develop a quasi-steady state (i.e. dynamical equilibrium denoted $RH_{qsi}$) is on the order of 5-10 minutes. $RH_{qsi}$ is where $d(RH_i)/dt \approx 0$, where the rate of vapor uptake by ice approximately balances the rate of supersaturation development. Between the time of initial in-cloud supersaturation (corresponding to cloud formation) and $\tau$, the cloud updraft w tends to be higher than the w

20   occurring after $RH_{qsi}$ is attained. Since cirrus cloud lifetimes tend to be considerably longer than 5-10 minutes, w is relatively low with $RH_i$ ~ 100 % for time $t > \tau$. This may explain the relatively high frequencies of occurrence of $RH_i$ near 100 % in Fig. 7 of Krämer et al. (2009). It can also be argued that cirrus clouds are formed by atmospheric wave activity, and that $RH_i$ near 100 % results from averaging transient wave-induced fluctuations of $RH_i$. However, Fig. 1 in Krämer et al. (2009) shows water vapor concentrations being fairly constant with time over

25   periods of 30 to 50 minutes, while also showing evidence of wave-induced fluctuations in $RH_i$ during another period.

Comparisons between the Fast Cloud Droplet Probe (FCDP) and the 2D-S probe during ATTREX when sampling tropical tropopause layer (TTL) cirrus clouds (Woods et al., 2018, Fig. 3) show that the smallest 2D-S size bin often, but not always, measured a lower $N(D)_1$ than did the FCDP. These clouds were not in close proximity to deep

30   convection and were sustained over relatively long periods.

[Figure]

**Figure S2: Same as Fig. 2 (top) and Fig. 3 (bottom) in the main paper, but in this case the first size-bin of the PSD is not included (i.e. $N(D)_1 = 0$). The dashed lines in the lower panel are where the curve-fit equations giving $1/D_e$ in $\mu m^{-1}$ are extrapolated (see Table 1 in main paper).**

[Figure]

**Figure S3: Same as Fig. 4 in the main paper, but in this case the first size-bin of the PSD is not included (i.e. N(D)₁ = 0)..** **The dashed lines are where the curve-fit equations are extrapolated (see Table 1 in main paper).**

**Relationship between $\beta_{eff}$, $\alpha_{ext}$, IWC, and N**

As seen from Eq. (6), (7) and (8), $\beta_{eff}$ and $\alpha_{ext}$ are the two key parameters retrieved from the CALIPSO IIR to derive N/IWC, IWC, and finally N. The interrelationship between $\beta_{eff}$, $\alpha_{ext}$, IWC, and N is illustrated in Fig. S4 (top row) for the SPARTICUS relationships using the unmodified N(D)₁ assumption, which also shows the range encountered for these properties in the selected cloud population. The red dashed lines are where N = 100 L⁻¹, 500 L⁻¹ and 1000 L⁻¹. The pink dashed lines are where IWC = 0.5 mg m⁻³, 5 mg m⁻³, or 30 mg m⁻³. The horizontal red dotted lines for $\beta_{eff}$ < 1.031 (or $D_e$ > 83 μm) indicate where the retrieval is not sensitive to N/IWC. For $\beta_{eff}$ <1.031, N/IWC is set to its limiting (minimum) value so that N is a priori overestimated in these conditions, but typically smaller than 100 L⁻¹. For $\beta_{eff}$ < 1.031, $D_e$ is set to 83 μm, as denoted by the horizontal pink lines, and IWC is a priori underestimated for these conditions. For our data selection, $\alpha_{ext}$ is mostly between 0.05 km⁻¹ and 5 km⁻¹. Large values of N (> 500 L⁻¹) result from larger values of $\beta_{eff}$ (yielding smaller $D_e$ and much larger N/IWC) and sufficiently large values of $\alpha_{ext}$ so that IWC is sufficiently large for these small values of $D_e$. Low values of N (< 100 L⁻¹) can be retrieved for small values of $\beta_{eff}$, yet larger than the low limit of 1.031, only if $\alpha_{ext}$ is sufficiently small.

This same analysis was repeated in Fig. S5, except the sample selection criteria for minimum OD was changed from 0.3 to 0.1. This increased the sample population considerably. The larger dispersion in $\beta_{eff}$ and in particular the larger portion of samples with $\beta_{eff}$ much smaller than 1 (Fig. S5, top row) are due to large uncertainties at OD between 0.1 and 0.3, which also explain the larger portion of samples with ΔN/N >1 (Fig. S5, bottom row). More samples now correspond to lower values of $\alpha_{ext}$ (down to 0.016 km⁻¹), IWC, and N.

[Figure]

**Figure S4: Top:** The interrelationship between $\beta_{eff}$ (X-axis), layer extinction coefficient $\alpha_{ext}$ (km$^{-1}$) (Y-axis, log10 scale), ice water content IWC, and ice particle number concentration N for the SPARTICUS N(D)$_1$ unmodified assumption. The red dashed lines are where N is equal to 100, 500, or 1000 L$^{-1}$. The pink dashed lines are where IWC is equal to 0.5, 5, or 30 mg m$^{-3}$. **Bottom:** 2D-distribution of $\beta_{eff}$ (X-axis) and relative uncertainty estimate $\Delta$N/N. The color bar gives the decimal logarithm of number of samples normalized to the maximum value. Relative uncertainty tends to be considerably smaller at larger $\beta_{eff}$ values. Left: ocean; right: land; all latitudes; based on December 2013, January and February 2014.

[Figure]

**Figure S5: Same as Fig. S4 but the sample selection criteria was changed to accept samples having OD > 0.1 approximately. Note the larger portion of samples having ΔN/N > 1.**

**Comparison of N/IWC with the Krämer cirrus dataset**

Although the cirrus cloud measurements in Krämer et al. (2009) occurred over both land and ocean, no distinction was made in this regard. But since CALIPSO IIR $\beta_{eff}$ uncertainties are greater over land, Fig. S6 separates in situ and satellite retrievals of N/IWC and $\beta_{eff}$ over ocean (top) and land (bottom). CALIPSO values are averaged over all seasons for 2013 and over the latitude range roughly corresponding to the field measurements (25° S to 70° N). Temperature intervals are 4 °C.

Shown in the left panels is N/IWC vs. $T_c$. The N/IWC curve fits describing the in situ measurements of Krämer et al. (2009) are shown by the grey curves, and correspond to the maximum, minimum and middle (i.e. mid-point) value of a cloud property as a function of temperature. They are compared with corresponding retrieved median values, based on our four formulations: SPARTICUS unmodified $N(D)_1$ (solid navy blue), SPARTICUS $N(D)_1 = 0$ (solid light blue), TC4 unmodified $N(D)_1$ (solid red), and TC4 $N(D)_1 = 0$ (solid orange), all derived from IIR $\beta_{eff}$ shown in the right panels (black curves). The dashed curves give the 25[th] and 75[th] percentile retrieval values. Using our four formulations, in situ N/IWC is converted into four in situ $\beta_{eff}$ plotted in the right panels for comparison with IIR $\beta_{eff}$ in black. Comparing both N/IWC and $\beta_{eff}$ allows visualizing the non-linear relationship between N/IWC and $\beta_{eff}$.

Note that the Krämer et al. (2009) data used in Fig. S6 contain several non-zero bins between 5 and 15 microns (i.e. the 1[st] size-bin of the 2DS probe). Thus, the in situ PSD do not conform with the $N(D)_1 = 0$ assumption. However, as shown in Fig. S6 (left panels), the retrieved N/IWC is weakly sensitive to the $N(D)_1$ assumption. Given the above ambiguities and uncertainties, the agreement between the median retrieved and in situ N/IWC is noticeable, especially for both SPARTICUS relationships over land. Both CALIPSO IIR $\beta_{eff}$ and in situ $\beta_{eff}$ are smaller than about 1.25 for temperatures greater than 203 K (-70 °C), in agreement with CALIPSO IIR $\beta_{eff}$ retrieved during SPARTICUS (Fig. 6a in main paper) and during TC4 (Fig. 6b in main paper).

[Figure]

**Fig. S6. Left: Comparisons of the median CALIPSO IIR N/IWC (g⁻¹) for the four formulations (colored) with in situ measurements from Krämer et al. (2009) shown by the grey curves; top and bottom being minimum and maximum values and middle grey solid curve being the middle value. Colored solid curves are median values while dashed curves indicate the 25th and 75th percentile values. Right: Comparisons of CALIPSO IIR β_eff shown by the black curves (solid curve gives the median value while dashed curves indicate the 25th and 75th percentile values) with the four (colored) in situ β_eff inferred from in situ N/IWC (from Krämer et al. (2009)) using the four formulations. Corresponding minimum and maximum values are not shown. The navy and light blue curves correspond to the SPARTICUS formulations for the unmodified N(D)₁ assumption and the N(D)₁ = 0 assumption, respectively. The red and orange curves are using the TC4 formulations for the N(D)₁ unmodified and N(D)₁ = 0 assumptions, respectively. The CALIPSO IIR retrievals are from 2013 and are for the approximate latitude range (25° S to 70° N) of the in situ data, over oceans (top) and over land (bottom).**

[Figure]

**Figure S7. Same as Fig. 7a in the main paper (comparing CALIPSO retrievals with SPARTICUS data), except using the $N(D)_1 = 0$ assumption.**

[Figure]

**Figure S8.** Same as Fig. 7b in the main paper (comparing CALIPSO retrievals with TC4 data), except using the $N(D)_1$ unmodified assumption.

[Figure]

**Figure S9: Median ice particle number concentration N (L$^{-1}$) vs. latitude and representative cloud altitude, Z$_c$, during 2008 and 2013 using three formulations: SPARTICUS N(D)$_1$ = 0 (left), TC4 N(D)$_1$ unmodified (center), and TC4 N(D)$_1$ = 0 (right). Panels from top to bottom are for DJF over oceans, DJF over land, JJA over oceans, and JJA over land.**

[Figure]

**Fig. S10. Same as Fig. 9 in the main paper, but by relaxing the OD threshold to OD > 0.1.**

[Figure]

**Figure S11. Samples count vs. the representative cloud temperature $T_c$ and $T_c$ - $T_{top}$ at 0-30° (TRO, left), 30-60° (MID, center), and 60-82° (HIGH, right) during 2008 and 2013. Overplotted are isolines of $T_{base}$ - $T_{top}$ (solid: 15 K, dashed: 25 K; dotted: 35 K). Panels from top to bottom are for winter over oceans, winter over land, summer over oceans, and summer over land.**

[Figure]

**Figure S12: Median retrieved $D_e$ (µm) using the SPARTICUS $N(D)_1$ unmodified formulation vs. the representative cloud temperature $T_c$ and $T_c$ - $T_{top}$ at 0-30° (TRO, left), 30-60° (MID, center), and 60-82° (HIGH, right) during 2008 and 2013. Panels from top to bottom are for winter over oceans, winter over land, summer over oceans, and summer over land.**

[Figure]

**Figure S13: Geographical distribution of (left) number of samples and (right) median $T_c - T_{top}$ values during 2008 and 2013 where the cloud layer representative temperature, $T_c$, is between 218 and 228 K.**